



# Snow Particle Motion in Process of Cornice Formation

Hongxiang Yu[1,2], Michael Lehning[3,4], Li Guang[1], Benjamin Walter[4], Jianping Huang[1], and
Ning Huang[2]

[1]School of Atmospheric Science, Lanzhou University, Lanzhou, 730000, China
[2]School of Civil Engineering and Mechanics, Lanzhou University, Lanzhou, 730000, China
[3]College of Architecture Civil and Environmental Engineering, Ecole Polytechnique Federal de Lausanne,
Lausanne, 1015, Switzerland
[4]WSL Institute for Snow and Avalanche Research SLF, Davos, 7260, Switzerland

**Correspondence:** Ning Huang (huangn@lzu.edu.cn)

**Abstract.** Snow cornices are a common snow pattern in cold regions, and their fracture and collapse can easily
trigger avalanches. Despite numerous observations and experimental simulations on their formation process, the
microscopic mechanism of their formation remains unclear. In this paper, based on wind-tunnel experiments and
high-speed photography, experimental studies on the trajectory of particles surrounding the snow cornice were carried
out. Results indicated that the cornice is composed of small-sized snow particles. Saltation is the most dominant
moving pattern for particles adhering to cornice. Notably, particles at the edge exhibit lower impact velocities and
a wider distribution of impact angles compared to those on the surface. Further analysis of force balance equations
of particles at the edge explains the shape-forming mechanism of wedged-like snow cornice. This work enhances
the understanding of the micro-mechanism of snow cornice formation, offering theoretical insights for avalanche
prediction.

## 1 Introduction

Snow consists of ice crystals. Snow particles may adhere to the surface at particle-bed collision. Therefore, wind can
shape the snow cover and produce special patterns by redistributing snow over various areas, such as sastrugi, snow
dunes (Sommer et al., 2018), and snow cornices. Snow cornice is one of the naturally formed accumulation patterns
in cold mountain regions. The collapse of snow cornices may induce snow avalanches (Vogel et al., 2012).

Previous field observations have consistently shown that the optimal wind speed range for cornice formation lies
between one and two times the threshold wind speed (Eckerstorfer et al., 2013b; Vogel et al., 2012; Hancock et al.,
2020). Recently, wind-tunnel experiments have further shown that cornice growth is maximized when the wind speed
exceeds the threshold value by 40% (Yu et al., 2022). However, the micr-mechanism for particle adhesion to the
cornice edge has not been studied in detail, due to the difficulty in observing the formation process at the particle
scale.

Currently, there are several hypotheses on how snow particles adhere at the edge: Irregularly-shaped snow particles
interlock with each other by their dendrites (Seligman et al., 1936); Charged snow particles are attracted by the





strong electric field above the snow cornice surface (Latham and Montagne, 1970); Pressure melt and frictional
heat (Latham and Montagne, 1970) when particles contact other surfaces leads to a quasi-liquid layer facilitating
fast sintering. In fact, when a particle impacts on a cornice surface, both the sintering time ($10^1$ s) and the fast-
sintering time ($10^{-1}$ s) (Bahaloo et al., 2022) are much longer than the collision time ($10^{-5}$ s). Thus, on the
particle sintering/fast-sintering time scale, the drifting-snow aerodynamic process dominates the redistribution of
snow particles and the patterns of snow cornices. However, the mechanical mechanism behind the wedge-shaped
(Seligman et al., 1936) snow cornice has not yet been investigated.

Cornice growth is often accompanied by drifting snow (Eckerstorfer et al., 2013a), in which a snow particle saltation
layer exists. Drifting snow particles move in three modes, namely, creep, saltation, and suspension, with the first
two modes contributing most to the snow mass transport (Bagnold, 2012). When snow particles collide with the
surface, three processes may occur: 1) Rebound occurs when a portion of particle kinematic energy is lost, but may
rebound from the surface; 2) Deposition occurs when the particle loses all its kinematic energy upon impact with
the ground; 3) Eject occurs when the particle transfers the kinematic energy to other particles on the ground upon
impact with surface, resulting in the entrainment of other particles resting on the surface.

During particle-surface collision, the particles transfer momentum and energy from air to the surface. These
processes are characterized using rebound and splash functions based on theoretical models (Lämmel et al., 2017;
Comola and Lehning, 2017) or observations (Anderson and Haff, 1991). Particle impact velocity and impact angle
(Walter et al., 2023) are two important parameters determining rebounding, deposition, and splashing processes.
It is however not clear how particle impact velocity and impact angle may affect particle adhesion snow cornice
formation.

Here, we carry out a wind-tunnel experiment of cornice formation, focusing on particle trajectory and adhesion
process in snow cornice formation. Based on the experimental results, we investigate the micro physical mechanism
for cornice formation.

## 2   Instruments and Methods

The wind-tunnel experiments are carried out in a ring wind tunnel in the cold lab of the WSL Institute for Snow and
Avalanche Research, in Davos, Switzerland. The experiment setup is shown in Fig. 1. For tracing particle trajectories,
a high-speed camera system is deployed. A feeding system is used to supply snow particles. The feeding rate of snow
particles is manually kept stable. The snow particles are produced by a snow maker (Schleef et al., 2014). The
geometrical diameters are 300 - 500 $\mu$m, and the shape of the snow crystals is dendritic, visually analyzed under a
microscope. The room temperature of the cold lab is controlled and set at -5 °C, and the wind speed is kept at 4 m
s$^{-1}$. A ridge model with a fixed size (Fig. 1) is built with compacted snow before each experiment.

The high-speed camera system consists of a high-speed camera (Phantom VEO710), an LED lamp as a light source,
and a transparent plane positioned at the opposite side of the camera to diffuse the light source and achieve uniform





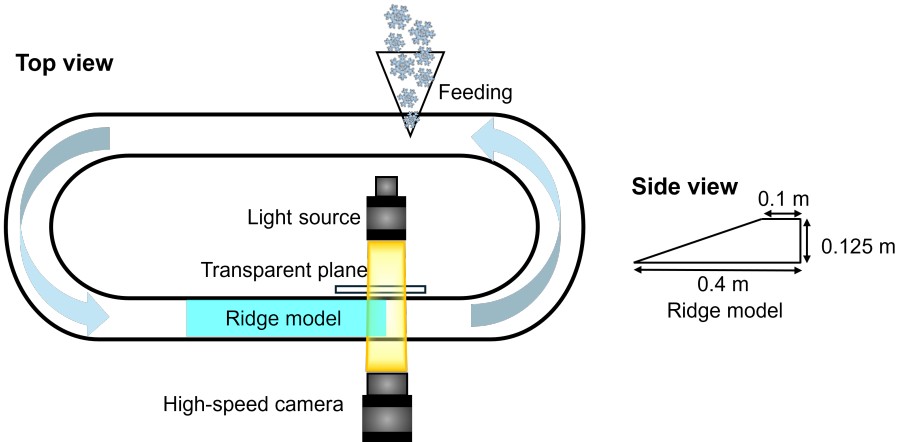

**Figure 1.** Schematic for experimental setups.

illumination. The system is employed for shadowgraphy analysis. The shadowgraphy is a technique used for extracting the moving particle's size, velocity, and trajectory. It is similar to PTV (particle tracking velocimetry)(Baek and Lee, 1996; Tagliavini et al., 2022), which has the same algorithms for particle information, and it has been frequently used in recent snow-related studies, for its advantages in robustness, non-intrusiveness, and accessibility. Shadowgraphy is particularly suitable for tracking snow particles, which are partially transparent and irregular in shape. It has been successfully applied to measuring particle mass flux, velocity, and size distribution (Paterna et al., 2016; Walter et al., 2023). The sampling frequency of the camera is set as 3 kHz, corresponding to a time interval of 333.32 µs. The images are captured for 4-5 s during cornice growth. In each case of the experiment, 12455 images are captured to record the particle trajectories. After the images are obtained, particle-bed collision events are selected visually. Then, particle sizes and trajectories in these events are analysed by image processing as follows.

## 2.1  Particle Recognition

The gray value of the snow cornice base is much higher than that of airborne snow particles. Thus, the images are first transferred into binary format by the two threshold gray values to recognize both the snow cornice base and airborne snow particles. The cornice zone is detected according to the first threshold gray value. The airborne snow particles are then detected according to the second threshold gray value after extracting the cornice from the image.

Particle size as recognized by the first-time binary analysis using the second threshold gray value is normally smaller than its real size, as shown in Fig. 2(a)-(b). To compensate for the underestimated particle areas caused by binarization, a dilation process (Gonzalez and Woods, 2017) is applied after the binarization, which fills the small spaces around the snow particle and smooths the particle's boundary, as depicted in Fig. 2(c) and its zoomed-in counterpart in (d).





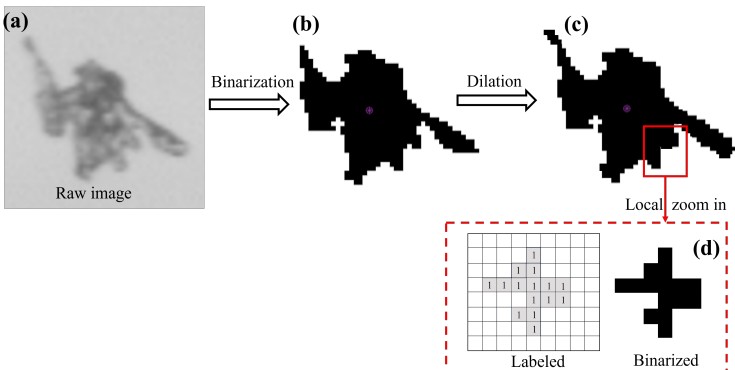

**Figure 2.** Image pre-processing method

To distinguish the noise points from air-flow snow particles, opening and closing processes (Solomon and Breckon, 2011) are then operated. The opening process eliminates very fine objects (noise points) and smooths the snow cornice boundary, and the closing process corrects the image such as filling the tiny empty holes inside of the snow

particles.

After the above image pre-processing, particle recognition is carried out by using the Seed-Filling Algorithm (SFA). SFA is an algorithm used to fill closed regions in an image. It starts from a seed point and gradually fills the regions adjacent to it with the same color until the boundary is reached. Snow particle area is calculated based on the connected component label analysis (CCLA), a common image processing method for connecting the adjacent

foreground pixels that have the same pixel value (Di Stefano and Bulgarelli, 1999; Rafael and Richard, 1993).

During the image processing, the particle's area is saved in binarized format in a numerical matrix. Thus, the particle's area $A_p$ can be estimated by calculating the sum of all the connected component labels. The particle's equivalent diameter is calculated based on the value of its projected area: $d_e = \sqrt{4A_p/\pi}$. All pre-processing of images (dilation, opening, closing operations) and particle recognition in this work are programmed by using the Matlab

software.

## 2.2 Particle Trajectory Tracking

By using the above particle recognition method, we obtained a series of images that contain the particle information. Then, the trajectory and velocity of snow particles are obtained using the trajectory recognition method that judges the relative position of the neighboring particles.

We paired each particle from the previous time series of images, by using the nearest-neighbor algorithms (NNA) (Crocker and Grier, 1996) to match the targeted particle's position in each frame image. NNA is a particle search method as shown in Fig. 3. In each time step, we search for the position of the target particle in the last frame of an image within the predefined search radius. The center of the corresponding circle area is assumed to be at the mass





center of the target particle in the last frame, and the radius is $d_t$=6 mm. We then match the target particle by its
shadow surface area similarity. By recognizing the position of the target particle in each frame image, we obtain the
velocity of the target particles.

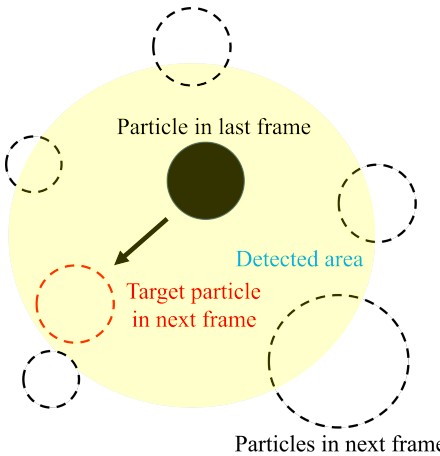

**Figure 3.** Schematic of particle detection method in post-processing high-speed camera images. The full circle is the particle
from the last frame, the dashed circle is the particles from the second frame, and the yellow region is the predefined search
area.

The $i_{th}$ target particle's horizontal and vertical velocity at a given time step $t$ and a time interval $\Delta t$ are calculated
as:

$$u_x(t) = \frac{x(t + \Delta t) - x(t)}{\Delta t} \tag{1}$$


$$u_y(t) = \frac{y(t + \Delta t) - y(t)}{\Delta t} \tag{2}$$

in which $x$ and $y$ are the coordinate position, $t$ is the current time, and $\Delta t$ is the time interval of a high-speed
camera. Therefore, the magnitude of particle velocity $u_i$ is:

$$u(t) = \sqrt{u_x(t)^2 + u_y(t)^2} \tag{3}$$

and the angle $\theta$ is:

$$\theta(t) = \arctan(\frac{u_y}{u_x}) \tag{4}$$





A particle contact event is defined when the particle's vertical velocity is from negative (downward) to zero or positive (upward) among several adjacent images. The impact velocity is then defined as the velocity before the contact, and the rebound velocity is defined as the velocity after the contact. We found the maximum error between visual observation (manually tracking the particle's trajectory) and program recognition of the edge particle velocity to be about 5%, the angle to be about 18%, and the diameter to be about 16%.

## 3 Results and Discussions

By maintaining the wind speed in the experiment at constant (4 m/s), 190 collision particles are analyzed with the image post-processing method, in which, 65 particles adhere on the cornice edge and 86 particles adhere on the cornice upper surface. The cornice edge in this paper refers to the spatial range of the 1 mm vertical front end of a dynamically growing cornice, and the cornice surface refers to the cornice's topside with the total length at the current time step minus the edge length, as is shown in Fig. 4(a).

The distributions of the snow particle impact velocity, impact angle, and size distribution are shown in Fig. 4(b). The blue points represent the particles that adhere on the edge and the red points represent those that adhere on the surface. The size of all points represents the particle's diameter. It can be concluded that the impact angle decreases with the increasing impact velocity. Particles that adhere on the edge have relatively lower and more narrowly-distributed values of impact velocity, compared to the particles that adhere on the surface. To investigate the differences between edge and surface particles, we analyzed the adhere particle's size distribution, impact velocity, and impact angle in the following sections.

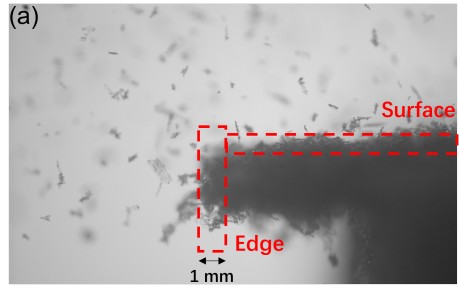

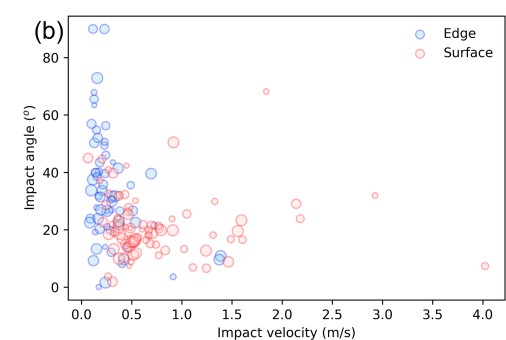

**Figure 4.** (a) Cornice edge and surface. (b) Distribution map of adhesion particles on the surface and edge.

### 3.1 Particle size distribution

The size distribution of particles adhering at different positions on a dynamically evolving cornice is analyzed, as is shown in Fig. 5. For all particles adhering to the cornice surface, their size distribution follows the log-normal





distribution function $\theta \sim N(\mu=277.17, \theta=0.45)$. For particles adhering at the edge, their size distribution follows the log-normal distribution function $\theta \sim N(\mu=264.42, \theta=0.38)$. And for particles adhering on the surface, their size

135 distribution follows the log-normal distribution function $\theta \sim N(\mu=342.69, \theta=0.28)$. It can be concluded that particles with smaller sizes adhere more likely on the edge, and larger particles adhere more likely on the cornice surface. The Stokes number (normally defined as $St = \frac{\rho_p d^2 U}{18\mu L}$) is a ratio of inertial effects to diffusive effects of a particle. For particles adhering on the edge, the Stokes number is $\approx 1.8$, thus those edge particles are dominant by the inertial force. The Stokes number of the surface particle is $\approx 3.1$, which is about 1.7 times greater than that of the edge particle, indicating that the inertia effect is more significant on the surface particle than on the edge particles.

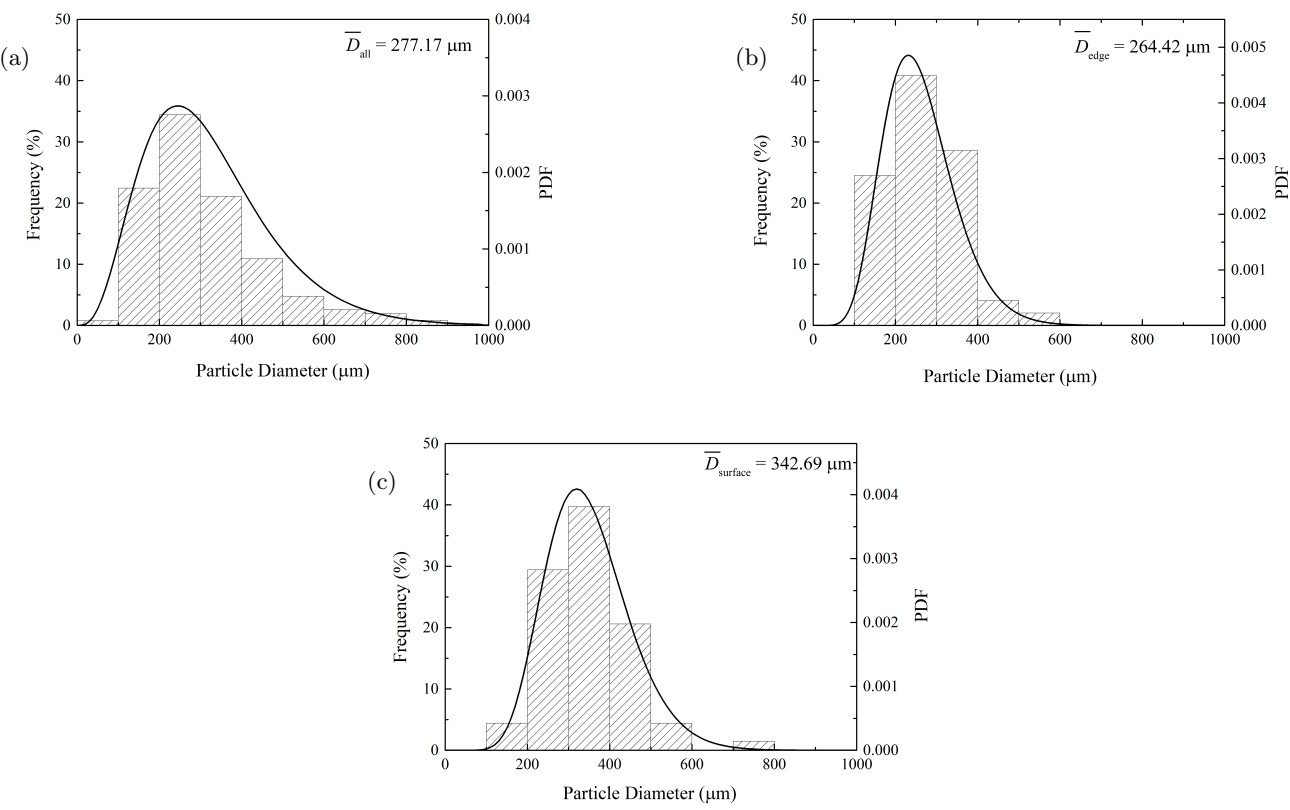

**Figure 5.** Size distribution of particles at different positions. (a) Airborne particles. (b) Adhesion particles at the edge. (c) Adhesion particles on the surface. The gray shadow represents the existence frequency of particles with different sizes.

140

In the experiment, the suspended particles in the higher region of the observation window fly over the cornice with a relatively high horizontal velocity and have little chance to deposit on the cornice, and almost all adhering particles reach the cornice tip first by saltating or creeping at the surface and then being captured by the edge. Thus, we mainly captured the particles moving very close to the surface by using a high-speed camera. It is observed that





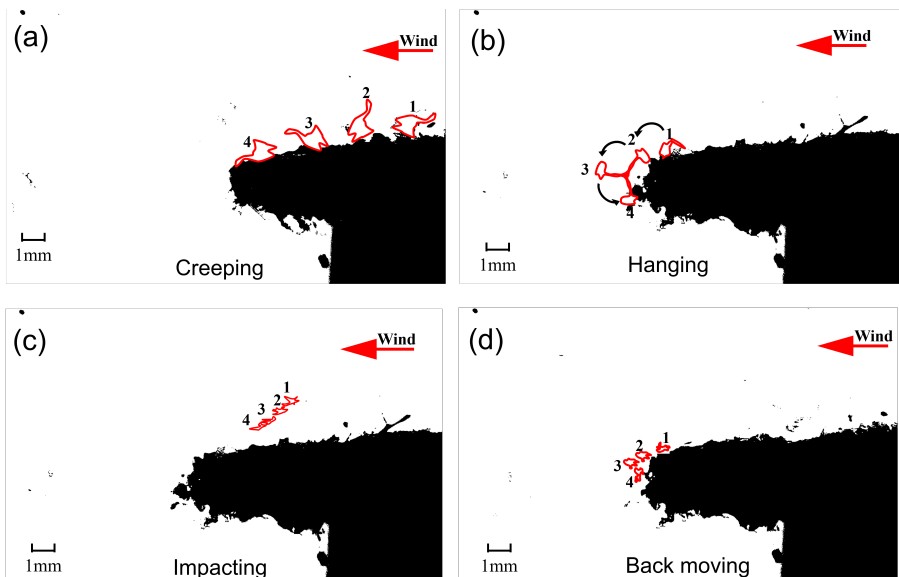

**Figure 6.** Four adhering patterns of snow particles (1,2,3,4 represents for different time steps).(a) Creeping particles. (b) Hanging particles. (c) Impacting particles. (d) Back moving particles.

creeping (particle rolling or sliding over the surface) and saltating (particle successively jumping over the surface) (Bagnold, 2012) are the two main types of particles contributing to the growth of the cornice. Creeping particles are entrained from the surface under the ejection of other particles. These creeping particles ($\sim 13.6\%$) normally have larger sizes and lower velocities. After undergoing multiple rotations in the downstream direction, some of the creeping particles come to a stop near the front end of the cornice, as is shown in 6(a), and some of them continue rolling to the tip and roll over the tip. Only a few creeping particles with elongated dendrites were found to physically interlock with adjacent particles at the front end of the cornice. Under the action of gravity, the hanging particles may start rotation with the interlocking point as the force center, as shown in Fig. 6(b), process 2-4.

Whether saltating particles rebound or deposit on the cornice surface after impaction depends strongly on their impact position, velocity, and angle. Most of the saltating particles deposit ($\sim 82.3\%$) before the cornice's front end, as is shown in Fig. 6(c). Only a few of particles deposit on the front end, which are smaller-sized particles ejected near the tip. Fewer saltating particles ($\sim 3.4\%$) going off the tip will later move backward to the cornice tip, under the action of the reflux vortex or the potential electric field, as shown in Fig. 6(d).

Here, we define the impact velocity/angle of the particle that deposits on the cornice surface as particle adherence velocity/angle (PAV/PAA). We first analyze the PAV and PAA of the 65 particles deposited on the edge and 86 particles deposited on the surface. In natural conditions, the snow cornice will slowly bend and deform under the gravity force. In our experiment, the snow cornice can be considered as horizontally growing in the whole process, and the bending effect of the snow cornice can be neglected for the short observation time. The impact angle is




defined as the angle of particle incidence on the horizontal cornice surface. We subdivided the values of particle
impact velocity and impact angle into different bins and analyzed the relative frequency of these values. The average
165  value of each bin is plotted as dots and the standard deviation is plotted as error bars in Fig. 7.

The relative frequency of PAV/PAA represents the probability of particle adhesion on the cornice with a cer-
tain impact velocity or impact angle. It is shown in Fig. 7(a) that for the two types of particles, the adhesion
probability exponentially decreases with the increasing value of PAV. The upper limit value is 4 m/s for the
surface particles and 1.5 m/s for edge particles. Specifically, the relative frequency of the edge PAV follows the
170  exponential function $f(v_{im}) = -0.03 + 0.64e^{-2.49v_{im}}$ $(R^2 = 0.89)$ and the relative frequency of the surface PAV fol-
lows the exponential function of $f(v_{im}) = -0.08 + 0.72e^{-1.32v_{im}}(R^2 = 0.98)$. Thus, the rebound probability of parti-
cles on surface is $f_{re}(v_{im}) = 1 - f_{st}(v_{im}) = 1.08 + 0.72e^{-1.32v_{im}}$, and the rebound probability of particles on edge is
$f_{re}(v_{im}) = 1 - f_{st}(v_{im}) = 1.03 + 0.64e^{-2.49v_{im}}$. The rebound probability functions are both exponential functions of
the impact velocity, which is consistent with the previous research (Anderson and Haff, 1991).

175  The frequency of PAA first increases and then decreases with the increasing impact angle $\theta_{im}$. Among these
151 particles that adhere on the cornice surface and edge, there exists a threshold PAA at which the particles are
most likely to adhere on the cornice, as is shown in Fig. 7(b). PAA on edge has a lower relative frequency and
is more widely distributed than that on the surface. PAA on edge follows the log-normal function $f(\theta_{im}) = 3.22 +$
$677.25/\sqrt{(2\pi)}0.39xe^{(-(ln(x/29.53))^2/(2*0.39^2))}(R^2 = 0.97)$, with the threshold value of 26°. PAA on surface follows
log-normal distribution $f(\theta_{im}) = -0.007 + \frac{7.17}{v_{im}}e^{-1.34(ln(\frac{v_{im}}{18.95})^2)}(R^2 = 0.996)$, with the threshold value of 15°.

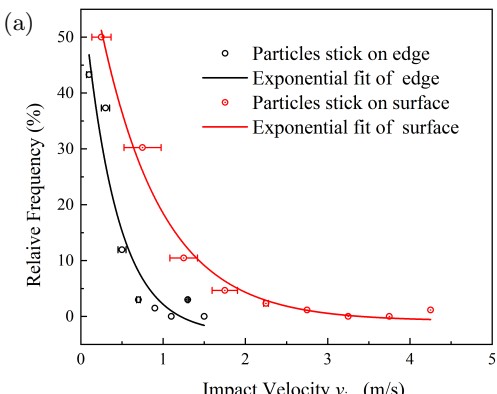
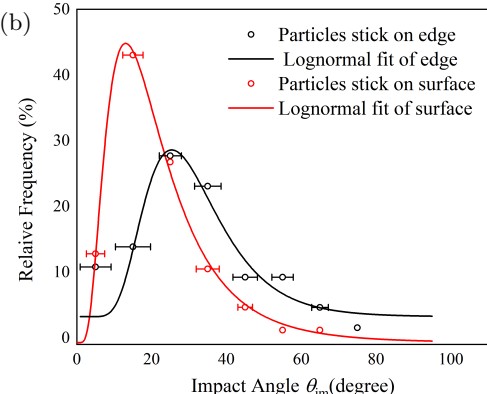

**Figure 7.** Relative frequencies of (a) PAV and (b) PAA on edge and surface.



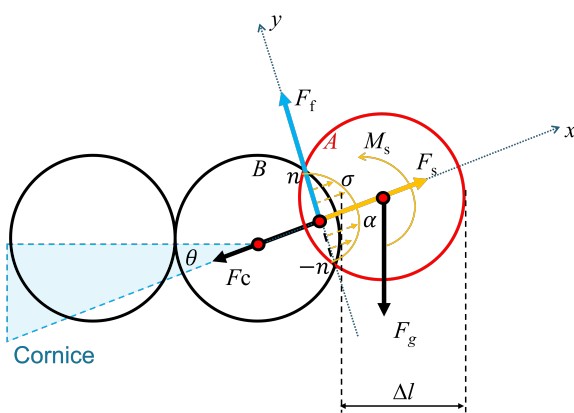

**Figure 8.** Schematic diagram of force analysis of particles adhering on the edge.

## 3.2 Static force analysis of adhering particles on the cornice edge

The cornice growth process can be divided into two stages (Yu et al., 2022). The first stage is the formation of a 1-2 particle-diameter thick snow slab, which is mainly determined by the spatial variation in the mass transport rate along the flow direction. The second stage is a repeated process of length growth and then thickness growth.

From the above work, we statistically characterized the kinematic properties of the moving particles around the snow cornice. However, the mechanics of particles stopping and adhering at the cornice edge have not been examined in detail. We now discuss the process of particle deposition at the edge by analyzing the balance of the static forces acting on the particles.

Considering one particle (particle B) at the edge of the cornice, and one newly deposited particle (particle A), as depicted in Fig. 8. The drag force and the lift force acting on particle A can be neglected due to the near-surface velocity being close to zero. The overlap area represents the compression deformation surface of the two particles.

By considering the center point on the surface as the analyzed balance point, the force and torque balance of the particle A can be expressed as:

$$F_g \cos\alpha + F_c = F_s \tag{5}$$

$$F_g \sin\alpha \leq F_f \tag{6}$$

$$F_g \sin\alpha \cdot (\frac{D}{2} - \delta_{max}) = M_s = \int_{-n}^{n} \sigma y \, dy \tag{7}$$





$$F_f = \mu_f F_s \tag{8}$$

in which, $F_g$ is the gravity force on particle A, $F_c$ is the cohesion force between two particles, $\alpha$ is the angle between the direction of gravity and cohesion, $D$ is the diameter of particle A, $\delta_{max}$ is the maximum compression displacement of the particle that can be calculated with the model proposed by Zhou (2011): $\delta_{max} = (1.25\pi R_i^{2.5} R_j^{2.5} \rho^2 \frac{(1-\mu_i^2)E_j + (1-\mu_j^2)E_i}{E_i E_j \sqrt{R_i + R_j}(R_i^2 - R_i R_j + R_j^2)})^{0.4}$, with $R_{i/j}$ is the radius of two contact particles, $\rho$ is particle density,

$E_{i/j}$ and $\mu_{i/j}$ are the Yong's modulus and the Poisson's ratio of particles, $\mu_f$ is the friction coefficient of the ice surface, and $\Delta l$ is the horizontal growth length which is lower than the diameter of particle A.

$F_s$ is the supporting force caused by the deformation on the contact surface, which is an integral elastic/plastic force along the normal direction of the contact surface: $F_s = \int_{-n}^{n} \sigma dy$, where $n$ is the radius of the contact surface, with $n = \sqrt{(\frac{D}{2})^2 - (\frac{D}{2} - \delta)^2}$. $\sigma$ is the elastic/plastic force per unit length of the contact surface, and its value is

defined by the compression displacement $\delta$ and the displacement rate $\dot{\delta}$ (Zhou, 2011). Due to the uneven compression displacement on the contact surface, there exists a counterclockwise torque of elastic/plastic force $M_s = \int_{-n}^{n} \sigma y dy$, balanced with the clockwise torque of gravity force. $F_f$ is the maximum static frictional force to counteract the tangential component of gravity force.

Substituting Eq. 5 and Eq. 8 into Eq. 6, the particle can stick together if:

$$\frac{F_c}{F_g} \geq \frac{\sin \alpha}{\mu_f} - \cos \alpha \tag{9}$$

$F_c$ is the cohesion force between particle A and particle B due to the sintering and electricity effects of snow particles, $F_c = F_e + F_b$. $F_e = q_i \sum_{j=1}^{N} k q_i q_j / r^2$ is the electricity force, which is the sum of electric force from air and surface distributed electric particles to the target particle, and $q_i$ is the charge amount of the target particle, $E = \sum_{j=1}^{N} k q_i q_j / r^2$ is the electric field formed by other particles. From the field measurement results, the electric

field around a snow cornice is the strongest at the cornice edge (350-480 V/m) and radially decreases (Latham and Montagne, 1970). For a snow particle with an average charge value of $10^2$ pC/kg (Latham and Montagne, 1970), the electric force subjected on the adhering particle can be estimated as $F_E \approx 10^{-8}$ N. The bond cohesion force is $F_b = Ae^{0.14T}$, $A = 3.18 \times 10^{-7}$ is a constant (Kuroiwa, 2007; Schmidt, 1980), and $T$ is air temperature. The bond force $F_b$ increases with the rising air temperature.

The angle of the cornice $\theta$ is the complementary angle of $\alpha$, and it is therefore affected by particle size $D$ and air temperature $T$. As the air temperature $T$ increases or the particle size $D$ decreases, the ratio of $\frac{F_c}{F_g}$ rises, and the cohesion force $F_c$ dominates over the gravity force $F_g$. Under this condition, the cornice is stable even with a relatively small angle $\theta$. On the other hand, the gravity force $F_g$ dominates over the cohesion force $F_c$ when the air temperature is low or particle size is large, and the shape of the cornice can only be stable with a large angle $\theta$. This

result is consistent with the previous experiment (Yu et al., 2022).



## 4    Conclusions

In this work, the micro-mechanism of snow cornice formation is investigated in a wind-tunnel experiment. Using a high-speed camera system, we recorded and analyzed particle trajectories surrounding the cornice, facilitated by a novel snow particle recognition program.

We found that the composition of the cornice predominantly consists of lightweight snow particles. We observe that small-sized particles exhibit a higher tendency to deposit on the cornice edge compared to the cornice surface. Near-surface saltation is the dominant mode for particles to adhere to the cornice. Most saltation particles deposit before the cornice's front end, which increases the cornice's thickness. Only a few saltating particles deposit on the front end, and a few creeping particles interlock and hang with other particles at the cornice tip, or fewer particles

go off the tip and move backward under certain forces that are in opposite directions with the flow.

Our investigation highlighted the indispensable role of snow particle-bed interactions in cornice growth, wherein the impact velocity and angle are pivotal determinants for particle adhesion along the cornice edge. Particle adherence velocity and angle distributions were analyzed. It is concluded that, both for particles deposited on the edge and the surface, the frequency of PAV decreases exponentially with the increasing impact velocity. The impact velocity

of edge particles is overall lower than that of surface particles. The impact angle of edge particles is more widely distributed than that of surface particles.

We then analyzed the static force balance for particles adhering to the edge of the cornice. The force balance suggests that the cornice angle is determined by the ratio of cohesion force and gravity force, which is influenced by air temperature and particle size.

In this work, the topography of the simulated mountain ridge featured an ascending slope leading to a flat expanse at its summit. This increasing slope induced flow separation at the edge, resulting in diminished wind velocity. Such conditions potentially elevate the probability of particle deposition. Future works should quantitatively study the influence of ridge morphology on cornice formation through experimental wind tunnel or numerical simulation.

*Author contributions.* YHX designed the experiments. YHX and LG carried out the experiments. YHX performed the data
analysis, and prepared the first draft. ML, BW, HJP and HN reviewed and edited the paper. HN and ML organized this study, contributed to its conceptualization, discussion, and finalized the paper.

*Competing interests.* The authors declare that they have no conflict of interest.

*Data availability.* The data archiving is underway and will be added after it is published. Data is available on request now.



*Acknowledgements.* The authors would like to thank the WSL Institute Snow and Avalanche Research SLF for making the
experimental facilities available for our study. We thank Alec van Herwijnen for his invaluable experiment support. This
work was supported by the National Natural Science Foundation of China (grant no.: U22A20564), the Third Comprehensive
Scientific Expedition and Research Program in Xinjiang (grant no.: 2022xjkk0101), the Second Tibetan Plateau Scientific
Expedition and Research Program (grant no.: 2019QZKK020109-2), and the Swiss National Science Foundation (grant 200020-
179130). The data and code will be uploaded to the Dryad repository after the paper is published.



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
