# Peer review of "Snow Particle Motion in Process of Cornice Formation"

_EGUsphere, 2024_

## Author Comment (AC1)

**Author's Response**

This is a very timely study demonstrating the microscale snow particle motion associated with an understudied snow bedform "Cornice", which is believed closely related to snow avalanches. The methods used in this paper is sound, the experiment settings were carefully tuned, which leads trustworthy experiment results. However, I think there are some minor issues in this manuscript and hopefully the authors can address them during the revision.

Response: Thanks for your time and positive feedback. We will revise it according to your suggestions.

1.  The title mentioned the cornice formation, however, the discussion is also related to cornice growth, suggesting the title can change to Snow particle motion during cornice development

Response: Thanks for your comments. We prefer to use "growth" instead of "development" because this word has a bit more of a dynamic and specifically emphasizes the ongoing increase in size and volume of the cornice.

2.  It is great to identify the four major adhering patterns, but it will be nice to link the force analysis with these four patterns

Response: Thanks for your suggestions. The four major adhering patterns are classified based on the dynamic analysis of particle movement. However, the static force analysis is carried out after the particle dynamic movement, which is the same for all the deposited particles. The dynamic force analysis would be the focus of future work.

3.  It will be nice to discuss the limitations of the study, e.g. wind speed, humidity and temperature impact on cornice development. The implications for this study should be also discussed, e.g. how this research helps the understanding of the mechanisms of snow avalanches.

Response: Thanks for your suggestions. The effects of wind speed on cornice growth have been extensively studied in recent work by Yu et al. (2023). The effects of humidity and air temperature on cornice growth have been tested and will be published in our next paper. In this work, we controlled the air temperature as a constant to isolate the environmental variables. In the static force analysis (Section 3.2), we indicated that the air temperature has an impact on the cornice angle by changing the bond cohesion force between snow particles.

This work helps deepen our understanding of cornice growth from a micro-view, it not only provides a new perspective for setting up the parameterizations for splash functions of snow particles but also provides a theoretical basis for avalanche predictions. We will add the above discussion to the next version of the manuscript after revision.

**References:**

Yu, H., Li, G., Walter, B., Lehning, M., Zhang, J., and Huang, N.: Wind conditions for snow cornice formation in a wind tunnel, The Cryosphere, 17, 639–651,

https://doi.org/10.5194/tc-17-639-2023, 2023.

---

## Author Comment (AC2)

**Reply to RC1**

**General comments**

This paper presents a detailed observation of snow particle motion in order to understand the process of snow cornice formation. It also investigates the conditions under which snow particles adhere to snow cornices through particle-level force analysis. There are few cases where snow cornice formation has been observed, so even though this is a very small-scale experiment in a wind tunnel rather than a full-scale snow cornice, this study is very informative. In addition, it is expected that the detailed force analysis will lead to the construction of a model for the snow cornice formation process, making this work worthy of publication.

**Reply:** We appreciate the reviewer's recognition of our work, and we will delve into the impact of dendritic snowflakes on force analysis in the next version of the manuscript.

**Specific comments**

There is a big question about the force analysis, which is the main topic of this paper. The wind tunnel experiment in this paper uses dendritic snow particles. Although it is not clearly stated in the paper, my personal experimental experience and personal communications with researchers suggest that snow cornices can only form when dendritic snow particles are used, whereas they do not grow when spherical particles are used. The reason for this is not entirely clear, but it is thought that the large contact surface of dendritic particles makes it easier for snow particles to adhere to each other than for spherical particles. **However, this paper discusses the balance of forces assuming that the particles are spherical, so it is possible that the contribution of the contact area of dendritic snow particles is sought in other forces when considering adhesion.** To make this paper fruitful, I recommend that the author re-examine whether there are differences in **snow cornice formation** and **adhesion forces** between spherical and dendritic particles. Of course, it may not be easy to discuss adhesion forces between dendritic particles, but I expect that the contribution of dendritic shapes can be estimated from the parts that cannot be explained by considering spherical particles.

We are grateful for the valuable feedback provided by the reviewer. In the next iteration of the manuscript, we will explore the impact of dendritic snowflake structures on our force analysis in more detail.

Reply: Thank you for the valuable suggestions. We do have tested the fresh snow particles and aged snow particles (by keeping the fresh snow for a few days, the particle shape becomes near-spherical), and found that: 1) both of them can form a snow cornice; 2) fresh snow particles are much easier to form a snow cornice than the aged ones.

The adhesion forces subjected on spherical particles are greater than that on dendritic particles, i.e., $F_c$(dendritic) $=$ $F_c$(spherical) $\times$ $C$, where the coefficient $C \approx$ 1.44 is derived from the previous experiment results (Eidevåg et al., 2022). The ratio of cohesion force to gravity $\frac{F_c}{F_g}$ determines the value of $\alpha$, which is related to the cornice angle. Dendritic particles therefore have greater values of $\alpha$, indicating that they have a wider range of balanced positions at the edge, and therefore they are more prone to adhere to the edge.

Moreover, it can also be inferred that smaller-sized particles or particles with lower-impact velocity particles tend to have greater values of $\alpha$. This is because smaller particles are subjected to lower gravity, while particles with lower impact velocity have greater cohesion force (Eidevåg et al., 2022). The higher $\frac{F_c}{F_g}$ ratios for both smaller-sized and lower impact velocity particles lead to greater values of $\alpha$, making them more prone to adhesion. This explains the phenomenon observed in the experiment – that snow particles with smaller size and lower impact velocity are more prone to adhesion on the edge.

The above discussion will be added to the new version of the manuscript.

**Technical corrections**

Line 19: micr-mechanism -> micro-mechanism?

**Reply:** Thanks for pointing it out. We have revised it in the new version of the manuscript.

**References:**
Eidevåg, T., Thomson, E.S., Kallin, D., Casselgren, J., Rasmuson, A. Angle of repose of snow: An experimental study on cohesive properties. *Cold Reg. Sci. Technol*. 2022, 194, 103470. doi: 10.1016/j.coldregions.

---

## Author Comment (AC3)

**Reply to CC2**

**General comments**

The study effectively addresses a critical gap in understanding the micro-mechanics of snow cornice formation and its role in avalanche initiation. Due to the limited research on cornices, this work stands out as a highlight. Using wind tunnel experiments and high-speed photography, it achieves precise and reproducible observations. Systematic analysis supports its conclusions with robust statistical and theoretical methods.

**Reply:** We appreciate the reviewer's positive feedback on our work.

**Specific suggestions**

1) While the wind tunnel experiments provide controlled conditions, they do not fully replicate natural environments with variable wind speeds, temperatures, and snow particle compositions. Including a brief discussion on these limitations and how they affect the results would enhance the study.

**Reply:** Thanks for your suggestions. In the wind tunnel experiment, we have tested various environmental factors such as wind speeds, air temperature, and snow particle type effect on the growth process of snow cornices. We found that wind speed is the primary condition for cornice growth (Yu et al., 2023). Air temperature, snow particle type, and other factors such as topography are secondary conditions. Air temperature affects the cohesion rate of snow particles and the related work is still under analysis. In this work, we aim to investigate the micro-mechanism of snow cornice formation, therefore, we analyze the results in a constant and steady environment condition.

We have tested the fresh snow particles and aged snow particles (by keeping the fresh snow for a few days, the particle shape becomes near-spherical), and found that: 1) both of them can form a snow cornice; 2) fresh snow particles are much easier to form a snow cornice than the aged ones. To distinguish the differences between spherical particles and dendritic particles, we change the cohesion force for dendritic particles in the mechanical model. Based on the previous experiment results (Eidevåg et al., 2022), we derived the magnitude of adhesion force for non-spherical particles is about 1.44 times that of spherical particles.

2) Discussing on how these findings could refine or enhance existing related or similar models would make the study more impactful.

**Reply:** Our experimental results analyzed the critical condition for particle stick on an edge or surface and concluded the distribution functions of impact velocity and impact angle. These results can be referred to splash functions of drifting snow dynamic models in the future, which could be helpful for researchers to simulate snow bed

features such as snow ripples, snow dunes, and snow cornices.

**References:**

Yu, H., Li, G., Walter, B., Lehning, M., Zhang, J., and Huang, N.: Wind conditions for snow cornice formation in a wind tunnel, The Cryosphere, 17, 639–651, https://doi.org/10.5194/tc-17-639-2023, 2023

---

## Author Comment (AC4)

**Reply to RC2**

**General comments**

I appreciate very much for the efforts to observe the particle motion carefully in the wind tunnel and investigate the growing mechanism of the thin snow plate which extends to leeward from the edge. However, according to my observations in the fields and the wind tunnel experiments, it is extremely fragile and always breaks down after **growing several centimeters long at the maximum.**

**Respond:** Thank you very much for your time and valuable comments. The cornice growth on the mountain ridge and causes snow avalanches has been observed by Eckerstorfer et al. (2013b), Vogel et al. (2012), and Hancock et al. (2020). From the previous observation results, cornice growth experiences multiple times of collapses and extends. At each time of growth, it may break down after a few centimeters of growth. This phenomenon is also observed in our experiment. Due to the limited field observation, it is tough to observe the particle moving around a real cornice on the mountain ridge. However, understanding the mechanism of snow cornice formation is still essential for avalanche prediction and simulation. Therefore, we carried out a wind tunnel experiment to observe the particle movement surrounding a small cornice we produced, for no matter in the field or in the wind tunnel, the particle moving and sticking laws are the same.

It never grows to the much larger one, such as, we find on the crest of a ridge along a mountain slope and occasionally causes the avalanche release.

**Reply:** Thanks for pointing this out. This is due to the involved dimensions and timescales which are way smaller/shorter in the ring wind tunnel. However, the size of crest of a ridge is a long-term result by multiple growth events, not in one precipitation event. In our experiments, the cornice can grow to more than 10 cm, which is almost the size of the base.

The authors need to make clear at the outset that the authors are looking at completely different phenomena. Even though it is allowed to say this miniature as "cornice" as well in a broad sense, the following numerous points should be taken into consideration before the publication.

**Reply:** Thanks for your suggestions. Although there are differences to real natural conditions, we are looking at snow particles that are transported by wind across a ridge while a fraction of the particles deposit on the lee edge resulting in horizontal and vertical snow slab (cornice) growth. As we mentioned above, no matter in the field or in the wind tunnel, the particle moving and sticking law are the same.

In all, we will revise the manuscript according to your suggestions. The following are the point-to-point responses:

**Specific comments**

Line 14: After "and snow cornice", Seligman et al. (1936) should be put as the reference.

**Reply:** We agree with this and will add it in the revised manuscript.

Line 29: "mechanical mechanism" sounds redundant and unnatural, although grammatically correct. I suppose only "mechanism" is fine.

**Reply:** Here, we emphasize the mechanism of the mechanics by carrying out the static force analysis on the single particle which deposits on the edge of a cornice. Therefore, we used "mechanical mechanism".

Line 53 to 54: 4 m/s is the wind speed at the center of wind tunnel? If the authors would like to analyze the experimental output physically, the friction velocity $u_*$ should be used in the manuscript instead. Furthermore, the threshold wind speed needs to be specified.

**Reply:** Thanks for your suggestions. The thickness of the boundary layer in this ring wind tunnel is very thin (~2 cm), and the wind speed inside the wind tunnel can be considered as uniformly distributed on the cross-section. Therefore, we used the center wind speed instead of friction velocity in the manuscript. We will mention this in Section 2 of the revised manuscript: "The wind speed inside the wind tunnel is nearly uniform with a very thin (~2 cm) boundary layer.".

Figure 1: 0.125 m on the side view corresponds to the height of snow on the floor?

**Reply:** Thank you for pointing this out. This value represents the height of the ridge model made of snow. The height was measured from the lower floor. We added description in line 56: "A ridge model with a fixed size (height 0.125 m, total length 0.4 m, flat surface length 0.1 m) is built with compacted snow before each experiment, and its side view is shown in Fig. 1."

Line 118: "190 collision particles": Are these particles with the snow surface? Such careful explanations are lacking overall. Please check the manuscript again by standing at the place of the readers.

**Reply:** Thank you for pointing this out. These 190 collision particles are moving particles in the air, and they also interact with the surface. This sentence will be revised to: "By maintaining the wind speed in the experiment at a constant (4 m/s), the study analyzed 190 collision particles as they interacted with the snow surface, including particles that rebounded, impacted, or deposited on the snow, using an image post-processing method."

Figure 4 (b): Although it is described that the size of points shows the particle diameter, no explanation are found in the figure caption. Further, the corresponding specific size should be added as well.

**Reply:** Thank you for pointing it out. We have re-plotted this figure and used color variation to represent the size difference. The explanation are added in the figure caption: "Impact velocity and impact angle of different sizes (deeper color represents for larger size) of snow particles on edge (in blue points) and surface (in red points)."

[Figure]

Fig. 4(b) Impact velocity and impact angle of different sizes (deeper color represents for larger size) of snow particles on edge (in blue points) and surface (in red points).

Line 137: Although Stokes number: $St$ is introduced, no explanations about the parameters are shown. Probably, ρp is particle density, $d$ is particle diameter, $U$ is wind speed, $m$ is kinematic viscosity and $L$ is the characteristic length scale. Specify each value you used, particularly $L$. It looks the difference of St between the surface and the edge is caused by the particle diameter only. If it is the case, I am wondering it is needed to take the trouble to introduce $St$. As you see, $St$ is a dimensionless number that characterizes the behavior of particles suspended in a fluid flow. It represents the ratio of the particle's inertial forces to the viscous forces exerted by the fluid. When $St$ is much smaller than one, the particle closely follows the fluid flow, whereas $St$ is much larger than one particle's inertia dominates, and it is less affected by the fluid flow, tending to maintain its original trajectory. On the other hand, at $St \approx$ 1 the particle behavior lies between these extremes, showing partial coupling with the fluid flow. In this case, both 1.8 and 3.1 are close to one and the difference is quite small; correspond to the third case together. Consequently, I have to say the discussions here are almost meaningless.

**Reply:** We agree with this and will add a notation for all variables existing in this manuscript. The discussion on the Stokes number will be deleted from the revised manuscript.

**NOTATION**

| Symbol | Definition and units |
|:------:|:--------------------:|
| $A$ | Constant [=$3.18 \times 10^{-7}$] |
| $A_p$ | Projected area of one particle [m²] |
| $D$ | Diameter of particle [m] |
| $d_e$ | Particle equivalent diameter [m] |

| | |
|---|---|
| dt | Time interval [s] |
| $E$ | Electric field [V/m] |
| $v_p$ | Particle velocity [m s$^{-1}$] |
| $v_{px}$ | Particle velocity component in x direction [m s$^{-1}$] |
| $v_{py}$ | Particle velocity component in y direction [m s$^{-1}$] |
| $v_{imv}$ | Vertical impact velocity of particle [m s$^{-1}$] |
| $E_{i/j}$ | Yong's modulus of particle $i/j$ |
| $F_b$ | Bond cohesion force [N] |
| $F_c$ | Cohesion force [N] |
| $F_e$ | Electricity force [N] |
| $F_f$ | Maximum static frictional force [N] |
| $F_g$ | Gravity force [N] |
| $F_s$ | Supporting force [N] |
| $q_i$ | Charge on particle [C] |
| $M_x$ | Torque of elastic/plastic force [Nm] |
| $n$ | Radius of particle contact surface [m] |
| $R_{i/j}$ | Contact radius of particle i/j [m] |
| $T$ | Air temperature [$^o$C] |
| $t$ | Current time [s] |
| $\Delta t$ | Time step [s] |
| $\Delta l$ | Horizontal growth length [m] |
| $\delta$ | Compression displacement [m] |
| $\delta_{max}$ | Maximum compression displacement on particle [m] |
| $\dot{\delta}$ | Displacement rate [m/s] |
| $\theta_p$ | Particle moving angle [$^o$] |
| $\theta_{im}$ | Particle impact angle [$^o$] |
| $\theta$ | Cornice angle [$^o$] |
| $\mu_{i/j}$ | Poisson's ratios of particle i/j |
| $\mu_f$ | Friction coefficient of ice surface |
| $\rho_p$ | Particle density [kg/m$^3$] |
| $\sigma$ | Elastic/plastic force per unit length [N] |
| $\alpha$ | Angle between direction of gravity and cohesion [$^o$] |

Figures 5 and 7: Generally, the number of particles you analyzed is extremely small. If the authors would like to induce concrete and statistically significant conclusions, at least more than 300 particles data for edge and surface each should be corrected and examined.

Figure 7: Authors are looking at the impact velocity and the angle separately. I suppose the combination of two factors for the particles on edge and surface reveals interesting findings,

supposing the enough data exists.

 **Reply:** Thanks for your suggestions. We have increased 383 surface particles and 121 edge particles. The p-values of both surface and edge particles are higher than 0.05 (not significant), which represents that the sample number satisfies the analysis results. The updated Figure 5 and Figure 7 are shown below.

In the updated Fig. 5, the size distribution of particles follows the same trend as the previous one, with the same distribution function but different parameters. We will revise lines 134 - 137 to: "For all particles adhering to the cornice surface, their size distribution follows the log-normal distribution function $\theta \sim N(\mu = 5.94, \theta = 0.36)$. For particles adhering at the edge, their size distribution follows the log-normal distribution $\theta \sim N(\mu = 5.76, \theta = 0.43)$. And for particles adhering on the surface, their size distribution follows the log-normal distribution function $\theta \sim N(\mu = 5.74, \theta = 0.47)$."

[Figure]

Figure 5. Size distribution of particles at different positions. (a) All particles. (b) Adhesion particles at the edge. (c) Adhesion particles on the surface. The grey shadow represents the existence frequency of particles with different sizes.

In the updated Fig. 7, the impact velocity of edge particles follows the exponential distribution function $y = 4.3 + 30e^{-0.9v_{imp}}$, with values mainly concentrated at below 1.5 m/s. While the impact velocity of surface particles follows the Gaussian distribution function $y = -2.8 + \frac{79}{3.4\sqrt{\pi/2}}e^{-2((v_{imp}-2.9)/3.4)^2}$, with values mainly concentrated at ~3 m/s. The impact angle of surface particles follows the Exponential distribution function $y = -0.8 + 80.6e^{-0.1v_{imp}}$, with values mainly concentrated below 17°. While the impact angle of edge particles follows the Gaussian distribution function $y = 7.1 + \frac{448.8}{45.7\sqrt{\pi/2}}e^{-2((v_{imp}-18.1)/45.7)^2}$, with values distributed more uniformly in range. The

average value of impact angle of surface particles is 13°, which is consistent with the previous experimental results of Nishimura (2000), as is shown in the red dash in Fig. 7(b).

[Figure]

Figure 7. Relative frequencies of (a) PAV, (b) PAA, and (c) vertical impact velocity of particles on edge and surface.

Furthermore, we combined the impact velocity and angle by analyzing the vertical impact velocity ($v_{imv}=v_{im}\sin\theta_{im}$) in Fig. 7 (c). Both surface particles and edge particles follow the exponential distribution, with surface particles $y = 0.5e^{-v_{imp}/0.7}$, and edge particles $y = 0.4e^{-v_{imp}/0.8}$. The relative frequency exponentially decreases with the increasing vertical impact velocity, this is due to the fact that particles with lower vertical impact velocity have relatively lower coefficient of restitution. The vertical impact velocity distributions of surface particles and edge particles are in same trend, although the impact velocity and impact angle distributions of edge particles and surface particles are different. It indicates that particle adhesion to the surface is mainly determined by the vertical velocity, and the differences in impact velocity and angle distributions between surface and edge is due to the topographic changes. It can be inferred from Fig. 7(c) that the threshold vertical impact velocity for surface particles (~3.25 m/s) is higher than that of the edge particles (~2 m/s). This is because cornice edge is a fragile structure, particles with relatively higher impact energy will induce the edge break off.

We will add the above discussion into the revised manuscript.

Further, here, authors set focus on saltating particles only. How do you estimate the contribution of the **creep particles**?

**Reply**: Thanks for pointing it out. The samples we analyzed also include the creep particles (rolling over the surface). We concluded four adhering patterns of snow particles, and one of them is creep. The creep particles contribute ~13.6% of the particles (using the number ratio) that stick on the cornice, which has been mentioned in lines 147-148.

Figure 8: Please make the figure clearer. Though many parameters are introduced, it is hard to recognize what they mean and indicate, such as, a. In addition, preferably, make the right and left sides reverse to align with Figures 4 and 6.

**Reply**: Thanks for your suggestions. To address the concerns raised, we will add a variable table to clearly define and explain the parameters as we mentioned above. Moreover, Fig. 8 has been adjusted by reversing the right and left sides to align with the structure of Fig. 4 and 6.

[Figure]

Fig. 8 Schematic diagram of force analysis of particles adhering on the edge

Line 190-191: In fact, the wind speed near the surface is getting smaller. However, you cannot neglect the wind shear stress acting there.

**Reply**: Thank you for this suggestion. There are two methods of calculating the drag force on a particle. First method calculates the drag force with the relative velocity between the particle and the flow:

$$F_d = 18\pi D^2 C_D (u - u_p)^2$$

where $u_p$ is the particle velocity, $u$ is the air wind speed, $C_D$ is the drag coefficient, and $D$ is the particle diameter.

The second method calculates the drag force with the surface shear stress [N/m²]:

$$F_d = \tau A$$

where $A$ [m²] is the wind-affected surface area of a particle.

The shear stress ($\tau = \rho u_*^2$) is small due to the sudden sharp decrease of the friction velocity at the edge (calculated by numerical CFD simulation, not published yet), as is shown in the Fig. S1.

[Figure]

Figure S1. Friction velocity along the snow model in different wind conditions.

The drag force of a snow particle with mean diameter calculated by the second method is about $10^{-10}$ N according to our simulation results, which is much smaller than the gravity $F_g$ ($10^{-8}$ N). Therefore, the results from these two calculation methods are almost same: The drag force on the particle that sticks on a cornice edge can therefore be ignored when carrying out static force analysis.

Line 191: According to your data, particle speed is quite low (nearly 0.5 m/s), thus, the compression deformation due to the collision is unlikely.

**Reply**: Based on the previous experimental studies (Wang et al., 2020), there are two kinds of deformation when snow particles contact collision happens: Plastic deformation and brittle failure, which is mainly determined on the loading rate (which refers to the relative moving speed of snow particles here). When the loading rate is relatively low, the snow particles will undergo continuous plastic deformation. During the collision process, the particles will be compressed and deformed, transforming from their original dendritic shape into irregular polygonal structures. When the loading rate is relatively high, it will lead particle sudden fracture and brittle failure. For particle speed with moving speed of 0.5 m/s, the particle will undergo continuous plastic deformation.

For clarity, we will add the sentences before introducing the maximum compression displacement of particle in Section 3.2: "During contact collision, snow particles will be compressed and deformed, which will undergo plastic deformation and brittle failure (Wang et al., 2020)".

Line 216: Authors say that the sintering time is much longer than the collision time in the manuscript at lines 26-27 and exclude the mechanism to create the wedged-shaped form. I do not understand why it appeared again. It looks contradictory statements. Furthermore, the static electric forces have been observed in the specific cases so far, and not always.

**Reply**: Thank you for pointing out the unclear point in the manuscript. I understand the concern about the seemingly contradictory statements regarding the role of sintering in the formation of snow cornices. I would like to clarify that in lines 26-27, it was stated that the sintering time is much longer than the collision time when a particle impacts the cornice surface. This means that on the timescale of particle sintering, the drifting snow aerodynamic process dominates the redistribution

of snow particles and the patterns of snow cornices. However, the intention was not to imply that the sintering force is unimportant in maintaining the shape of snow cornice. Rather, the point is that during the initial stage of cornice formation, the drifting snow aerodynamic process is the primary driver of the distribution of snow particles. After these particles are deposited on the edge, the sintering force then dominates the main place in keeping the particle stationary, thus shaping and preserving the cornice structure.

To be clear, sentences in lines 26-27 will be deleted, and add "In which, $F_b$ is the sintering force, which can be calculated as the product of the ice tensile strength and the area of the contact surface (Szabo and Schneebeli, 2007), $\sigma_{tensile} n^2 \pi$. Sintering happens after the particle deposit and dominates the main place in keeping the particle stationary, thus shaping and preserving the cornice structure." in Section 3.2 of the new version of manuscript.

Although authors tried to introduce all the conceivable forces which may act between particles on edge, time scales are not always consistent. Furthermore, all the discussions are qualitative from beginning to end and no quantitative estimates, which is enough to keep the thin plate growing, are shown. To say the least, quantitative approval of their idea, based on the snow and environmental data obtained in the experiment, is essential to make the manuscript reasonable and worthwhile.

**Reply**: As response above, during snow particle transportation, deposition and erosion on surface always happens when particle is in stationary state. Although sintering has longer time compared to particle collision process, the sintering effects starts from the beginning of particle deposit on the surface/edge of the cornice. Thus, the sintering force should also be considered, especially in this static mechanical analysis.

The size of snow particles follows a distribution function, and different sized particles are subjected to varying values of particle forces. Therefore, we have presented the order of magnitude instead of the exact values of each force. This allows us to analyze which forces are dominant and which forces can be neglected.

In the new version of manuscript, we will revise the discussion on mechanical force analysis. We also added the comparison of dendritic particle and round particle. We found that the ratio of cohesion force to gravity determines the shape of snow cornice. Due to the fact that the cohesion force for dendritic particle is 1.44 times of that value for spherical particle, indicating that the dendritic particle is more prone to stick on the edge. This theoretical analysis explained our experimental phenomenon. Moreover, it can also be used to explain the phenomenon in experiment that particles with smaller size and lower impact velocity are more prone to adhere on edge.

**Reference:**
Enliang Wang, Xiang Fu, Hongwei Han, Xingchao Liu, Yao Xiao, Yupeng Leng, Study on the mechanical properties of compacted snow under uniaxial compression and analysis of influencing factors, Cold Regions Science and Technology, Volume 182, 2021, 103215, ISSN 0165-232X, https://doi.org/10.1016/j.coldregions.2020.103215.

---

## Author Response (AR1)

**Responses to the editor and each reviewer's comments**

Title: Snow Particle Motion in Process of Cornice Formation

ID: EGUSPHERE2024-2458

Authors: Hongxiang Yu, Guang Li, Benjamin Walter, Jianping Huang, Ning

Huang, and Michael Lehning Submitted to: The Cryosphere

The comments are in blue. Page and line numbers refer to the revised manuscript version with changes marked in italic.

**Responses to Editor:**

Dear authors,

Based on your responses to referees' and community comments in the interactive discussion, you are invited to submit an appropriately revised version of your manuscript. Please make sure all issues raised in the reviews are effectively addressed in the revision. Note that the revised manuscript will be returned to the referees for further assessment before making a final editorial decision.

Best regards,

Guillaume Chambon / TC Topical Editor

Dear Editor,

We would like to thank you for the assessment of our manuscript. We have modified the manuscript accounting for all reviewer comments, which improved the quality of our work.

Based on the recommendations of reviewer 1:

- We have added the comparison of cohesive force of dendritic particles and spherical particles.

Based on the recommendations of reviewer 2:

- We have increased the particle number so that the analysis satisfies the requirements.
- We have analysis the vertical impact velocity of particles that adhere on edge and surface.
- We have reconsidered the static force analysis.
- We have clarified all the variables to increase the readability of this manuscript.
- We have highlighted the quantitative results in the manuscript.

Based on the above mentioned modifications, the manuscript has been improved. The discussion on the mechanism of particle adhesion on cornice edge and surface has been

extended. Also, the revised mechanism model can be used to explain the experimental results, which increases the coherence of manuscript.

We have clarified all the variables to improve the readability of this manuscript. We hope these modifications adequately address all the comments by the reviewer. Please find a point-by-point reply to each of the comments below.

Sincerely, Hongxiang Yu, on behalf of all authors

**Responses to Reviewer #1:**

**General comments**

This paper presents a detailed observation of snow particle motion in order to understand the process of snow cornice formation. It also investigates the conditions under which snow particles adhere to snow cornices through particle-level force analysis. There are few cases where snow cornice formation has been observed, so even though this is a very small-scale experiment in a wind tunnel rather than a full-scale snow cornice, this study is very informative. In addition, it is expected that the detailed force analysis will lead to the construction of a model for the snow cornice formation process, making this work worthy of publication.

**Response:** We sincerely thank the reviewer for the positive evaluation of our work and for recognizing the significance of our study in understanding the process of snow cornice formation. We appreciate the acknowledgment of the value of our wind tunnel experiments and the potential of our force analysis to contribute to modeling the snow cornice formation process.

**Specific comments**

**Comment 1:**

There is a big question about the force analysis, which is the main topic of this paper. The wind tunnel experiment in this paper uses dendritic snow particles. Although it is not clearly stated in the paper, my personal experimental experience and personal communications with researchers suggest that snow cornices can only form when dendritic snow particles are used, whereas they do not grow when spherical particles are used. The reason for this is not entirely clear, but it is thought that the large contact surface of dendritic particles makes it easier for snow particles to adhere to each other than for spherical particles. However, this paper discusses the balance of forces assuming that the particles are spherical, so it is possible that the contribution of the contact area of dendritic snow particles is sought in other forces when considering adhesion. To make this paper fruitful, I recommend that the author re-examine whether there are differences in snow cornice formation and cohesive forces between spherical and dendritic particles. Of course, it may not be easy to discuss cohesive forces between dendritic particles, but I expect that the contribution of dendritic shapes can be estimated from the parts that cannot be explained by considering spherical particles.

**Response:** Thank you for your valuable comment. We have tested the fresh snow particles and aged snow particles (by keeping the fresh snow for a few days, the particle shape becomes near spherical) and found that: 1) both of them can form a snow cornice; 2) fresh snow particles are much easier to form a snow cornice than the aged ones.

In the revised manuscript, we have added the description on snow particle type we use

**in section 2 Instruments and Methods.**

Lines 67 to 73:

Before conducting the experiment, we have performed preliminary tests on both fresh snow particles and aged snow particles. Fresh snow particles, characterized by their highly dendritic shapes, were compared to decomposed snow particles, which are characterized by small rounded shapes after being stored for several days at a constant temperature of  $T_{air} = -10$ °C. The results show that both types of snow particles are capable of forming a snow cornice. However, fresh snow particles exhibit a significantly higher propensity for cornice formation, as they are much easier to consolidate into a stable structure compared to aged snow particles. Therefore, fresh snow particles were used in the subsequent experiments.

Regarding the differences between dendritic particles and spherical particles, we add the following discussion in the section 3.4 Static force analysis of adhering particles on the cornice edge.

Lines 260 to 266:

Dendritic snow particles, which have a cohesive force approximately 1.44 times greater than that of spherical particles (Eidevåg et al., 2022), exhibit a larger angle  $\alpha$ . This larger angle indicates a broader range of balanced positions at the edge, making dendritic particles more prone to adhering at the edges. This tendency explains the experimental phenomenon that fresh snow is more likely to form snow cornices.

Additionally, smaller snow particles experience lower gravity forces. Consequently, the ratio of  $F_c$  to  $F_g$  is higher, leading to increased values of  $\alpha$  and enhancing their tendency to adhere. This finding aligns with the results of this experiment, that smaller snow particles are more likely to adhere at the edges.

**Technical corrections**

Line 19: micr-mechanism -> micro-mechanism?

**Response:** Thanks for pointing it out. We have revised it in line 29: "However, the micro-mechanism for particle adhesion to the cornice edge has not been studied in detail, due to the difficulty in observing the formation process at the particle scale"

**References:**

Eidevåg, T., Thomson, E.S., Kallin, D., Casselgren, J., Rasmuson, A. Angle of repose of snow: An experimental study on cohesive properties. *Cold Reg. Sci. Technol.* 2022, 194, 103470. doi: 10.1016/j.coldregions.

**Responses to Reviewer #2:**

**General comments**

**Comment 1:**

I appreciate very much for the efforts to observe the particle motion carefully in the wind tunnel and investigate the growing mechanism of the thin snow plate which extends to leeward from the edge. However, according to my observations in the fields and the wind tunnel experiments, it is extremely fragile and always breaks down after growing several centimeters long at the maximum.

Response: Thank you very much for your time and valuable comments. The cornice growth on the mountain ridge and causes snow avalanches has been observed by Eckerstorfer et al. (2013b), Vogel et al. (2012), and Hancock et al. (2020). From the previous observation results, cornice growth experiences multiple times of collapses and extends. At each time of growth, it may break down after a few centimeters of growth. This phenomenon is also observed in our experiment. Due to the limited field observation, it is tough to observe the particle moving around a real cornice on the mountain ridge. However, understanding the mechanism of snow cornice formation is still essential for avalanche prediction and simulation. Therefore, we carried out a wind tunnel experiment to observe the particle movement surrounding a small cornice we produced, for no matter in the field or in the wind tunnel, the particle moving and sticking laws are the same.

**Comment 2:**

It never grows to the much larger one, such as, we find on the crest of a ridge along a mountain slope and occasionally causes the avalanche release.

**Response:** Thanks for pointing this out. This is due to the involved dimensions and timescales which are way smaller/shorter in the ring wind tunnel. However, the size of the crest of a ridge is a long-term result of multiple growth events, not one precipitation event. In our experiments, the cornice can grow to more than 10 cm, which is almost the size of the model base.

**Comment 3:**

The authors need to make clear at the outset that the authors are looking at completely different phenomena. Even though it is allowed to say this miniature as "cornice" as well in a broad sense, the following numerous points should be taken into consideration before the publication.

**Response:** Thanks for your suggestions. Although there are differences to real natural conditions, we are looking at snow particles that are transported by wind across a ridge

while a fraction of the particles deposit on the lee edge resulting in horizontal and vertical snow slab (cornice) growth. As we mentioned above, no matter whether in the field or the wind tunnel, the particle moving and sticking laws are the same.

**Specific comments**

**Comment 1:**

Line 14: After "and snow cornice", Seligman et al. (1936) should be put as the reference.

**Response:** We agree with this and have revised this sentence in lines 21-23 to: "Therefore, wind can shape the snow cover and produce special patterns by redistributing snow over various areas, such as sastrugi, snow dunes (Sommer et al., 2018), and snow cornices (Seligman et al., 1936)."

**Comment 2:**

Line 29: "mechanical mechanism" sounds redundant and unnatural, although grammatically correct. I suppose only "mechanism" is fine.

**Response:** Thanks for your suggestion. We have revised this sentence in lines 36-37 to: "However, the mechanism behind the wedge-shaped (Seligman et al., 1936) snow cornice has not yet been investigated."

**Comment 3:**

Line 53 to 54: 4 m/s is the wind speed at the center of wind tunnel? If the authors would like to analyze the experimental output physically, the friction velocity  $u^*$  should be used in the manuscript instead. Furthermore, the threshold wind speed needs to be specified.

**Response:** Thanks for your suggestions. The thickness of the boundary layer in this ring wind tunnel is estimated being very thin (~2 cm) (Sommer et al., 2017; Yu et al., 2023), and the wind speed inside the wind tunnel can be considered as uniformly distributed on the cross-section. Therefore, we used the center wind speed instead of friction velocity in the manuscript. We have added a sentence in Section 2 line 63: "The wind speed inside the wind tunnel is nearly uniform with a very thin (around 2 cm) boundary layer (Sommer et al., 2017; Yu et al., 2023)."

**Reference:**

Sommer, C. G., Lehning, M., & Fierz, C. (2017). Wind crust formation: snowMicroPen data. *Journal of Glaciology*. WSL Institute for Snow and Avalanche Research SLF, Davos, Switzerland. <a href="https://doi.org/10.16904/21">https://doi.org/10.16904/21</a>

Yu, H., Li, G., Walter, B., Lehning, M., Zhang, J., and Huang, N.: Wind conditions for snow cornice formation in a wind tunnel, The Cryosphere, 17, 639–651, https://doi.org/10.5194/tc-17-639-2023, 2023.

**Comment 4:**

Figure 1: 0.125 m on the side view corresponds to the height of snow on the floor?

**Response:** Thank you for pointing this out. This value represents the height of the ridge model made of snow. The height was measured from the lower floor. In section 2, we have revised the sentence which describe the snow model in lines 64-66 to: "A ridge model with a fixed size (height 0.125 m, total length 0.4 m, flat surface length 0.1 m) is built with compacted snow before each experiment, and its side view is shown in Fig. 1."

**Comment 5:**

Line 118: "190 collision particles": Are these particles with the snow surface? Such careful explanations are lacking overall. Please check the manuscript again by standing at the place of the readers.

**Response:** Thank you for pointing this out. These 190 collision particles are moving particles in the air, and they also interact with the surface. This sentence has been revised in lines 136-139 "By maintaining the wind speed in the experiment at a constant (4 m/s), the study analyzed 655 collision particles interacting with the snow surface. These interactions included particles that rebounded, impacted, or deposited on the snow bed, as determined through an image post-processing method. Among these particles, 186 adhered to the cornice edge, while 469 adhered to the cornice upper surface."

**Comment 6:**

Figure 4 (b): Although it is described that the size of points shows the particle diameter, no explanation are found in the figure caption. Further, the corresponding specific size should be added as well.

**Response:** Thank you for pointing it out. We have re-plotted this figure and used color variation to represent the size difference, as is shown in Fig. 4(b). The explanation has been added in the figure caption: "Figure 4. (a) Cornice edge and surface, with the dashed box indicating the regions of edge and surface. (b) Impact velocity and impact angle of snow particles of different sizes (deeper color represents larger size) on edge (in blue points) and surface (in red points)."

**Figure 4.** (a) Cornice edge and surface, with the dashed box indicating the regions of edge and surface. (b) Impact velocity and impact angle of snow particles of different sizes (deeper color represents larger size) on edge (in blue points) and surface (in red points).

**Comment 7:**

Line 137: Although Stokes number: St is introduced, no explanations about the parameters are shown. Probably,  $\rho_p$  is particle density, d is particle diameter, U is wind

speed, m is kinematic viscosity and L is the characteristic length scale. Specify each value you used, particularly L. It looks the difference of St between the surface and the edge is caused by the particle diameter only. If it is the case, I am wondering it is needed to take the trouble to introduce St. As you see, St is a dimensionless number that characterizes the behavior of particles suspended in a fluid flow. It represents the ratio of the particle's inertial forces to the viscous forces exerted by the fluid. When St is much smaller than one, the particle closely follows the fluid flow, whereas St is much larger than one particle's inertia dominates, and it is less affected by the fluid flow,

tending to maintain its original trajectory. On the other hand, at  $St \approx 1$  the particle

behavior lies between these extremes, showing partial coupling with the fluid flow. In this case, both 1.8 and 3.1 are close to one and the difference is quite small; correspond to the third case together. Consequently, I have to say the discussions here are almost meaningless.

**Response:** We agree with this and have added a notation for all variables existing in this manuscript. The discussion on the Stokes number has been deleted from the revised manuscript. The notation is shown below.

**NOTATION**

| Symbol                         | Definition and units                                            |
|--------------------------------|-----------------------------------------------------------------|
| A                              | Constant $[=3.18 \times 10^{-7}]$                               |
| $A_{p}$                        | Projected area of one particle [m 2 ]                |
| D                              | Diameter of particle [m]                                        |
| $v_{ m p}$                     | Particle velocity [m s -1 ]                          |
| $v_{\rm px}$                   | Particle velocity component in x direction [m s -1 ] |
| $v_{ m py}$                    | Particle velocity component in y direction [m s -1 ] |
| $v_{ m imv}$                   | Vertical impact velocity of particle [m s -1 ]       |
| $F_{\rm c}$                    | Cohesive force [N]                                              |
| $F_{ m f}$                     | Frictional force [N]                                            |
| $F_{ m g}$                     | Gravity force [N]                                               |
| $F_{\rm s}$                    | Supporting force [N]                                            |
| $M_{ m s}$                     | Torque [Nm]                                                     |
| x                              | Radius of contact surface [m]                                   |
| t                              | Current time [s]                                                |
| $\Delta t$                     | Time step [s]                                                   |
| $T_{\rm air}$                  | Air temperature [°C]                                            |
| $oldsymbol{	heta}_{	exttt{p}}$ | Particle moving angle [°]                                       |
| $	heta_{ m im}$                | Particle impact angle [°]                                       |
| $\theta$                       | Cornice angle [°]                                               |
| $\mu_{ m f}$                   | Friction coefficient of ice surface                             |
| $ ho_{ m p}$                   | Particle density [kg/m³]                                        |
| σ                              | Tensile strength at failure [kPa]                               |
| α                              | Angle between direction of gravity and cohesion [o]             |

**Comment 7:**

Figures 5 and 7: Generally, the number of particles you analyzed is extremely small. If the authors would like to induce concrete and statistically significant conclusions, at least more than 300 particles data for edge and surface each should be corrected and examined.

Figure 7: Authors are looking at the impact velocity and the angle separately. I suppose the combination of two factors for the particles on edge and surface reveals interesting findings, supposing the enough data exists.

**Response:** Thanks for your suggestions. We have increased 383 surface particles and 121 edge particles from the experimental data. We used Kolmogorov-Smirnov test to evaluate whether the observed data significantly deviate from the fitting function. The

p-values for both surface and edge particles being higher than 0.05 suggest that the sample size is sufficient to support the analysis results.

The updated Fig. 5 and Fig. 7 are shown below. In the updated Fig. 5, the size distribution of particles follows the same trend as the previous one, with the same distribution function but different parameters. In the updated Fig. 7(a), the impact velocity distribution of edge particles maintains the same trend as before, while the distribution pattern of surface particles changed to a Gauss distribution function. In the updated Fig. 7(b), the impact angle distribution of edge particles also follows the same trend as before, but the distribution of surface particles changes to an Exponentail distribution function.

Furthermore, we combined the impact velocity and the angle to plot the vertical impact velocity, as shown in Fig. 7(c). The vertical impact velocity of both surface particles and edge particles follows the same trend.

We have revised the relevant paragraphs in Section 3.1.

**Section 3.1 Particle size distribution**

Lines 150-164:

The size distribution of particles adhering at different positions on a dynamically evolving cornice is analyzed, as is shown in Fig. 5. For all particles adhering to the cornice, their size distribution follows the log-normal distribution described by  $\theta \sim N$  ( $\mu = 5.94$ ,  $\theta = 0.36$ ). For particles adhering at the edge, their size distribution follows the log-normal distribution described by  $\theta \sim N$  ( $\mu = 5.76$ ,  $\theta = 0.43$ ). For particles adhering on the surface, their size distribution follows the log-normal distribution described by  $\theta \sim N$  ( $\mu = 5.74$ ,  $\theta = 0.47$ ). It can be concluded that particles with smaller sizes adhere more likely on the edge, and larger particles adhere more likely on the cornice surface.

**Figure 5.** Size distribution of particles at different positions. (a) All particles. (b) Adhesion particles at the edge. (c) Adhesion particles on the surface. The grey shadow represents the existence frequency of particles with different sizes.

The observed differences in particle size distribution across various locations suggest that environmental conditions, such as fluid field and gravitational effect, play an important role in influencing the adhesion of particles of different sizes. The flow on surface is a boundary layer flow, which provides a more stable environment with less turbulence, facilitating the deposition of larger particles. In contrast, the flow on the edge is a separation flow, where the ability to counteract particle gravity differs. On edge, larger particles are more likely to fall due to gravity, while smaller particles can stay on edge under cohesive forces. Moreover, smaller particles, with better followability with the wind, are more likely to stay on edge under the influence of reflux vortex behind the edge.

Moreover, we have revised the relevant paragraphs in Section 3.3.

**Section 3.3 Particle impact velocity and angle**

**Line 183-185:**

Here, we define the impact velocity/angle of the particle that deposits on the cornice surface as particle adherence velocity/angle (PAV/PAA). We first analyze the PAV and PAA of 469 particles deposited on the surface and 186 particles deposited on the edge.

**Line 191-207:**

The relative frequency of PAV/PAA represents the probability of particle adhesion on the cornice with a certain impact velocity or impact angle. As is shown in Fig. 7(a) that

for edge particles, the PAV of edge particles follows the exponential distribution function  $f_s(v_{im}) = 4.3 + 30e^{-0.9v_{imp}}$  ( $R^2 = 0.96$ ), with values mainly concentrated at below 1.5 m/s. While the PAV of surface particles follows the Gaussian distribution function of  $f_s(v_{im}) = -2.8 + \frac{79}{3.4\sqrt{\pi/2}}e^{-2((v_{imp}-2.9)/3.4)^2}$  ( $R^2 = 0.91$ ), with values mainly concentrated at 3 m/s. This indicates that particles deposited on the surface normally have a higher impact velocity than the edge. The low number of particles adhering to the surface at low impact velocities can be attributed to the wind speed in the wind tunnel, which is set at 4 m/s. At this wind speed, the majority of particles are entrained and transported at higher velocities, leaving only a small fraction of particles moving at very low velocities near the cornice's surface.

As is shown in Fig. 7(b) that the frequency of PAA of surface particles follows the Exponential distribution function  $f(\theta_{im}) = -0.8+80.6e^{-0.1v_{imp}}(R^2=0.97)$ , with values mainly concentrated below 17°. While the PAA of edge particles follows the Gaussian distribution function  $f(\theta_{im}) = 7.1 + \frac{448.8}{45.7\sqrt{\pi/2}}e^{-2((v_{imp}-18.1)/45.7)^2}$  ( $R^2=0.72$ ), with values distributed more uniformly in range. The average PAA of surface particles is 13°, which is consistent with the previous experimental results of Nishimura and Hunt (2000), as is shown in the red dash in Fig. 7(b).

Figure 7. Relative frequencies of (a) PAV, (b) PAA, and (c) vertical impact velocity of particles on edge and surface.

Furthermore, we combined the impact velocity and angle by analyzing the vertical

impact velocity ( $v_{imv}=v_{im}\sin\theta_{im}$ ) in Fig. 7 (c). The relative frequency of both surface particles and edge particles follow the exponential distribution, with surface particles  $f_s(v_{imv})=0.5e^{-v_{imv}/0.7}(R^2=0.96)$ , and edge particles  $f_s(v_{imv})=-0.1+0.5e^{-0.9v_{imp}}(R^2=0.95)$ . Particles adhere at low vertical impact velocities, whether on edges or surfaces. For both edge and surface particles, the threshold vertical impact velocity ranges from 2-2.5 m/s, with edge particles having a lower threshold velocity compared to surface particles.

It is noted that the vertical impact velocity distributions of surface particles and edge particles are in the same trend, although the impact velocity and impact angle distributions of edge particles and surface particles are different. It indicates that particle adhesion to the surface is mainly determined by the vertical impact velocity, and the differences in impact velocity and angle distributions between surface and edge is due to the fluid field differences caused by topographic changes.

**Comment 8:**

Further, here, authors set focus on saltating particles only. How do you estimate the contribution of the creep particles?

**Response**: The samples we analyzed also include creep particles (rolling over the surface). We concluded four adhering patterns of snow particles, and one of them is creep. The creep particles contribute  $\sim 13.6\%$  of the particles (using the number ratio) that stick on the cornice, which is mentioned in Section 3.2 Particle movement pattern.

**Comment 9:**

Figure 8: Please make the figure clearer. Though many parameters are introduced, it is hard to recognize what they mean and indicate, such as, a. In addition, preferably, make the right and left sides reverse to align with Figures 4 and 6.

**Response**: Thanks for your suggestions. To address the concerns raised, we have added a notation to clearly define and explain the parameters as we mentioned in Comment 7. Moreover, Fig. 8 has been adjusted by reversing the right and left sides to align with the structure of Fig. 4 and 6.

**Comment 10:**

Line 190-191: In fact, the wind speed near the surface is getting smaller. However, you cannot neglect the wind shear stress acting there.

**Response**: Thank you for this suggestion. There are two methods of calculating the drag force on a particle. First method calculates the drag force with the relative velocity between the particle and the flow:

$$F_d = 18\pi D^2 C_D (u - u_p)^2$$

where  $u_p$  is the particle velocity, u is the air wind speed,  $C_D$  is the drag coefficient, and D is the particle diameter. For particle at the edge, the air wind speed u and particle velocity are both equal to zero, therefore, the drag force is zero and can be ignored.

The second method calculates the drag force with the surface shear stress  $\tau [N/m^2]$ :

$$F_d = \tau A$$

where A [m2] is the wind-affected surface area of a particle. However, the equation  $\tau = \rho u_*^2$  is most suitable for flat surface and long-term averaged fluid field (Schlichting and Gersten, 2016). Thus, this calculation method for cornice edge is not suitable here.

Moreover, from the previous simulation and experiment studies on the fluid field of backward-facing step (DeBonis, 2022; Shehadi and Edmond. 2018), as is shown in

Fig.R1, the skin friction coefficient  $C_f = \tau_w/0.5 \rho U_{ref}^2$  drops to a very small value at the

edge (x/H=0). In which,  $\tau_w$  is the wall shear stress,  $\rho$  is the fluid density,  $U_{\rm ref}$  is the freestream velocity. It can be concluded that the drop in  $C_f$  at the edge of the backward-facing step is caused by boundary layer separation due to sudden geometric discontinuity. This separation creates a recirculating region with low or negative wall shear stress, leading to a significant reduction in  $C_f$ . Similar with our case, the edge of a cornice is the flow separation point, with a wall shear stress approximately equal to zero. Therefore, the drag force of particles on the edge can be ignored.

Figure R1. Skin friction coefficient (DeBonis, 2022)

In summary, no matter using which method of calculating the drag force of particles at the edge, its value can be considered negligible.

**References:**

Schlichting, H., & Gersten, K. (2016). Boundary-Layer Theory. Springer.

DeBonis, J.R. A Large-Eddy Simulation Of Turbulent Flow Over A Backward Facing Step. In Proceedings of the AIAA SCITECH2022 Forum, San Diego, CA, USA, 29 December 2022; p. 0337

Shehadi, Edmond. (2018). Large Eddy Simulation of Turbulent Flow over a Backward-Facing Step. 10.13140/RG.2.2.17703.24480.

**Comment 11:**

Line 191: According to your data, particle speed is quite low (nearly 0.5 m/s), thus, the compression deformation due to the collision is unlikely.

**Response**: Thank you for pointing out this important aspect. We agree that the particle speed at the edge of the snow cornice is relatively low, which makes significant compression deformation unlikely. In our analysis, we primarily focus on the role of the static balance of forces in particle accumulation at the edge. The compression deformation is expected to be minimal and negligible in this context. We have revised the relevant sections to clarify this point and ensure that the role of compression deformation is not overstated.

The revised Fig. 8 and the corresponding paragraph are shown as below:

Considering the differences in particle size distribution between the edge particles and surface particles, we conducted a static analysis of the particles at the edge. As shown in Fig. 8, a newly deposited particle i adheres to the foremost particle j at the edge of the cornice. Particle i is subjected to gravity  $F_g$ , the cohesive force  $F_c$  exerted by particle j, and the frictional force  $F_f$  at the contact surface. Due to the separation of

flow, the wind velocity and surface shear stress near the edge of the cornice are close to zero (DeBonis, 2022; Shehadi and Edmond. 2018), allowing the drag and lift forces acting on particle i to be neglected compared to other forces (Schmidt, 1980).

Figure 8. Schematic diagram of force analysis of particles adhering to the edge.

The force balance equations for particle i can expressed as:

$$F_{\rm g}\cos\alpha + F_{\rm c} = F_{\rm s} \tag{5}$$

$$F_{\rm g} \sin \alpha \leq F_{\rm f}$$
 (6)

$$F_{\rm f} = \mu_{\rm f} F_{\rm s} \tag{7}$$

Here,  $F_g$  is the gravity force on particle i,  $F_c$  is the cohesive bond force, given by  $\pi x^2 \sigma$  (Szabo and Schneebeli, 2007), where  $\sigma$  is the tensile strength at failure and x is the radius of the bond (blue shadowed area).  $F_s$  is the supporting force, and  $\alpha$  is the angle between the direction of gravity and cohesive force.  $R_i$  is the radius of particle i.

When snow particles adhere to the surface, both the gravity force  $F_g$  and the adhesive force  $F_c$  are in the vertical direction, resulting in an upward support force from the surface that maintains their stationary position. However, when snow particles adhere to the edge, the gravity force  $F_g$  and the adhesive force  $F_c$  are not in the same direction. The component of the cohesive force in the direction of the gravity force is balanced by the support force generated by the edge, while the component of the gravity force perpendicular to the adhesive force needs to be balanced by friction force  $F_f$ . Once this component exceeds the frictional force, the particles will fall.

By substituting Eq. (5) and Eq. (7) into Eq. (6), we can derive the condition for particle i to maintain mechanical equilibrium if:

$$\frac{F_c}{F_g} \ge \frac{\sin\alpha}{\mu_f} - \cos\alpha \tag{8}$$

To analyze the stability of particle i, overturning moments are calculated around point P (at the edge of the bond). The supporting force  $F_s$  and cohesive force  $F_c$  act through the center of particle i and operate on point P through the moment arm x. The gravity force  $F_g$  acts on point P through the moment arm  $R\sin(\alpha - \arcsin(x/R))$ , where the

angle between  $F_g$  and line  $\overline{OP}$  (distance from particle center to point P) is  $\beta = \alpha - \arcsin(x/R)$ . The friction force  $F_f$  acts on point P through the moment arm  $R\cos(\arcsin(x/R))$ . The condition for the particle to remain in equilibrium is  $M_s \leq 0$ . Therefore:

$$(F_s - F_c)x + F_g R sin(\alpha - \arcsin\left(\frac{x}{R}\right)) - F_f R cos(\arcsin\left(\frac{x}{R}\right)) \le 0$$
 (9)

Substituting Eq. (5) and Eq. (7) into Eq. (9) yields:

$$\frac{F_c}{F_g} \ge \frac{x/R \cdot cos\alpha + sin(\alpha - \arcsin(x/R)) - \mu_f cos\alpha cos (\arcsin(x/R))}{\mu_f cos(\arcsin(x/R))} \tag{10}$$

For the bond radius  $x \le R$  (Gubler, 1982),  $x/R \ge 0$ ,  $arcsin(x/R) \ge 0$ . Thus, we can simplify the Eq. (10) to:

$$\frac{F_c}{F_g} \ge \frac{\sin\alpha}{\mu_f} - \cos\alpha \tag{11}$$

Eq. (11), derived from the momentum equilibrium analysis, is consistent with Eq. (8), which is derived from the force balance analysis. This equation indicates that particles adhering to the edge within the range of  $[0, \alpha]$  can remain stable. The ratio of cohesive force  $(F_c)$  to gravity force  $(F_g)$  is proportional to the upper limit of angle  $\alpha$ , indicating that a higher cohesive force or lower gravity force results in a wider stable angle range for  $\alpha$ .

Dendritic snow particles, which have a cohesive force approximately 1.44 times greater than that of spherical particles (Eidevag et al., 2022), exhibit a larger angle  $\alpha$ . This larger angle indicates a broader range of balanced positions at the edge, making dendritic particles more prone to adhering at the edges. This tendency explains the experimental phenomenon that fresh snow is more likely to form snow cornices.

Additionally, smaller snow particles experience lower gravity forces. Consequently, the ratio of  $F_c$  to  $F_g$  is higher, leading to increased values of  $\alpha$  and enhancing their tendency to adhere. This finding aligns with the results of this experiment, that smaller snow particles are more likely to adhere at the edges.

**References:**

Eidevåg, T., Thomson, E. S., Kallin, D., Casselgren, J., and Rasmuson, A.: Angle of repose of snow:An experimental study on cohesive properties, Cold Regions Science and Technology, 194, 103 470,https://doi.org/https://doi.org/10.1016/j.coldregions.2021.103470, 2022.

Gubler H. Strength of bonds between ice grains after short contact times[J]. Journal of Glaciology, 1982, 28(100): 457-473.

DeBonis, J.: A Large-Eddy Simulation Of Turbulent Flow Over A Backward Facing Step, https://doi.org/10.2514/6.20220337, 2022

Shehadi, E.: Large Eddy Simulation of Turbulent Flow over a Backward-Facing Step,

Schmidt, R.. (1980). Threshold Wind-Speeds and Elastic Impact in Snow Transport. Journal of Glaciology. 26. 453-467. 10.3189/s0022143000010972.

**Comment 12:**

Line 216: Authors say that the sintering time is much longer than the collision time in the manuscript at lines 26-27 and exclude the mechanism to create the wedged-shaped form. I do not understand why it appeared again. It looks contradictory statements. Furthermore, the static electric forces have been observed in the specific cases so far, and not always.

Response: Thank you for pointing out the unclear point in the manuscript. I understand the concern about the seemingly contradictory statements regarding the role of sintering in the formation of snow cornices. I would like to clarify that in lines 26-27, it was stated that the sintering time is much longer than the collision time when a particle impacts the cornice surface. This means that on the timescale of particle sintering, the drifting snow aerodynamic process dominates the redistribution of snow particles and the patterns of snow cornices. However, the intention was not to imply that the sintering force is unimportant in maintaining the shape of snow cornice. Rather, the point is that during the initial stage of cornice formation, the drifting snow aerodynamic process is the primary driver of the distribution of snow particles. After these particles are deposited on the edge, the sintering force then dominates the main place in keeping the particle stationary, thus shaping and preserving the cornice structure.

Additionally, the duration of a particle's contact time on the surface is not particularly significant. It only needs to be sufficient for the sintering force and gravity to overcome the shear and rebound forces, ensuring the particle remains stationary. Besides, the impact time is strongly influenced by surface roughness and particle morphology. The rougher the surface, the more likely the particle gets caught and stay long enough so that sintering can make it ultimately sticking at the cornice edge.

To be clear, sentences in lines 26-27 have been deleted. Besides, we agree on the point that the static electric force is only important in specific cases, such as low wind speeds and high charge on particles. Therefore, we have deleted the definition of electricity force and the electric field.

**Comment 13:**

Although authors tried to introduce all the conceivable forces which may act between particles on edge, time scales are not always consistent. Furthermore, all the discussions are qualitative from beginning to end and no quantitative estimates, which is enough to

keep the thin plate growing, are shown. To say the least, quantitative approval of their idea, based on the snow and environmental data obtained in the experiment, is essential to make the manuscript reasonable and worthwhile.

Response: We sincerely thank the reviewer for their thoughtful feedback. We understand the concern regarding the lack of quantitative estimates and the consistency of time scales in our manuscript. Below, we address these points in detail and outline the revisions we have made to strengthen the manuscript. Regarding the time scale consistency, this concern has been answered in Comment12 (the main point is the duration of a particle's contact time on the surface is not particularly significant. It only needs to be sufficient for the sintering force and gravity to overcome the shear and rebound forces, ensuring the particle remains stationary). Regarding the quantitative estimates, we have addressed the reviewer's request to highlight the quantitative results more explicitly in the abstract and conclusion sections and added additional explanations in the relevant sections.

**Abstract:**

Snow cornices are a common snow pattern in cold regions, and their fracture and collapse can easily trigger avalanches. Despite numerous observations and experimental simulations on their formation process, the microscopic mechanism of their formation remains unclear. In this paper, based on wind-tunnel experiments and high-speed photography, experimental studies on the trajectory of particles surrounding the snow cornice were carried out. The experiment results reveal the distinct differences in particle size, impact velocity, and impact angle between the surface and edge of a cornice. The findings show that the edge of a cornice is primarily composed of small snow particles, with saltation being the dominant movement pattern for particle adhesion. The distributions of impact velocity and angle of particles differ between the edge and the surface. The relative frequency of particle adhesion on the edge exponentially decreases with increasing impact velocity, while surface adhesion follows a Gaussian distribution. These differences are primarily attributed to topographic effects. Analysis of vertical impact velocity distributions reveals that both edge and surface particles follow the same exponential trend, with threshold velocities ranging from 2 to 2.5 m/s, indicating a similar adhesion mechanism. To further explain the observed differences in particle size between the edge and surface, the forces acting on particles adhering to the edge were analyzed. The results show that smaller or dendritic particles are more likely to adhere to the edge due to a higher cohesive-togravity force ratio  $(F_c/F_g)$ . This study quantitatively provides insights into the micromechanism of snow cornice formation, offering a theoretical foundation for improving avalanche prediction.

**Conclusion:**

Our findings reveal that near-surface saltation and creeping are the primary modes for particles to adhere to the cornice. Among the adhered particles, the majority are saltating particles that settle on the surface, while only a few deposit directly at the front end. Additionally, some creeping particles interlock and hang with others at the cornice edge, and a small number of particles may detach from the edge and move

backward due to opposing forces acting against the flow. These varying movement patterns of particles contribute to the increase in both the thickness and length of the cornice, which is essential for its structural growth.

The experiment results show that although the distributions of the size, impact velocity, and impact angle of particles on surface and edge are different, the vertical impact velocity distribution is consistent, with threshold velocities ranging from 2 to 2.5 m/s for adhesion. Quantitative analysis demonstrates that the relative frequency of edge particle adhesion decreases exponentially with impact velocity, while surface adhesion follows a Gaussian distribution. This indicates that the particle adhesion is dominated by the vertical impact velocity. The variations in particle size, impact velocity, and impact angle distributions arise from the distinct fluid field generated by sudden topographic change.

Moreover, the cornice edge is primarily composed of lightweight snow particles compared to the cornice surface. This phenomenon can be attributed to the mechanics of particle adhesion, where the ratio of cohesive forces to gravity forces plays a critical role. Smaller particles, particularly dendritic ones, are more likely to adhere to the edge due to their favorable physical properties (with higher  $F_c/F_g$  ratios), which enhance their stability in the presence of wind and other forces.

Overall, this research provides valuable insights into the micro-mechanisms of snow cornice formation, emphasizing the critical roles of particle size, movement patterns, and environmental conditions. The findings have important implications for avalanche prediction and management, as understanding snow cornice dynamics can help mitigate

risks associated with their fracture and collapse in cold regions. Furthermore, these insights may extend to related phenomena, such as the formation of snow bridges in ice crevasse, wire icing, and snow accumulation on train bogies, highlighting the broader relevance of particle adhesion mechanisms. Future studies should continue to explore the interactions between environmental factors and particle behavior to refine our understanding of snow cornice dynamics.

---

## Referee Report (RR1)

**General Comments**

As noted in my previous review, the experimental procedures presented in this manuscript are highly informative and will undoubtedly be valuable for researchers interested in particle motion. The authors' efforts to investigate the growth mechanism of the thin snow plate extending leeward from the edge are commendable, and the overall experimental procedure is well documented.

However, as also noted in my earlier review, I still feel that the study falls short of clarifying the growth mechanism of natural snow cornices. The differences between this miniature experiment and real snow cornices found in nature cannot be explained solely by variations in terrain size, successive precipitation, and duration. The authors' explanations remain unsatisfactory in this regard. I would like to emphasize once again that the authors appear to be examining fundamentally different phenomena.

If the authors wish to assert that the thin plate observed in this study is relevant to understanding natural snow cornice formation, they should provide a clear scenario—ideally illustrated with schematic figures—that shows how the thin plate would develop step by step into a real cornice, specifying the key mechanisms involved at each stage.

Before addressing the detailed points within the manuscript, I must also point out several discrepancies between the authors' responses and the content of the revised manuscript:

Discrepancies Between Author Replies and Manuscript

Contact Collision Explanation:

The authors state in their reply:

"For clarity, we will add the following sentences before introducing the maximum compression displacement of particles in Section 3.2: 'During contact collision, snow particles will be compressed and deformed, undergoing plastic deformation and brittle failure (Wang et al., 2020).'"

However, neither the text nor the reference appears in the manuscript.

Sintering Force Description:

In the reply, the authors note:

"Sentences in lines 26–27 will be deleted, and we will add: 'In which, Fb is the sintering force, calculated as the product of the ice tensile strength and the contact surface area

(Szabo and Schneebeli, 2007). Sintering begins upon particle deposition and plays a crucial role in stabilizing and preserving the cornice structure." Yet, this explanation and the notation for Fb are not included in the manuscript. The same applies to subsequent mentions of sintering effects—the manuscript still lacks these explanations. Order of Magnitude Description: The authors replied: "The size of snow particles follows a distribution function, and different sized particles experience different magnitudes of force. Therefore, we present orders of magnitude rather than exact values." Yet the manuscript contains no mention or expression of "order of magnitude." Missing Reference: The following reference, cited in the authors' reply, does not appear in the manuscript:

Enliang Wang et al., 2021, Cold Regions Science and Technology, 182, 103215.

Specific Comments

**Figure S1:**

It seems the authors intended to show that shear stress becomes negligible, but it is difficult to interpret what is being presented. More detailed explanations are required, particularly regarding the reason for plotting two data series between 0.7 and 2 (x/H). Since computer simulations of airflow were conducted, I strongly recommend including a representative airflow pattern around the edge. This would be extremely helpful in explaining the particle movements discussed later.

**Line 58 & Figure 1:**

The description of the wind tunnel setup is inadequate. Please include key specifications such as the length and size of the working section.

**Line 67:**

The authors mention preliminary tests comparing two types of snow and concluding that dendritic snow is more suitable. Were the particle sizes of both types the same?

Since the manuscript later emphasizes particle size as a key factor, it's important to clarify this point.

**Line 80:**

The phrase "for 4–5 s during cornice growth" would be clearer if supplemented with figures showing the time evolution of the thin plate's length and thickness.

**Line 109:**

Since fresh dendritic snow tends to orient perpendicular to the wind to maximize resistance, particle size might be underestimated when viewed from the side. This effect would be negligible for rounded particles but should be considered for dendritic ones.

**Line 137:**

Are any particles ejected by collisions with saltating particles? If so, they might move slowly and contribute to edge growth. Please clarify this point.

**Line 143:**

The statement "smaller particles, with better followability with the wind" suggests that their impact speed should be higher, which conflicts with Figure 7. Additionally, larger dendritic particles typically have more branches, potentially increasing their likelihood of being trapped at the edge. I recommend introducing quantitative parameters such as specific surface area to strengthen this discussion.

**Lines 158-164:**

These descriptions are speculative and qualitative. As noted earlier, it would greatly improve the discussion to include airflow patterns around the edge. If direct measurements using a hot-wire anemometer were not conducted, simulated streamlines or vortex separations would be very helpful.

**Lines 169–170:**

There's an inconsistency here. The authors state that creeping particles (about 14%) are larger and settle near the front end of the cornice, but later conclude that smaller particles are more likely to adhere at the edge. This contradiction needs to be resolved.

**Lines 176–181 & Figures 6–7:**

How many particle trajectories were analyzed to derive the appearance ratios in Figure 6? Is the sample size sufficient for quantitative conclusions? Also, how were impact speeds and angles in Figure 5 determined under the complicated particle movement?

Were these captured at time step 4? If so, negative angles should appear in Figure 7 as well.

Moreover, Figure 7's horizontal axis is labeled "impact velocity or angle," while the figure caption refers to "particle adherence velocity or angle." This should be unified for clarity.

**Lines 182–218 & Figure 7:**

The explanation for why higher-speed particles adhere at certain positions but not near the edge remains unclear. Generally, high-speed, low-angle impacts would result in rebound. Further, do you have evidence that the vertical component of velocity predominantly dictates particle behavior? A clearer discussion is needed.

**Lines 218-266:**

The model describing forces between particles at the edge feels redundant, as all subsequent discussions are qualitative. As noted previously, quantitative validation using experimental snow and environmental data is essential. The claim that smaller dendritic particles adhere more readily due to a higher Fc/Fg ratio can be easily speculated without introducing the model.

Given the complexity of dendritic particle shapes, factors such as branching and interlocking likely have greater influence. Additionally, the dependence of Fc/Fg on particle size is unclear. If the authors wish to explore particle size effects more rigorously, I strongly recommend analyzing the cornice structure post-experiment to

measure particle size distributions and dendricity. This would greatly enhance the study's credibility.

Line 224:

The authors state that wind velocity and shear stress near the cornice edge approach zero. Please provide clear evidence for this. Figure S1 is unsatisfactory and hard to interpret. Moreover, since the ridge model differs from natural terrain shown in the reference, the airflow at the edge may not reduce to zero but instead decrease rapidly on the leeward side.

Line 275:

The claim about increases in both the thickness and length of the cornice should be supported by time-series data. If such data were obtained, please present them.

Additional Comment:

Looking at Figures 4 and 6, it appears that snow accumulation on the flat surface at the model's right end increased. This area may play a key role in conveying rolling particles toward the edge and increasing the cornice root's thickness. This process could be crucial to understanding how thin plates evolve into full cornices in nature. This process deserves attention in future discussions and analyses.

---

## Referee Report (RR2)

General comments

An explanation of the effect of particle shape (spherical/dendritic) on snow cornice formation has been added.

The discussion of the balance of forces has been simplified without introducing excessive parameters, resulting in increased reliability, and consistency with the experimental results has also been noted.

Other explanations of the experimental results have also been revised for clarity, and as a result the paper is now acceptable for publication with some corrections.

Specific comments

Equation (3): Although this notation is understandable, it would be clearer to write the expression within the square root as $(v_{px}(t))^2 + (v_{py}(t))^2$.

Line 248: Rcos(arcsin(x/R)
A bracket is missing.

Lines 293-294: …and snow accumulation on train bogies…
It is unclear which state of accumulation is considered. If the accumulation is not due to natural wind, but rather to the movement of the train, additional factors such as different speed ranges and mechanical heat generation would have to be considered.

Lines 327-330: The same paper by Eckerstorfer is listed as 2013a and 2013b.

Line 351: The author Seligman is listed twice.

---

## Editor Decision (ED1)

**Snow Particle Motion in Process of Cornice Formation**

Hongxiang Yu1,2, Li Guang1, Benjamin Walter4, Jianping Huang1, Ning Huang2, and Michael Lehning3,4

Correspondence: Ning Huang (huangn@lzu.edu.cn) and Michael Lehning (lehning@slf.ch)

**Abstract.** Snow cornices are a common snow pattern in cold regions, and their fracture and collapse can easily trigger avalanches. Despite numerous observations and experimental simulations on their formation process, the microscopic mechanism of their *initial stage of* formation remains unclear. In this paper, based on wind-tunnel experiments and high-speed photography, experimental studies on the trajectory of particles surrounding the snow cornice were carried out. The experiment results reveal the distinct differences in particle size, impact velocity, and impact angle between the surface and edge of a cornice. The findings show that edge-deposited particles are generally smaller and more dendritic, attaching mainly through low-velocity saltation and mechanical interlocking. while surface deposition is dominated by larger, faster particles, the edge of a cornice is primarily composed of small snow particles, with saltation being the dominant movement pattern for particle adhesion. The different probability distributions of impacting velocities and angles in these two regions are attributed to variations in airflow and local cornice topography. Both surface and edge regions, however, exhibit a characteristic vertical impact velocity threshold of 2-2.5 m/s, which is the dominant parameter governing particle adherence or rebound. The distributions of impact velocity and angle of particles differ between the edge and the surface. The relative frequency of particle adhesion on the edge exponentially decreases with increasing impact velocity, while surface adhesion follows a Gaussian distribution. These differences are primarily attributed to topographic effects. Analysis of vertical impact velocity distributions reveals that both edge and surface particles follow the same exponential trend, with threshold velocities ranging from 2 to 2.5 m/s, indicating a similar adhesion mechanism, A static adhesion model incorporating particle morphology parameters for edge deposition was developed and experimentally validated, confirming its effectiveness in predicting the influence of particle size, shape, and adhesion thresholds. Overall, this research reveals the microdynamics underlying initial cornice growth, providing a theoretical basis for avalanche modeling and infrastructure protection in alpine environments, as well as offering a methodological and mechanical framework for studying snow and ice adhesion in both natural and engineered systems. To further explain the observed differences in particle size between the edge and surface, the forces acting on particles adhering to the edge were analyzed. The results show that smaller or dendritic particles are more likely to adhere to the edge due to a higher cohesive-to-gravity force ratio

<sup>1College of Atmospheric Science, Lanzhou University, Lanzhou, 730000, China

<sup>2College of Civil Engineering and Mechanics, Lanzhou University, Lanzhou, 730000, China

<sup>3College of Architecture Civil and Environmental Engineering, Ecole Polytechnique Federal de Lausanne, Lausanne, 1015, Switzerland

<sup>4WSL Institute for Snow and Avalanche Research SLF, Davos, 7260, Switzerland

25  $(F_c/F_g)$ . This study quantitatively provides insights into the micro-mechanism *involved in the early stage* of snow cornice formation, offering a theoretical foundation for improving avalanche prediction. The finding can be further generalized to address snow accumulations on mechanical structures, power lines, and other systems where similar force balance considerations are critical.

**NOTATION**

| Symbol            | Definition and units                                                       |
|-------------------|----------------------------------------------------------------------------|
| A                 | Weighting parameter for dendricity]                                        |
| $A_p$             | Projected area of one particle $[m^2]$                                     |
| D                 | Diameter of particle [m]                                                   |
| $R_{th}$          | threshold radius for particles adhering to the edge. [m]                   |
| $v_{ m p}$        | Particle velocity [m s -1 ]                                     |
| $v_{ m px}$       | Particle velocity component in x direction [m $\rm s^{\text{-}1}]$         |
| $v_{\mathrm{py}}$ | Particle velocity component in y direction $[m\ s^{-1}]$                   |
| $v_{imv}$         | Vertical impact velocity of particle $[m\ s^{-1}]$                         |
| $F_c$             | Cohesive force [N]                                                         |
| $F_f$             | Frictional force [N]                                                       |
| $F_g$             | Gravity force [N]                                                          |
| $F_s$             | Supporting force [N]                                                       |
| $M_s$             | Total moment [Nm]                                                          |
| x                 | Radius of contact surface [m]                                              |
| t                 | Current time [s]                                                           |
| $	au_b$           | Bond shear stress [Pa]                                                     |
| $\Delta t$        | Time step [s]                                                              |
| $T_{air}$         | Air temperature $[^{o}C]$                                                  |
| $	heta_p$         | Particle moving angle $[^o]$                                               |
| $	heta_{im}$      | Particle impact angle $[^o]$                                               |
| $\theta$          | Cornice angle $[^o]$                                                       |
| $\alpha$          | Angle between direction of gravity and cohesion force $\left[ ^{o}\right]$ |
| $\alpha$          | Angle between direction of gravity and cohesion force $\left[ ^{o}\right]$ |
| $\beta$           | Angle between direction of gravity and line $\overline{OP}$ $[^o]$         |
| $\mu_f$           | Friction coefficient of ice surface                                        |
| $ ho_i$           | Ice density $[kg/m^3]$                                                     |
| $\delta$          | Ratio of contact radius to particle radius                                 |
| dd                | Dendricity                                                                 |
| SSA               | Specific surface area $[mm^2/mm^3]$                                        |

**30 1 Introduction**

35

Snow consists of ice crystals. Snow particles may adhere to the surface at particle-bed collision. Therefore, wind can shape the snow cover and produce special patterns by redistributing snow over various areas, such as sastrugi, snow dunes (Sommer et al., 2018b), and snow cornices (Seligman et al., 1936). Snow cornice is one of the naturally formed accumulation patterns in cold mountain regions. *Their*The collapse of snow cornices may induce snow avalanches (Vogel et al., 2012).

Previous field observations have consistently shown that the optimal wind speed range for cornice formation lies between one and two times the threshold wind speed (Vogel et al., 2012; Eckerstorfer et al., 2013; Hancock et al., 2020). Recently, wind-tunnel experiments have further shown that cornice growth is maximized when the wind speed exceeds the threshold value by 40% (Yu et al., 2022). However, the micro-mechanism for particle adhesion to the cornice edge has not been studied in detail, due to the difficulty in observing the formation process at the particle scale.

The growth of a snow cornice can be divided into several stages (Montagne, 1980; Vogel et al., 2012; Eckerstorfer et al., 2013). In the initial stage, a thin slab forms at the mountain edge (highlighted in red in Fig. 1), mainly by adhesion of wind-transported snow particles. When more snow accumulates on the relatively flat surface above the edge, it can gradually be conveyed toward the slab tip—especially via wind-transported particles—thereby increasing the thickness at the cornice root. This sustained supply of snow from the platform region plays a key role in the transformation of a small slab into a fully developed cornice in nature. In the subsequent stage, repeated deposition from intermittent drifting and precipitation successively adds new layers of snow to the cornice. This layer-by-layer accumulation is accompanied by a gradual increase in both length and thickness of the cornice. As the cornice grows larger, the overhanging mass of snow is increasingly influenced by gravitational forces, which may cause it to bend downward (shown in the white dashed line in Fig. 1) and promote internal compaction near the edge. Eventually, when the cornice becomes too large and shear stress exceeds a critical threshold, it breaks off and collapses. The evolution of a wedge-shaped cornice—from initial slab formation to subsequent snow accumulation on the flat surface—has been experimentally investigated in our previous work (Yu et al., 2022), with particular focus on the relationship between cornice growth rate and air mass transport. However, the specific mechanisms governing the very initial stage, that is, how airborne snow particles first adhere and accumulate to form the small slab at the edge, remain unexplored.

Previous field research mainly focused on the morphology variation due to limitations of observation equipment (Vogel et al., 2012; Eckerstorfer et al., 2013; van Herwijnen and Fierz, 2014; Hancock et al., 2020), for observing how particles adhering to mountain edges are hardly realized. Currently, there are several hypotheses on how snow particles adhere at the edge: Irregularly-shaped snow particles interlock with each other by their dendrites (Seligman et al., 1936); Charged snow particles are attracted by the strong electric field above the snow cornice surface (Latham and Montagne, 1970); Pressure melt and frictional heat (Latham and Montagne, 1970) when particles contact other

Figure 1. Schematic figure of different stages of a snow cornice growth.

surfaces leads to a quasi-liquid layer facilitating fast sintering. However, the mechanism behind the wedge-shaped (Seligman et al., 1936) snow cornice has not yet been investigated.

Cornice growth is often accompanied by drifting snow (Eckerstorfer et al., 2013; Yu et al., 2022), in which a snow particle saltation layer exists. Drifting snow particles move in three modes, namely, creep, saltation, and suspension, with the first two modes contributing most to the snow mass transport (Bagnold, 2012). When snow particles collide with the surface, three processes may occur: 1) Rebound occurs when a portion of particle kinematic energy is lost, but may rebound from the surface; 2) Deposition occurs when the particle loses all its kinematic energy upon impact with the ground; 3) Eject occurs when the particle transfers the kinematic energy to other particles on the ground upon impact with surface, resulting in the entrainment of other particles resting on the surface.

During particle-surface collision, the particles transfer momentum and energy from the air to the surface. These processes are characterized using rebound and splash functions based on theoretical models (Lämmel et al., 2017; Comola and Lehning, 2017) or observations (Anderson and Haff, 1991). Particle impact velocity and impact angle (Walter et al., 2023) are two critical factors influencing key processes such as rebounding, deposition, and splashing. These parameters play a significant role in determining how particles interact with surfaces upon collision. However, the specific method by which particle impact velocity and impact angle influence particle adhesion, particularly in the context of snow cornice formation, remains poorly understood. Thus, investigation is needed to clarify the relationship between these impact parameters and the adhesion mechanisms.

Here, we carry out a wind-tunnel experiment of cornice formation, focusing on particle trajectory and adhesion process in snow cornice formation. Based on the experimental results, we investigate the micro-physical mechanism for cornice formation.

**2 Instruments and methods**

100

105

110

The wind-tunnel experiments are carried out in a ring wind tunnel in the cold lab of the WSL Institute for Snow and Avalanche Research, in Davos, Switzerland. The experiment setup is shown in Fig. 2. The working section of the wind tunnel is 1 m in length, with a cross-section area of 0.2 m (width) × 0.5 m (height), and has been successfully used for several drifting snow experiments (Wahl et al., 2024; Walter et al., 2024; Yu et al., 2022; Sommer et al., 2017, 2018a). Further details of the wind tunnel can be found in Yu et al. (2022). For tracing particle trajectories, a high-speed camera system is deployed. A feeding system is used to supply snow particles. The feeding rate of snow particles is manually kept stable. The snow particles are produced by a snow maker (Schleef et al., 2014). The geometrical diameters are 300 - 500  $\mu$ m, and the shape of the snow crystals is dendritic, visually analyzed under a microscope. The room temperature of the cold lab is controlled and set at -5 °C, and the wind speed is kept at 4 m s-1. The wind speed inside the wind tunnel is nearly uniform with a very thin (around 2 cm) boundary layer (Sommer et al., 2017; Yu et al., 2022). A ridge model with a fixed size (Fig. 2) is built with compacted snow before each experiment. A ridge model with a fixed size (height 0.125 m, total length 0.4 m, flat surface length 0.1 m) is built with compacted snow before each experiment, and its side view is shown in Fig. 2.

Before conducting the experiment, we have performed preliminary tests on both fresh snow particles and aged snow particles. Fresh snow particles, characterized by their highly dendritic shapes, were compared to decomposed snow particles, which are characterized by small rounded shapes after being stored for several days at a constant temperature of  $T_{air} = -10$  °C. The results show that both types of snow particles are capable of forming a snow cornice. However, fresh snow particles exhibit a significantly higher propensity for cornice formation and are much easier to consolidate into a stable structure compared to aged snow particles. Therefore, fresh snow particles were used in the subsequent experiments.

The high-speed camera system consists of a high-speed camera (Phantom VEO710), an LED lamp as a light source, and a transparent plane positioned at the opposite side of the camera to diffuse the light source and achieve uniform illumination. The system is employed for shadowgraphy analysis. The shadowgraphy is a technique used for extracting the moving particle's size, velocity, and trajectory. It is similar to PTV (particle tracking velocimetry) (Baek and Lee, 1996; Tagliavini et al., 2022), which has the same algorithms for particle information, and it has been frequently used in recent snow-related studies, for its advantages in robustness, non-intrusiveness, and accessibility. Shadowgraphy is particularly suitable for tracking snow particles, which are partially transparent and irregular in shape. It has been successfully applied to measuring particle mass flux, velocity, and size distribution (Paterna et al., 2016; Walter et al., 2023). The sampling frequency of the camera is set as 3 kHz, corresponding to a time interval of 333.32 µs.

Figure 2. Schematic for experimental setups.

A total of 18 cases were conducted during the cornice growth, with each case lasting 4-5 seconds and yielding 12455 images to record the particle trajectories. The duration of each case was limited by the camera memory. A sequence of the different growth steps is illustrated in Fig. 3(a) of Yu et al. (2022). The images are captured for 4-5 s during cornice growth. In each case of the experiment, 12455 images are captured to record the particle trajectories After the images are obtained, particle-bed collision events are selected visually. Then, particle sizes and trajectories in these events are analysed by image processing as follows.

**2.1 Particle recognition**

[revised manuscript text omitted]

The  $k_{th}$  target particle's horizontal and vertical velocity at a given time step  $\Delta t$  is calculated as:

$$v_{px}(t) = \frac{x(t + \Delta t) - x(t)}{\Delta t} \tag{1}$$

165
$$v_{py}(t) = \frac{y(t + \Delta t) - y(t)}{\Delta t} \tag{2}$$

in which x and y are the coordinate positions in x-axis and y-axis, t is the current time, and  $\Delta t$  is the time interval of a high-speed camera. Therefore, the magnitude of particle velocity  $v_p$  is:

$$v_p(t) = \sqrt{(v_{px}(t))^2 + (v_{py}(t))^2}$$
(3)

where  $v_{px}$  and  $v_{py}$  are the velocity component in x and y direction, respectively. The particle moving angle  $\theta_p$  can be calculated as:

$$\theta_p(t) = \arctan(\frac{|v_{py}(t)|}{|v_{px}(t)|}) \tag{4}$$

A particle contact event is defined when the particle's vertical velocity is from negative (downward) to zero or positive (upward) among several adjacent images. The impact velocity is then defined as the velocity before the contact, and the rebound velocity is defined as the velocity after the contact. We found the maximum error between visual observation (manually tracking the particle's trajectory) and program recognition of the edge particle velocity to be 5%, the angle to be 18%, and the diameter to be 16%.

**3 Results and discussions**

175

180

185

195

By maintaining the wind speed in the experiment at a constant (4 m/s), the study analyzed 655 collision particles interacting with the snow surface. These interactions included particles that rebounded, impacted, or deposited on the snow, as determined through an image post-processing method. Among these particles, 186 adhered to the cornice edge, while 469 adhered to the cornice upper surface. The cornice edge here refers to the spatial range of the 1 mm vertical front end of a dynamically growing cornice, and the cornice surface refers to the cornice's topside with the total length at the current time step minus the edge length, as is shown in Fig. 5(a).

The distributions of the snow particle impact velocity, impact angle, and size distribution are shown in Fig. 5(b). The blue points represent the particles that adhere to the edge and the red points represent those that adhere to the surface. The size of all points represents the particle's diameter. It can be concluded that particles that adhere to the edge have relatively lower and more narrowly-distributed values of impact velocity, but widely-distributed impact angle, compared to the particles that adhere to the surface.

To investigate the differences between edge and surface particles, we analyzed the adhere particle's size distribution, impact velocity, and impact angle in the following sections.

**3.1 Particle size distribution Particle size and shape**

The size distribution of particles adhering at different positions on a dynamically evolving cornice is analyzed, as is shown in Fig. 6. For all particles adhering to the cornice surface, their size distribution follows the Log-normal distribution described by  $\theta \sim N(\mu=277.17, \theta=0.45)$ , with the average diameter  $\overline{D}_{all}=340$  um. For particles adhering at the edge, their size distribution follows the Log-normal distribution described by  $\theta \sim N(\mu=264.42, \theta=0.38)$ , with the average diameter  $\overline{D}_{edge}=329$  um. For particles adhering to the surface, their size distribution follows the Log-normal distribution function described by  $\theta \sim N(\mu=342.69, \theta=0.28)$ , with the average diameter  $\overline{D}_{surface}=405$  um. It can be concluded that particles with smaller sizes adhere more likely on the edge, and larger particles adhere more likely on the cornice surface.

Figure 5. (a) Cornice edge and surface, with the dashed box indicating the regions of edge and surface. (b) Impact velocity and impact angle of snow particles of different sizes (deeper color represents larger size) on edge (in blue points) and surface (in red points).

The observed differences spatial variations in particle size distribution across various locations suggest that environmental conditions, such as specifically the fluid field and gravitational effect, play an important role in influencing the adhesion of particles of different sizes. The flow on the cornice surface is characterized as a boundary layer flow, providing a comparatively stable which provides a more stable environment with less low turbulence intensity, which facilitates fact the deposition of larger particles. In contrast, the flow on the edge is a separation flow, where the ability to counteract particles larger particles adhering on the edge are more readily removed by gravity and turbulent fluctuations, while smaller particles are

In addition to particle size, dendricity and specific surface area (SSA) are important indicators of particle morphology and surface characteristics. As illustrated in Fig. 7(a), the average dendricity of edge particles is 1.9, higher than that of surface 1.4. Meanwhile, the distribution range of the edge particles (1 to 4.7) is broader than that of the surface particles (1 to 3.1). These results indicate that the edge particles have more fragmented or branched morphologies, while particles on the surface are generally more regular and compact.

However, the SSA values of edge particles and surface particles are similar, with the average value of  $20 \text{ mm}^2/\text{mm}^3$ , as is shown in Fig. 7(b). This similarity arises because both edge and surface particles originate from the same snow source. Therefore, dendricity is a more critical factor in determining whether a particle can adhere to an edge or the surface. In particular, edge particles with high dendricity have more contact points with neighboring particles on the cornice, which may lead to greater cohesion force  $F_c$  - the force counteracts the gravity force  $F_g$  and allows the edge particle to adhere. In contrast, surface particles may experience less cohesion force, and their gravity acts in the same direction as the cohesion force, making gravity either irrelevant or even beneficial for particle adherence.

215

**Figure 6.** Size distribution of particles at different positions. (a) Airborne particles. (b) Adhesion particles at the edge. (c) Adhesion particles on the surface. The gray shadow represents the existence frequency of particles with different sizes.

Combining the analysis of particle size, dendricity, and SSA distribution, we find that smaller particles and dendritic particles are more prone to adhere to the edge, while larger and more spherical particles tend to deposit on the surface. This phenomenon is closely related to the aerodynamic behavior of particles in air. Specifically, the pattern of particle deposition is primarily governed by the Stokes number (Comola et al., 2019), a dimensionless parameter that compares the inertial response time (particle relaxation time) of a particle to the characteristic time scale of the fluid flow. In general, particles with smaller sizes, as well as large particles with irregular shapes, tend to have lower relaxation times than spherical particles of the same size (Loth, 2008). As a result, for such small particles and large, highly dendritic ones, viscous forces dominate over inertia, enabling the particles to quickly respond to changes in local fluid velocity and closely follow the streamlines. Such particles therefore preferentially deposit on the edges—where the wind speed is near zero and accompanied by a reflux vortex (DeBonis, 2022). In contrast, for spherical particles, especially those with larger size, inertial forces become more significant relative to viscous drag, making these particles less responsive to fluid velocity changes and more likely to deposit on the main surfaces—which are generally characterized by a stable boundary layer and low turbulence.

Figure 7. Frequency distributions of (a) dendricity and (b) specific surface area (SSA) for particles adhering to the edge and surface.

**3.2 Particle movement pattern**

235

240

245

Near surface moving particles were captured by using high-speed camera. Creeping (particles rolling or sliding over the surface before retaining) and saltating (particle successively jumping over the surface before settling) (Bagnold, 2012) are the two primary modes contributing to cornice growth. Creeping particles (Fig. 8(a)), which account for about 14 % of the observed particles, represent the minority of larger-sized adhered particles, and they typically move slowly. are typically larger and move more slowly. These particles are mainly entrained from the surface under the ejection of other particles. Most of them retain on the cornice surface, and only a small fraction - with elongated dendrites - are able to interlock and remain adhered at the edge (Fig. 8(b)).

Here, we mainly captured the particles moving very close to the surface by using a high-speed camera. It is observed that creeping (particle rolling or sliding over the surface before retain) and saltating (particle successively jumping over the surface before settling) are the two main types of particles contributing to the growth of the cornice. Creeping particles are entrained from the surface under the ejection of other particles. These creeping particles (about 14%) normally have larger sizes and lower velocities, which could contribute to edge growth. After undergoing multiple rotations in the downstream direction, some of the creeping particles come to a retain near the front end of the cornice (Fig. 6(a)), as is shown in 6(a), and some of them keep rolling until they reach the edge and rollover. Only a few creeping particles with clongated dendrites were found to physically interlock with adjacent particles at the front end of the cornice. Under the action of gravity, the hanging particle starts rotation with the interlocking point as the force center (Fig. 8(b), steps 2-4).

Figure 8. Four adhering patterns of snow particles (1,2,3,4 represents for different time steps).(a) Creeping particles. (b) Hanging particles. (c) Impacting particles. (d) Back moving particles.

In addition, saltating particles observed near the cornice originate either from airborne trajectories or from ejection off the surface. Whether saltating particles rebound or deposit on the cornice surface after impact is strongly influenced by their impact position, velocity, and angle. Among all the particles that settle on the cornice surface, about 82% are saltating particles that deposit before reaching the cornice's front end, as is shown in Fig. 8(c). Only an extremely small number of Only a few of particles deposit on the front end, which are smaller-sized particles ejected near the edge. Fewer saltating particles going off the edge will later move backward to the cornice edge, under the action of the reflux vortex or the potential electric field, as shown in Fig. 8(d).

**3.3 Particle impact velocity and angle**

260

265

Here, we define the  $v_{im}$  is the impact velocity/angle of the particle that deposits on the cornice surface as particle adherence velocity/angle (PAV/PAA). $\theta_{im}$  is defined as the angle of particle incidence on the horizontal cornice surface, ranging from  $\theta^o$  (parallel to the surface) to  $\theta^o$  (perpendicular to the surface), and this angle measures how steeply a particle approaches the surface or edge before sticking. Only particles impacting the cornice surface from above (incidence angle  $\theta - \theta^o$ ) are considered, while particles with trajectories suggesting backward-moving ( $\theta_{im} > \theta^o$ ) are excluded. We first analyze the  $v_{im}$ PAV and  $\theta_{im}$ PAA— of 469 particles adhereddeposited on the surface and 186 particles adhereddeposited on the edge. In natural conditions, the snow cornice will slowly bend and deform under the gravity force. In our experiment, the snow cornice can be considered as horizontally growing in the whole process, and the bending effect of the snow cornice can be neglected for the short observation time. The

PAA impact angle is defined as the angle of particle incidence on the horizontal cornice surface. We subdivided the values of  $v_{im}$  and  $\theta_{im}PAA$  particle impact velocity and impact angle into different bins and analyzed the their relative frequency of these values. The average value of each bin is plotted as dots and the standard deviation is plotted as error bars in Fig. 9.

270

275

Figure 9. Relative frequencies of (a) particle adherence velocity, impact velocity, (b) particle adherence angle impact angle, and (c) the vertical adherence velocity vertical impact velocity of particles adhering to the on edge and surface.

The relative frequencies of  $v_{im}$  and  $\theta_{im}$ PAV/PAA represent the probabilities of particle adhesion on the cornice with a certain impact velocity or impact angle. It is shown in Fig. 9(a) that for edge particles, the adhesion probability exponentially decreases with the increasing value of  $v_{im}$ . Specifically, the relative frequency of the edge PAV $v_{im}$  of edge particles follows the exponential function  $f(v_{im}) = 4.3 + 30e^{-0.9v_{im}}$  ( $R^2 = 0.96$ ). This indicates that particles with lower impact velocities are more likely to adhere to the edge. The majority of edge particle adhesion occurs at velocities below 1.5 m/s, highlighting the critical role of low-velocity impacts in cornice growth. While the

relative frequency of  $v_{im}$  of surface particlesthe surface PAV follows the the Gaussian distribution function of  $f(v_{im}) = -2.8 + \frac{79}{3.4\sqrt{\pi/2}}e^{-2(\frac{v_{imp}-2.9}{3.4})^2}(R^2 = 0.91)$ , with values mainly concentrated at 3 m/s. This indicates that particles deposited on the surface normally have a higher impact velocity than the edge. This is because the snowpack on the surface is thicker than at the edge, as the cornice has a wedge shape. As a result, the surface can absorb more impact energy through longer force chains, allowing particles with higher impact velocities to adhere. In contrast, the thinner snowpack at the edge cannot effectively dissipate kinetic energy, so high-speed particle impacts often leads to erosion or fracture at the edge. The low number of particles adhering to the surface at low impact velocities can be attributed to the wind speed in the wind tunnel, which is set at 4 m/s. At this wind speed, the majority of particles are entrained and transported at higher velocities, leaving only a small fraction of particles moving at very low velocities near the cornice's surface.

280

285

290

295

300

305

As is shown in Fig. 9(b) that the frequency of  $\theta_{im}$  PAA of surface particles follows the Exponential distribution function  $f(\theta_{im}) = -0.8 + 80.6e^{-0.1v_{imp}}$  ( $R^2 = 0.97$ ), with values mainly concentrated below 17°. While the frequency of  $\theta_{im}$ PAA of edge particles follows the Gaussian distribution function  $f(\theta_{im}) = 7.1 + \frac{448.8}{45.7\sqrt{\pi/2}}e^{-2((v_{imp}-18.1)/45.7)^2}$  $(R^2 = 0.72)$ , with values distributed more uniformly in range. The average value of  $\theta_{im}$  PAA of surface particles is 13°, which is consistent with the previous experimental results of Nishimura and Hunt (2000), as is shown in the red dash in Fig. 9(b). Particles at the edge can adhere even at higher impact angles, as the edge provides a partially sheltered micro-environment where the local wind speed is lower. This reduction in wind velocity decreases the tendency for particle rebound and thereby increases the likelihood of adherence, even under conditions of increased impact angle. Furthermore, we combined the impact velocity and angle by analyzing the vertical impact velocity ( $v_{imv}$  =  $v_{im}sin\theta_{im}$ ) in Fig. 9(c). The relative frequency of  $v_{imv}$  for both surface particles and edge particles follow the exponential distribution, with surface particles  $f(v_{imv}) = 0.5e^{-v_{imv}/0.7}(R^2 = 0.96)$ , and edge particles  $f(v_{imv}) = 0.5e^{-v_{imv}/0.7}(R^2 = 0.96)$  $-0.1 + 0.5e^{-0.9v_{imv}}(R^2 = 0.95)$ . Particles adhere at low vertical impact velocities, whether on edges or surfaces. For both edge and surface particles, the threshold vertical impact velocity ranges from 2-2.5 m/s, with edge particles having a lower threshold velocity compared to surface particles. The differences in impact velocity and angle distributions between surface and edge are due to the fluid field differences caused by topographic changes. It is noted that the vertical impact velocity distributions of surface particles and edge particles are in the same trend, although the impact velocity and impact angle distributions of edge particles and surface particles are different. It indicates that particle adhesion can be affected by the vertical impact velocity, is mainly determined by the vertical impact velocity, and tThe differences in impact velocity and angle distributions between surface and edge are due to the fluid field differences caused by topographic changes. This is because only the vertical kinetic energy provided by the normal velocity can be used to overcome the adhesion energy barrier of the surface, whereas the tangential velocity cannot

assist the particle in detaching from the surface in the vertical direction (John, 1995).

Figure 10. Schematic diagram of force analysis of particles adhering to the edge.

**3.4 Static force analysis of adhering particles on the cornice edge**

310 To investigate the effect of dendricity on the article adherence at the edge, forces acting on a particle adhering to the edge are analyzed in this section.

Considering the differences in particle size distribution between the edge particles and surface particles, we conducted a static analysis of the particles at the edge. As shown in Fig. 10, a newly deposited particle i adheres to the foremost particle j at the edge of the cornice. Particle i is subjected to gravity  $F_g$ , the cohesive force  $F_c$  exerted by particle j, and the frictional force  $F_f$  at the contact surface. Due to the separation of flow, the wind velocity and surface shear stress near the edge of the cornice are close to zero (DeBonis, 2022; Shehadi, 2018), allowing the drag and lift forces acting on particle i to be neglected compared to other forces (Schmidt, 1980).

The force balance equations for particle i can be expressed as:

$$F_a \cos \alpha + F_c = F_s \tag{5}$$

320

315

$$F_q \sin \alpha \le F_f \tag{6}$$

$$F_f = \mu_f F_s \tag{7}$$

Here,  $F_g$  is the gravity force on particle i,  $F_c$  is the cohesive bond force, given by  $\pi x^2 \sigma$  (Szabo and Schneebeli, 2007), 325 where  $\sigma$  is the tensile strength at failure and x is the radius of the bond (blue shadowed area).  $F_s$  is the supporting force, and  $\alpha$  is the angle between the direction of gravity and cohesive force.  $R_i$  is the radius of particle i.

When snow particles adhere to the surface, both the gravity force  $F_g$  and the adhesive force  $F_c$  are in the vertical direction, resulting in an upward support force from the surface that maintains their stationary position. However, when snow particles adhere to the edge, the gravity force  $F_g$  and the adhesive force  $F_c$  are not in the same direction.

The component of the cohesive force in the direction of the gravity force is balanced by the support force generated by the edge, while the component of the gravity force perpendicular to the adhesive force needs to be balanced by friction force  $F_f$ . Once this component exceeds the frictional force, the particles will fall.

By substituting Eq. (5) and Eq. (7) into Eq. (6), we can derive the condition for particle i to maintain mechanical equilibrium if:

$$\quad \frac{F_c}{F_g} \ge \frac{\sin \alpha}{\mu_f} - \cos \alpha \tag{8}$$

To analyze the stability of particle i, overturning moments are calculated around point P (at the edge of the bond). The supporting force  $F_s$  and cohesive force  $F_c$  act through the center of particle i and operate on point P through the moment arm x. The gravity force  $F_g$  acts on point P through the moment arm  $Rsin(\alpha - arcsin(x/R))$ , where the angle between  $F_g$  and line  $\overline{OP}$  (distance from particle center to point P) is  $\beta = \alpha - arcsin(x/R)$ . The friction force  $F_f$  acts on point P through the moment arm Rcos(arcsin(x/R)). The condition for the particle to remain in equilibrium is when the total moment  $M_s = 0$ . Therefore:

$$(F_s - F_c)x + F_g R sin(\alpha - arcsin(\frac{x}{R})) - F_f R cos(arcsin(\frac{x}{R})) = 0$$

$$(9)$$

Substituting Eq. (5) and Eq. (7) into Eq. (9) yields:

$$\frac{F_c}{F_g} = \frac{x/R \cdot cos\alpha + sin(\alpha - arcsin(x/R)) - \mu_f cos\alpha cos(arcsin(x/R))}{\mu_f cos(arcsin(x/R))}$$
(10)

In which  $F_c$  is the cohesion force, which can be expressed as:

345

$$F_c = \pi x^2 \tau_b \tag{11}$$

where x is the contact radius of ice bridge, assumed here to vary linearly with particle radius  $x = \delta R$ , with ratio  $\delta = 0.1 - 0.25$  (Golubev and Frolov, 2001).  $\tau_b$  is the bond shear stress. While for non-spherical particles, particularly those with dendritic structures, the cohesion force is higher than that of spherical particles, due to the stronger geometrical interlocking between particles. Thus, dendricity should be considered in the calculation of the cohesion force for non-spherical particles. Here, we introduce a weighting parameter A into the cohesion force equation for dendritic particles, and its value will be derived later.

$$F_c = \pi x^2 \tau_b (1 + A(dd - 1)) \tag{12}$$

where dd is the dendricity value of non-spherical particles. For spherical particles, dendricity is a constant: dd =1
355 and cohesive force is only determined by contact radius,  $F_c = \pi x^2 \tau_b$ . For those particles that adhere on the edge, the average value of dendricity dd= 1.9, as is shown in Fig. 7.

**Figure 11.** Variation of  $\phi$  as a function of friction coefficient  $\mu_f$  and angle  $\alpha$

The gravitational force of the particle is:

$$F_q = 4/3\pi R^3 \rho_i g \tag{13}$$

By substituting Eq. (12) and (13) into the left side of Eq. (10), we obtained the expression for the ratio of  $F_c/F_g$ :

360
$$\frac{F_c}{F_g} = \frac{\pi x^2 \tau_b (1 + A(dd - 1))}{4/3\pi R^3 \rho_i g}$$
 (14)

Meanwhile, the right side of Eq. (10) can be defined as a function  $\phi$ , which is affected by the ratio  $\delta$ , angle  $\alpha$ , and friction coefficient  $\mu_f$ :

$$\phi = \delta cos\alpha + sin(\alpha - arcsin(\delta)) - \mu_f cos\alpha cos(arcsin(\delta)) \mu_f cos(arcsin(\delta))$$

$$\tag{15}$$

In which, the friction coefficient μf ranges from 0.2 to 0.7 (McClung and Schaerer, 2006), and the angle α varies

from 0 to 90° according to the experiment result. The resulting values of φ are illustrated in the contour plot shown in Fig. 11. Notably, φ decreases with increasing μf at all α, which indicates that higher friction reduces the need for a strong cohesive force to maintain stability. For a given μf, φ increases at larger angles, meaning that higher cohesion is required to keep the particle stable at the edge. When angle α=90°, the cohesion force is perpendicular to the gravity force; under this condition, the particle is most difficult to adhere to the edge. The bottom-right blank

areas correspond to the instability status for particles where adhesion doesn't happen.

Considering the most challenging condition for a particle to adhere to the edge occurs when the angle  $\alpha = 90^{\circ}$  and the friction coefficient  $\mu_f = 0.2$ . The corresponding  $\phi \approx 5$ . Therefore, by combining Eq. (10) (14) and (15), we can obtain:

$$\frac{F_c}{F_g} = \frac{\pi x^2 \tau_b (1 + A(dd - 1))}{4/3\pi R^3 \rho_i g} = 5 \tag{16}$$

390

395

400

Based on the experimental result shown in Fig. 6(b), the maximum radius of particles adhering on the edge is 325 um. Therefore, we have:

$$\frac{4\delta^2 \tau_b (1 + A(dd - 1))}{15\rho_i g} = 325 \times 10^{-6} \tag{17}$$

with  $\delta = 0.1$ ,  $\tau_b = 1$  kPa (Jamieson and Johnston, 1990), dd = 1.9, and  $\phi_{max} = 5$ , we can derive the value of parameter 380  $A \approx 0.07$ .

Therefore, in general, the cohesion force for edge particles, considering the shape effect, can be expressed as:

$$F_c = \pi x^2 \tau_b (1 + 0.07(dd - 1)) \tag{18}$$

and the threshold radius  $R_{th}$  for particles that can adhere to the edge can be estimated by:

$$R_{th} = \frac{4\delta^2 \tau_b (1 + 0.07(dd - 1))}{3\rho_i g \phi_{max}} \tag{19}$$

The threshold radius  $R_{th}$  linearly increases with the increasing dendricity, in different  $\delta$  values, as is shown in the Fig. 12. With the higher value of ratio  $\delta$ , it means a larger contact surface, therefore it allows larger particles adhering on the edge.  $\delta$  is mainly dependent on the air temperature and relative humidity (Colbeck, 1982). Furthermore, we divided the radius into different bins based on the dendricity values. By comparing the averaged radius of various dendricity, we found that the experimental result is in good agreement with the model-predicted results. The experimental data lie between the theoretical curves for  $\delta$ =0.15 and  $\delta$ =0.25. It can be concluded that the maximum particle size capable of adhering to the edge increases with increasing dendricity. Which means greater dendricity enables larger particles to remain attached, suggesting that the complexity of the particle shape helps counteract gravity.

Additionally, smaller snow particles experience lower gravity forces. Consequently, the ratio of  $F_c$  to  $F_g$  is higher, leading to increased values of  $\alpha$  and enhancing their tendency to adhere. This finding aligns with the results of this experiment, that smaller snow particles are more likely to adhere at the edges.

**4 Conclusions**

In this work, the micro-mechanism of snow cornice formation is investigated in a wind-tunnel experiment. Using a high-speed camera system, we recorded and analyzed particle trajectories surrounding the cornice, facilitated by a novel snow particle recognition program.

This study elucidates the micro-mechanisms underlying the initial formation of snow cornices through wind-tunnel experiments and high-speed particle tracking, facilitated by a novel snow particle recognition program. Direct observation and quantitative analysis reveal that near-surface saltation and creeping are the dominant mechanisms by which snow particles adhere to cornice structures. The vast majority of adhered particles settle on the upper surface

Figure 12. Experimental and theoretical threshold radius of edge-adhering particles as a function of dendricity for various  $\delta$  values.

405 via saltation, while only a smaller proportion is able to deposit or interlock at the front edge, often aided by their dendritic morphology.

Our experiments demonstrate that, despite local variations in particle size, velocity, and impact angle, the adhesion of snow particles—on both edge and surface—is governed primarily by the vertical impact velocity, with a clear adhesion threshold between 2–2.5 m/s. Edge-adhering particles are consistently smaller and exhibit higher dendricity compared to those on the surface, a trend explained by the force analysis showing that smaller and more branched particles possess a higher ratio of cohesive to gravitational forces  $(F_c/F_g)$ , enhancing their attachment stability especially under edge conditions.

410

415

Based on the experimental result, the static force on the particle adhering to the edge has been analyzed. The model quantitatively explains the preferential adhesion of smaller and dendritic snow particles observed in experiments. The cohesion force model incorporates dendricity, reflecting the enhanced contact area and thus higher cohesion for non-spherical, dendritic particles compared to spherical ones. By introducing dendricity into the expression for cohesion force, the model can predict the threshold radius for particle adhesion as a function of dendricity, particle properties, and environmental parameters. The model highlights the critical role of particle shape and microstructure, alongside environmental conditions, in determining edge adhesion in snow cornices.

Overall, this research provides new insights into the micro-mechanisms of snow cornice formation at the initial stage, emphasizing the critical roles of single moving particles, bridging the gap between particle-scale physics and cornice-scale hazare.

This experimental method provides direct observations that help bridge the gap between theoretical models and natural phenomenometric physics.

Although this study focuses on the initial stage of snow cornice formation at the micro-scale, the fundamental processes of adhesion of wind-transported snow particles are consistent across all scales, from laboratory conditions to natural environments. The direct experimental observations of single particles help bridge the gap between theoretical models and natural phenomena. Our experiments and findings enhance predictions of cornice growth and avalanche risk, with broader implications for understanding snow adhesion on both natural features and infrastructure, such as ice crevasse formation and wire icing.

Future studies should continue to explore the interactions between environmental factors and particle behavior to refine our understanding of snow cornice dynamics. Numerical simulations will be essential for a more comprehensive understanding of the coupling between the flow field and snow cornice dynamics, and investigate the effects of mountain morphology on cornice growth.

Author contributions. YHX designed the experiments. YHX and LG carried out the experiments. YHX performed the data analysis, and prepared the first draft. ML, BW, HJP and HN reviewed and edited the paper. HN and ML organized this study, contributed to its conceptualization, discussion, and finalized the paper.

Competing interests. The authors declare that they have no conflict of interest.

425

445

Data availability. The data archiving is underway and will be added after it is published. Data is available on request now.

Acknowledgements. The authors would like to thank the WSL Institute Snow and Avalanche Research SLF for making the experimental facilities available for our study. We thank Alec van Herwijnen for his invaluable experiment support. This work was supported by the Joint Funds of the National Natural Science Foundation of China (grant no.: U22A20564), the Young Scientists Fund of the National Natural Science Foundation of China (grant no.: 42406255), the Third Comprehensive Scientific Expedition and Research Program in Xinjiang (grant no.: 2022xjkk0101), China Postdoctoral Science Foundation (grant no.: 2024M751257), the Natural Science Foundation of Gansu Province (grant no.:24JRRA525) and the Swiss National Science Foundation (grant no.: 200020-179130). The data and code will be uploaded to the Dryad repository after the paper is published. Additionally, we are deeply grateful to the two anonymous reviewers and two scholars (Li Hongyi and Li Bailiang) for their insightful comments and constructive suggestions, which have greatly contributed to improving the quality and clarity of this paper.

---

## Author Response (AR2)

**Responses to the editor and each reviewer's comments**

Title: Snow Particle Motion in Process of Cornice Formation

ID: egusphere-2024-2458

Authors: Hongxiang Yu, Guang Li, Benjamin Walter, Jianping Huang, Ning Huang,

and Michael Lehning

Submitted to: The Cryosphere

The comments are in blue. Page and line numbers refer to the revised manuscript version with changes marked in italic.

**Responses to Editor:**

Dear authors,

I have received feedback from the two referees on your revised manuscript. Referee 2 is satisfied with the changes and has only pointed out a few minor corrections. However, Referee 1 remains much more critical, considering that the discussions still need improvement in several areas, particularly with regard to the study's relevance to real cornice formation and the quantitative evaluation of the proposed model against experimental data. I am also concerned that the referee mentions several discrepancies between your responses and the changes made to the paper. As your experimental results are highly novel and valuable to the snow community, I would like to give you another opportunity to address the referee's comments fully. Please consider all of their points and be sure to make the necessary amendments to your manuscript.

Best regards, Guillaume Chambon / TC Topical Editor

**Response:**

Dear Editor,

We would like to thank you for providing us with another opportunity to revise our manuscript. Regarding the discrepancies mentions by the referee, we would like to clarify that the discrepancy may have arisen because the referee reviewed an earlier version of response file for discussion ("Reply to RC2" uploaded on Feb 11, accessible at: <a href="https://egusphere.copernicus.org/preprints/2024/egusphere-2024-2458/egusphere-2024-2458/egusphere-2024-2458-AC4-supplement.pdf">https://egusphere.copernicus.org/preprints/2024/egusphere-2024-2458/egusphere-2024-2458-AC4-supplement.pdf</a>). However, an updated and more comprehensive response for review section was submitted on March 24, which can be found at: <a href="https://editor.copernicus.org/index.php?mdl=msovermd&jrl=25&lcm=oc73lcm74a&\_acm=get\_authors\_response\_file&\_ms=122384&id=2566119&salt=19379774151539860887">https://editor.copernicus.org/index.php?mdl=msovermd&jrl=25&lcm=oc73lcm74a&\_acm=get\_authors\_response\_file&\_ms=122384&id=2566119&salt=19379774151539860887</a>. We apologize for any confusion that may have arisen and greatly appreciate your understanding and continued suggestions.

We fully understand the importance of robust quantitative analysis, as highlighted by Referee 1, and appreciate the reviewer's persistent efforts to improve the quality of our study. In response, we have further quantified the dendricity and specific surface area (SSA) of snow particles, and included more comprehensive quantitative analysis in the revised results. The model section has also been expanded and directly linked to experimental findings.

**Based on the recommendations of **Referee 1**:**

- We have provided a more detailed discussion of the study's relevance to real cornice formation, emphasizing its implications in natural environments.
- We have conducted and included additional quantitative analysis on the dendricity and SSA of both surface and edge particles, with further evidence showing the tendency of dendritic particles to adhere on edges.
- We have revised the model, which now can better explain the phenomenon that dendritic particles are more prone to adhere on edge, and we have derived the expressions for the cohesion force and threshold radius for dendritic particle, directly based on the experimental data.

**Based on the recommendations of **Referee 2**:**

- All minor corrections and suggested revisions have been carefully implemented.

We hope these modifications address all the concerns raised by the reviewers. Please find a point-by-point reply to each of the comments below.

Thank you again for your constructive feedback and for the opportunity to further improve our work.

Sincerely,

Hongxiang Yu, on behalf of all authors

**Responses to Reviewer #1:**

**General comments**

**Comment 1:**

As noted in my previous review, the experimental procedures presented in this manuscript are highly informative and will undoubtedly be valuable for researchers interested in particle motion. The authors' efforts to investigate the growth mechanism of the thin snow plate extending leeward from the edge are commendable, and the overall experimental procedure is well documented. However, as also noted in my earlier review, I still feel that the study falls short of clarifying the growth mechanism of natural snow cornices. The differences between this miniature experiment and real snow cornices found in nature cannot be explained solely by variations in terrain size, successive precipitation, and duration. The authors' explanations remain unsatisfactory in this regard. I would like to emphasize once again that the authors appear to be examining fundamentally different phenomena. If the authors wish to assert that the thin plate observed in this study is relevant to understanding natural snow cornice formation, they should provide a clear scenario— ideally illustrated with schematic figures—that shows how the thin plate would develop step by step into a real cornice, specifying the key mechanisms involved at each stage. Before addressing the detailed points within the manuscript, I must also point out several discrepancies between the authors' responses and the content of the revised manuscript:

**Response:** Thank you very much for your thoughtful and constructive comments, and for the considerable time and effort you have devoted to reviewing our work. We truly appreciate your critical insights, which have been invaluable in helping us improve the manuscript.

As is shown in Fig. 1, the growth of a snow cornice can be divided into several stages. In the initial stage, a thin slab forms at the mountain edge (highlighted in red in Fig. 1), primarily as a result of wind-driven accumulation and the adhesion of newly precipitated snow particles. In the subsequent stage, continued deposition from drifting and precipitation leads to further development of the cornice, which gradually increases in both length and height. As the cornice grows larger, gravitational forces cause the overhanging volume to bend downward. Eventually, when the volume of the cornice becomes excessive, or more specifically, when the shear stress at the base exceeds the strength of the snow cornice will break off.

Figure 1. Schematic diagram of different stages in snow cornice growth.

The thin slab observed in our study corresponds to this initial stage of cornice formation, during which micro-scale adhesion mechanisms dominate. Clarifying these early mechanisms, as depicted in the schematic, provides important insight into how small-scale structures evolve step by step into much larger overhanging cornices found in natural alpine environments. We point out, however, that the principle mechanism should not be different between our small-scale model and the large-scale field.

Due to the limitations of field observation and equipment, it is difficult to observe the particle movement around a real-sized cornice on the mountain ridge. However, understanding the mechanism of snow cornice formation is still essential for avalanche prediction and simulation. Whether in the field or in the wind tunnel, the fundamental micro-scale processes-such as particle movement and adhesion-at the mountain top (top (plateau or ridge) are expected to be consistent. Therefore, we carried out a wind tunnel experiment specifically to observe the particle trajectories during the initial stage of cornice formation, focusing on when a cornice first appears as a thin snow plate at a ridge or plateau edge. The motivation for our study is precisely to address this observation gap by focusing on the very earliest stages of cornice formation-where a cornice begins as a thin snow plate at a ridge or plateau edge.

Although there are certainly differences between our small-scale laboratory setup and complex, evolving conditions in nature (including terrain size, weather variability, and duration of precipitation), we believe the fundamental physics governing particle motion, entrainment, and adhesion at the initial stage are directly relevant to the earliest moments of natural cornice development. In other words, the micro-scale processes we observe-how wind -driven snow particles move and adhere at the edge-must also

operate during the formation of real cornices, even if the later stages involve further complexity.

In the revised manuscript, we will more explicitly outline the key mechanisms that are shared between the model system and full-size cornice formation and clarify the limitations and scope of our findings. We have revised the abstract in lines 2-3 to: "Despite numerous observations and experimental simulations on their formation process, the microscopic mechanism of their initial stage of formation remains unclear." Lines 13-16: "Overall, this research reveals the microdynamics underlying initial cornice growth, providing a theoretical basis for avalanche modeling and infrastructure protection in alpine environments, as well as offering a methodological and mechanical framework for studying snow and ice adhesion in both natural and engineered systems"

In the Introduction, we have added Figure 1 that illustrates the various stages in cornice growth, and the explanation of the whole process of cornice growth in lines 29-47: "The growth of a snow cornice can be divided into several stages (Montagne, 1980; Vogel et al., 2012; Eckerstorfer et al., 2013). In the initial stage, a thin slab forms at the mountain edge (highlighted in red in Fig. 1), mainly by adhesion of wind-transported snow particles. When more snow accumulates on the relatively flat surface above the edge, it can gradually be conveyed toward the slab tip—especially via windtransported particles—thereby increasing the thickness at the cornice root. This sustained supply of snow from the platform region plays a key role in the transformation of a small slab into a fully developed cornice in nature. In the subsequent stage, repeated deposition from intermittent drifting and precipitation successively adds new layers of snow to the cornice. This layer-by-layer accumulation is accompanied by a gradual increase in both length and thickness of the cornice. As the cornice grows larger, the overhanging mass of snow is increasingly influenced by gravitational forces, which may cause it to bend downward (shown in the white dashed line in Fig. 1) and promote internal compaction near the edge. Eventually, when the cornice becomes too large and shear stress exceeds a critical threshold, it breaks off and collapses. The evolution of a wedge-shaped cornice—from initial slab formation to subsequent snow accumulation on the flat surface—has been experimentally investigated in our previous work (Yu et al., 2022), with particular focus on the relationship between cornice growth rate and air mass transport. However, the specific mechanisms governing the very initial stage, that is, how airborne snow particles first adhere and accumulate to form the incipient slab at the edge, remain unexplored.

Previous field research mainly focused on the morphology variation due to limitations of observation equipment (Vogel et al., 2012; Eckerstorfer et al., 2013; van Herwijnen and Fierz, 2014; Hancock et al., 2020), for observing how particles adhering to mountain edges are hardly realized."

In Conclusions, we have added lines 376-379: "Although this study focuses on the initial stage of snow cornice formation at the micro-scale, the fundamental processes of adhesion of wind-transported snow particles are consistent across all scales, from laboratory conditions to natural environments. The direct experimental observations of single particles help bridge the gap between theoretical models and natural phenomena."

Besides, we would also like to mention that there may be some confusion due to different versions of our response documents being available in the discussion system. It is possible that the latest version of our replies has not been reviewed. We sincerely apologize for any resulting misunderstanding and will address each of your comments point by point below to ensure that all concerns are fully resolved.

Discrepancies Between Author Replies and Manuscript Contact Collision Explanation: The authors state in their reply:

"For clarity, we will add the following sentences before introducing the maximum compression displacement of particles in Section 3.2: 'During contact collision, snow particles will be compressed and deformed, undergoing plastic deformation and brittle failure (Wang et al., 2020).'" However, neither the text nor the reference appears in the manuscript.

Sintering Force Description: In the reply, the authors note:

"Sentences in lines 26–27 will be deleted, and we will add: 'In which, Fb is the sintering force, calculated as the product of the ice tensile strength and the contact surface area (Szabo and Schneebeli, 2007). Sintering begins upon particle deposition and plays a crucial role in stabilizing and preserving the cornice structure." Yet, this explanation and the notation for Fb are not included in the manuscript. The same applies to subsequent mentions of sintering effects—the manuscript still lacks these explanations.

Order of Magnitude Description: The authors replied:

"The size of snow particles follows a distribution function, and different sized particles experience different magnitudes of force. Therefore, we present orders of magnitude rather than exact values." Yet the manuscript contains no mention or expression of

"order of magnitude." Missing Reference: The following reference, cited in the authors' reply, does not appear in the manuscript: Enliang Wang et al., 2021, Cold Regions Science and Technology, 182, 103215.

**Response:** Thank you very much for your comments and careful review. We would like to clarify that the discrepancy may have arisen because the response file reviewed was an earlier version ("Reply to RC2" uploaded on Feb 11, accessible at: <a href="https://egusphere.copernicus.org/preprints/2024/egusphere-2024-2458/egusphere-2024-2458/egusphere-2024-2458-AC4-supplement.pdf">https://egusphere.copernicus.org/preprints/2024/egusphere-2024-2458/egusphere-2024-2458-AC4-supplement.pdf</a>). However, an updated and more comprehensive response for review section was submitted on March 24, which can be found at: <a href="https://editor.copernicus.org/index.php?mdl=msovermd&jrl=25&lcm=oc73lcm74a&\_acm=get\_authors\_response\_file&\_ms=122384&id=2566119&salt=19379774151539860887">https://editor.copernicus.org/index.php?mdl=msovermd&jrl=25&lcm=oc73lcm74a&\_acm=get\_authors\_response\_file&\_ms=122384&id=2566119&salt=19379774151539860887</a>.

We kindly invite you to refer to this updated response for the new revisions. We apologize for any confusion that may have arisen and greatly appreciate your understanding and continued suggestions.

**Specific Comments**

**Comment 1:**

**Figure S1:**

It seems the authors intended to show that shear stress becomes negligible, but it is difficult to interpret what is being presented. More detailed explanations are required, particularly regarding the reason for plotting two data series between 0.7 and 2 (x/H). Since computer simulations of airflow were conducted, I strongly recommend including a representative airflow pattern around the edge. This would be extremely helpful in explaining the particle movements discussed later.

**Response:** Thank you for your valuable suggestion regarding the inclusion of numerical simulation results and the airflow pattern around the edge. In fact, conducting such simulations has required considerable effort and has been comprehensively addressed in our recent publication (Yu et al., 2025), which has been accepted and will be available online soon. To avoid repetition and ensure the focus of the present manuscript, we have not included those simulation details here, but we have cited relevant numerical simulation results from previous studies on the flow around steps to support our discussion regarding the airflow structure near the edge in the last response: "Moreover, from the previous simulation and experiment studies on the fluid field of backward-facing step (DeBonis, 2022; Shehadi and Edmond. 2018), as is shown in Fig.R1, the skin friction coefficient  $C_f = \tau_w/0.5 \rho U_{ref}^2$  drops to a very small value at the edge (x/H=0). In which,  $\tau_w$  is the wall shear stress,  $\rho$  is the fluid density,  $U_{ref}$  is the freestream velocity. It can be concluded that the drop in  $C_f$  at the edge of the backwardfacing step is caused by boundary layer separation due to sudden geometric discontinuity. This separation creates a recirculating region with low or negative wall shear stress, leading to a significant reduction in  $C_f$ . Similar with our case, the edge of a cornice is the flow separation point, with a wall shear stress approximately equal to zero. Therefore, the drag force of particles on the edge can be ignored."

Figure R1. Skin friction coefficient (DeBonis, 2022)

Correspondingly, we have added the relevant description in lines 206-207: "Such particles therefore preferentially deposit on the edges—where the wind speed is near zero and accompanied by a reflux vortex (DeBonis, 2022)."

And lines 278-280: "Due to the separation of flow, the wind velocity and surface shear stress near the edge of the cornice are close to zero (DeBonis, 2022; Shehadi, 2018), allowing the drag and lift forces acting on particle i to be neglected compared to other forces (Schmidt, 1980)."

**References:**

Schlichting, H., & Gersten, K. (2016). Boundary-Layer Theory. Springer.

DeBonis, J.R. A Large-Eddy Simulation Of Turbulent Flow Over A Backward Facing Step. In Proceedings of the AIAA SCITECH2022 Forum, San Diego, CA, USA, 29 December 2022; p. 0337

Shehadi, Edmond. (2018). Large Eddy Simulation of Turbulent Flow over a Backward-Facing Step. 10.13140/RG.2.2.17703.24480.

Yu, H., Li, G., Jafari, M., Lehning, M., Huang, J., & Huang, N. (2025). Effect of snow cornice formation on wind fields and snow deposition: Insights from numerical simulations. Journal of Geophysical Research: Atmospheres, 130, e2024JD042702. https://doi.org/10.1029/2024JD042702

**Comment 2:**

Line 58 & Figure 1: The description of the wind tunnel setup is inadequate. Please include key specifications such as the length and size of the working section

**Response:** Thank you for your comment. The specifications of the wind tunnel, including the length and size of the working section, were described in detail in our previous work (Yu et al., 2022). To provide a clear description, we have added the main parameters and cited the previous study. The sentences have been revised in lines 73-76: "The experiment setup is shown in Fig. 2. The working section of the wind tunnel is 1 m in length, with a cross-section area of 0.2 m(width)×0.5 m(height), and has been successfully used for several drifting snow experiments (Wahl et al., 2024; Walter et al., 2024; Yu et al., 2022; Sommer et al., 2017, 2018a). Further details of the wind tunnel can be found in Yu et al. (2022)."

**Comment 3:**

**Line 67:**

The authors mention preliminary tests comparing two types of snow and concluding that dendritic snow is more suitable. Were the particle sizes of both types the same? Since the manuscript later emphasizes particle size as a key factor, it's important to clarify this point.

**Response:** Thank you for your comment. Regarding the comparison between the two types of snow, we used artificial fresh snow (dendritic) and aged snow that stored for 2-3 days (rounded). Due to snow metamorphism, it was impossible to keep the particle sizes identical. Aged snow tends to have smaller, rounded particles, while fresh snow forms larger, dendritic grains.

We acknowledge that this makes it impossible to strictly separate the effects of particle size and grain shape. Nonetheless, our experiment aimed to reflect realistic changes that occur in snow exposed to aging. Our results indicate that fresh, dendritic particles are considerably more conductive to cornice formation. Thus, we used the dendritic particle in the experiments.

We have clarified this point in lines 86 to 90: "Fresh snow particles, characterized by their highly dendritic shapes, were compared to decomposed snow particles, which are characterized by small, rounded shapes after being stored for several days at a constant temperature of  $T_{air} = -10$  °C. The results show that both types of snow particles are capable of forming a snow cornice. However, fresh snow particles exhibit a significantly higher propensity for cornice formation and are much easier to consolidate into a stable structure. Therefore, fresh snow particles were used in the subsequent experiments."

**Comment 4:**

Line 80: The phrase "for 4–5 s during cornice growth" would be clearer if supplemented with figures showing the time evolution of the thin plate's length and thickness.

**Response:** Thank you for this comment. The cornice evolution in length and thickness has been already reported in the previous work (Yu et al., 2022). In this work, only short time periods (4-5 s) were selected in each experiment to capture particle movement. In our experiments, a total of 18 cases were conducted, each lasting 4-5 seconds, due to memory limitations of our high-speed camera. These short-duration cases were distributed over the entire cornice growth process.

To address the reviewer's suggestion, we have clarified this point in the revised manuscript, lines 100-102: "A total 18 cases were conducted during the cornice growth, with each case lasting 4-5 seconds and yielding 12455 images to record the particle trajectories. The duration of each case was limited by the camera memory. A sequence of the different growth steps is illustrated in Fig. 3 (a) of Yu et al. (2022)."

Figure. 3(a) Cornice profiles in the growth process (Yu et al., 2022)

**Reference:**

Yu, H., Li, G., Walter, B., Lehning, M., Zhang, J., and Huang, N.: Wind conditions for snow cornice formation in a wind tunnel, The Cryosphere, 17, 639–651, https://doi.org/10.5194/tc-17-639-2023, 2023.

**Comment 5:**

**Line 109:**

Since fresh dendritic snow tends to orient perpendicular to the wind to maximize resistance, particle size might be underestimated when viewed from the side. This effect would be negligible for rounded particles but should be considered for dendritic ones.

**Response:** Thank you for raising this important point regarding the potential influence of dendritic particle orientation on measured particle size. Although fresh snow tends to orient perpendicular to the wind to maximize resistance, this generally occurs for freely suspended particles moving steadily in the air (without colliding with the surface). In our experiment, most particles collided with the surface and exhibited rotation, resulting in varying projected areas across consecutive frames. To account for this, we

recorded multiple frames as each particle rotated and averaged the projected areas to estimate particle size. This approach effectively minimized the influence of particle orientation on the measured size. Therefore, we consider the effect of dendritic particle orientation on particle size estimation is considered negligible in our results. This clarification has been added to the revised manuscript, lines 133 - 135: "For dendritic particles, the projected area  $A_p$  and perimeter P were averaged across these frames for each particle, which effectively minimizes the influence of particle orientation on the calculated size."

**Comment 6:**

**Line 137:**

Are any particles ejected by collisions with saltating particles? If so, they might move slowly and contribute to edge growth. Please clarify this point.

**Response:** Thank you for this insightful comment. Yes, we have observed that particles ejected by collisions with saltating particles can indeed contribute to edge growth, and we have included these into creeping particles in our analysis. These ejected particles typically move as creeping particles. We have added a description of this particle type in lines 214-218: "Creeping particles (Fig. 8(a)), which account for about 14% of the observed particles, represent the minority of larger-sized adhered particles, and they typically move slowly. These particles are mainly entrained from the surface under the ejection of other particles. Most of them retain on the cornice surface, and only a small fraction—with elongated dendrites—are able to interlock and remain adhered at the edge (Fig. 8(b))."

Besides, those observed saltating particles might also come from the ejection particles, therefore, we added the description in lines 219-220: "In addition, saltating particles observed near the cornice originate either from airborne trajectories or from ejection off the surface."

**Comment 7:**

Line 143: The statement "smaller particles, with better followability with the wind" suggests that their impact speed should be higher, which conflicts with Figure 7. Additionally, larger dendritic particles typically have more branches, potentially increasing their likelihood of being trapped at the edge. I recommend introducing quantitative parameters such as specific surface area to strengthen this discussion.

**Response:** Thank you for your insightful comment. We appreciate your suggestions regarding the relationship between particle size, followability with the wind, and impact velocity, as well as the value of introducing quantitative parameters such as specific surface area (SSA) to strengthen our discussion.

To further address your suggestions for adding quantitative morphological parameters, we have included descriptions of how dendricity and specific surface area are calculated in the revised manuscript.

**2.1 Particle recognition**

Lines 124-135:

"During image processing, the particle's area and perimeter are saved in a numerical matrix in binarized format. Thus, the particle's projected area  $A_p$  can be estimated by calculating the sum of all the connected component labels, and the particle's perimeter P is the sum of all the boundary labels. The particle's equivalent diameter is calculated based on the value of its projected area:  $d_e = \sqrt{4A_p/\pi}$ . The dendricity of each particle was quantified using the method proposed by (Bartlett et al., 2008), based on two-dimensional image analysis. Specifically, dendricity was calculated as  $De = \frac{P^2}{4\pi A_p}$ . The specific surface area (SSA) of particles was estimated from two-dimensional images by measuring the perimeters and areas of particles, following the stereological approach proposed by (Ren et al., 2021). According to this method, SSA is calculated as  $SSA = \frac{4\bar{P}_1}{\pi A_1}$ . This approach enables the statistical analysis of SSA distribution based on 2D image data, and its validity and limitations have been demonstrated in comparison with three-dimensional and conventional measurement methods. For dendritic particles, the projected area  $A_p$  and perimeter P were averaged across these frames for each particle, which effectively minimizes the influence of particle orientation on the calculated size."

Additionally, in the Section 3.1, we conducted a comparative analysis of the dendricity and SSA of edge and surface particles.

**3.1 Particle size and shape**

Lines 186-197:

"In addition to particle size, dendricity and specific surface area (SSA) are important indicators of particle morphology and surface characteristics. As illustrated in Fig. 7(a), the average dendricity of edge particles is 1.9, higher than that of surface 1.4. Meanwhile, the distribution range of the edge particles (1 to 4.7) is broader than that of the surface particles (1 to 3.1). These results indicate that the edge particles have more fragmented or branched morphologies, while particles on the surface are generally more regular and compact.

However, the SSA values of edge particles and surface particles are similar, with the average value of 20 mm²/mm³, as is shown in Fig. 7(b). This similarity arises because both edge and surface particles originate from the same snow source. Therefore, dendricity is a more critical factor in determining whether a particle can adhere to an edge or the surface. In particular, edge particles with high dendricity have more contact points with neighboring particles on the cornice, which may lead to a greater cohesion force Fc—the force counteracts the gravity force Fg and allows the edge particle to adhere. In contrast, surface particles may experience less cohesion force, and their

gravity acts in the same direction as the cohesion force, making gravity either irrelevant or even beneficial for particle adherence.

Figure 7. Frequency distributions of (a) dendricity and (b) specific surface area (SSA) for particles adhering to the edge and surface.

To better explain the reason for larger particles sticking to the surface preferentially and smaller particles to the edge, we have added a paragraph in lines 198-210: "Combining the analysis of particle size, dendricity, and SSA distribution, we find that smaller particles and dendritic particles are more prone to adhere to the edge, while larger and more spherical particles tend to deposit on the surface. This phenomenon is closely related to the aerodynamic behavior of particles in the air. Specifically, the pattern of particle deposition is primarily governed by the Stokes number (Comola et al., 2019), a dimensionless parameter that compares the inertial response time (particle relaxation time) of a particle to the characteristic time scale of the fluid flow. In general, particles with smaller sizes, as well as large particles with irregular shapes, tend to have lower relaxation times than spherical particles of the same size (Loth, 2008). As a result, for such small particles and large, highly dendritic ones, viscous forces dominate over inertia, enabling the particles to quickly respond to changes in local fluid velocity and closely follow the streamlines. Such particles therefore preferentially deposit on the edges—where the wind speed is near zero and accompanied by a reflux vortex (DeBonis, 2022). In contrast, for spherical particles, especially those with larger size, inertial forces become more significant relative to viscous drag, making these particles less responsive to fluid velocity changes and more likely to deposit on the main surfaces—which are generally characterized by a stable boundary layer and low turbulence."

**Comment 8:**

**Lines 158-164:**

These descriptions are speculative and qualitative. As noted earlier, it would greatly improve the discussion to include airflow patterns around the edge. If direct

measurements using a hot-wire anemometer were not conducted, simulated streamlines or vortex separations would be very helpful.

**Response:** Thank you for your valuable suggestions. We agree that inclusion of simulated streamlines or vortex separation would enhance the discussion of airflow patterns around the cornice edge. Conducting detailed flow simulations for the given geometry has been an effort in its own and covered in our recent publication (Yu et al., 2025), which paper has been accepted and will be online soon.

In this revision, we have clarified that the location of cornice growth corresponds to a step-flow, and we have referenced relevant numerical simulation results from previous research to support our interpretations. The revised paragraph is shown in lines 206 to 207: "Such particles therefore preferentially deposit on the edges—where the wind speed is near zero and accompanied by a reflux vortex (DeBonis, 2022)."

Besides, in the conclusion, we have added a sentence in lines 383-385: "Numerical simulations will be essential for a more comprehensive understanding of the coupling between the flow field and snow cornice dynamics, and investigate the effects of mountain morphology on cornice growth."

**Reference:**

DeBonis, J.R. A Large-Eddy Simulation Of Turbulent Flow Over A Backward Facing Step. In Proceedings of the AIAA SCITECH2022 Forum, San Diego, CA, USA, 29 December 2022; p. 0337

Yu, H., Li, G., Jafari, M., Lehning, M., Huang, J., & Huang, N. (2025). Effect of snow cornice formation on wind fields and snow deposition: Insights from numerical simulations. Journal of Geophysical Research: Atmospheres, 130, e2024JD042702. https://doi.org/10.1029/2024JD042702

**Comment 9:**

Lines 169–170:

There's an inconsistency here. The authors state that creeping particles (about 14%) are larger and settle near the front end of the cornice, but later conclude that smaller particles are more likely to adhere at the edge. This contradiction needs to be resolved.

**Response:** Thank you for your attention to this point, which may cause misunderstanding for readers. We would like to clarify that the creeping particles refer to larger particles, which only represent a small proportion of the particles able to adhere to the edge. When we state smaller particles are more likely to adhere at the edge, it is compared to all the particles. To avoid confusion, we have revised this sentence as lines 214-216: "Creeping particles (Fig. 8(a)), which account for about 14% of the observed particles, represent the minority of larger-sized adhered particles, and they typically move slowly."

**Comment 10:**

Lines 176–181 & Figures 6–7:

How many particle trajectories were analyzed to derive the appearance ratios in Figure 6? Is the sample size sufficient for quantitative conclusions? Also, how were impact speeds and angles in Figure 5 determined under the complicated particle movement? Were these captured at time step 4? If so, negative angles should appear in Figure 7 as well.

**Response:** Thank you for helping us improve the clarity and rigor of presentation. We have clarified this question regarding the particle number in Comment 7 in the last response uploaded on Mar 24th. We have increased 383 surface particles and 121 edge particles from the experimental data. Therefore, there are 655 collision particles in total. We used Kolmogorov-Smirnov test to evaluate whether the observed data significantly deviate from the fitting function. The p-values for both surface and edge particles being higher than 0.05 suggest that the sample size is sufficient to support the analysis results.

The impact speed and angle are both defined as relative to the horizontal axis, as is shown in Figure below:

Figure. Schematic figure of impact speed and angle definition

Back-moving particles indeed result in negative angles, which are not included in the PAV in Fig. 9. Firstly, the number of particles adhering via back-moving is extremely small and does not significantly affect the overall statistical trends. Secondly, since these recaptured particles tend to attach at positions deviating from the horizontal surface of the cornice, it is impossible to define their impact angles in a manner consistent with the other particles. Therefore, the collision parameters of these few particles were not included in the statistical analysis presented in this study. Although this model of back-ward moving adhesion was not included in the main statistical analysis, it is worth noting that such process may be caused by the possible electric field and fluid field, which needs further investigation in future studies.

We have added the clarification in lines 227-231: "Here, we define the  $v_{im}$  as the impact velocity of the particle  $\theta_{im}$  is defined as the angle of particle incidence on the horizontal cornice surface, ranging from  $0^{\circ}$  (parallel to the surface) to  $90^{\circ}$  (perpendicular to the surface), and this angle measures how steeply a particle approaches the surface or edge before sticking. Only particles impacting the cornice surface from above (incidence angle 0- $90^{\circ}$ ) are considered, while particles with trajectories suggesting backward-moving ( $\theta_{im} > 90^{\circ}$ ) are excluded."

**Comment 11:**

Moreover, Figure 7's horizontal axis is labeled "impact velocity or angle," while the figure caption refers to "particle adherence velocity or angle." This should be unified for clarity.

**Response:** Thank you for pointing this out. We have revised the caption of Figure 9 (Figure 7 in the previous version) and unified the terminology throughout the manuscript.

**Figure 9**. Relative frequencies of (a) impact velocity, (b) impact angle, and (c) vertical impact velocity of particles adhering to the edge and surface.

**Comment 12:**

**Lines 182–218 & Figure 7:**

The explanation for why higher-speed particles adhere at certain positions but not near the edge remains unclear. Generally, high-speed, low-angle impacts would result in rebound. Further, do you have evidence that the vertical component of velocity predominantly dictates particle behavior? A clearer discussion is needed

**Response:** Thank you very much for this comment. We agree that the explanation for particle adhesion in different regions requires clarification. In our experiments, we focused on the particles that adhered to the surface after impact, and did not specifically analyze those that rebounded, as the rebound has been extensively studied in the previous works (e.g., Sugiura and Maeno, 2000).

Generally, on the thicker surface region, the snowpack can absorb more impact energy via longer force chains, allowing some particles (especially those with sufficient vertical velocity component) to adhere. In contrast, the snowpack thickness of edge is thin and cannot effectively dissipate the kinetic energy through force chains. The impact of high-speed particles will cause the edge break. We have added the descriptions in lines 245-248: "This is because the snowpack on the surface is thicker than at the edge, as the cornice has a wedge shape. As a result, the surface can absorb more impact energy through longer force chains, allowing particles with higher impact velocities to adhere. In contrast, the thinner snowpack at the edge cannot effectively dissipate kinetic energy, so high-speed particle impacts often leads to erosion or fracture at the edge."

Concerning the role of the vertical velocity component, the conclusion that whether a particle can adhere or rebound on the surface is determined by the vertical velocity is consistent with the well-established knowledge. We have added the discussion in lines 269-271: "This is because only the vertical kinetic energy provided by the normal velocity can be used to overcome the adhesion energy barrier of the surface, whereas the tangential velocity cannot assist the particle in detaching from the surface in the vertical direction (John, 1995)"

**References:**

Sugiura, K., Maeno, N. Wind-Tunnel Measurements Of Restitution Coefficients And Ejection Number Of Snow Particles In Drifting Snow: Determination Of Splash Functions. *Boundary-Layer Meteorology* **95**, 123–143 (2000).

Walter, John. Particle-Surface Interactions: Charge Transfer, Energy Loss, Resuspension, and Deagglomeration, Aerosol Science and Technology, 23:1, 2-24, (1995).

**Comment 13:**

Lines 218–266: The model describing forces between particles at the edge feels redundant, as all subsequent discussions are qualitative. As noted previously, quantitative validation using experimental snow and environmental data is essential. The claim that smaller dendritic particles adhere more readily due to a higher Fc/Fg ratio can be easily speculated without introducing the model.

**Response:** Thank you for your valuable comments. We have added more quantitative results in the model and directly using experimental snow data. We have further added detailed explanations for the phenomena that smaller dendritic particles adhere more readily. Furthermore, based on the mode, we have derived the cohesion force of dendritic particles, and derived the threshold radius of particles that can adhere on the edge.

**3.4 Static force analysis of adhering particles on the cornice edge**

Lines 307 to 355:

$$\frac{F_c}{F_g} = \frac{x}{R} \cos \alpha + \sin(\alpha - \arcsin(\frac{x}{R})) - \mu_f \cos \alpha \cos(\arcsin(\frac{x}{R})) \mu_f \cos(\arcsin(\frac{x}{R}))$$
(10)

*In which*  $F_c$  *is the cohesion force, which can be expressed as:*

$$F_c = \pi x^2 \tau_b \tag{11}$$

where x is the contact radius of ice bridge, assumed here to vary linearly with particle radius  $x = \delta R$ , with ratio  $\delta = 0.1$ -0.25(Golubev and Frolov, 2001).  $\tau_b$  is the bond shear stress. While for non-spherical particles, particularly those with dendritic structures, the cohesion force is higher than that of spherical particles, due to the stronger geometrical interlocking between particles. Thus, dendricity should be considered in the calculation of the cohesion force for non-spherical particles. Here, we introduce a weighting parameter A into the cohesion force equation for dendritic particles, and its value will be derived later.

$$F_c = \pi x^2 \tau_h (1 + A(De - 1)) \tag{12}$$

where De is the dendricity value of non-spherical particles. For spherical particles, dendricity is a constant: De =1 and cohesive force is only determined by the contact radius,  $F_c = \pi x^2 \tau_b$ . For those particles adhere on the edge, the average value of dendricity De = 1.9, as is shown in Fig. 7.

The gravitational force of the particle is:

$$F_g = \frac{4}{3}\pi R^3 \rho_i g \tag{13}$$

By substituting Eq. (12) and (13) into the left side of Eq. (10), we obtained the expression for the ratio of  $F_c/F_g$ :

$$\frac{F_c}{F_g} = \frac{\pi x^2 \tau_b (1 + A(De - 1))}{4/3\pi R^3 \rho_{ig}} \tag{14}$$

Meanwhile, the right side of Eq. (10) can be defined as a function  $\varphi$ , which is affected by the ratio  $\delta$ , angle  $\alpha$ , and friction coefficient  $\mu_f$ :

$$\varphi = \delta \cos \alpha + \sin(\alpha - \arcsin(\delta)) - \mu_f \cos \alpha \cos (\arcsin(\delta)) \mu_f \cos (\arcsin(\delta))$$
(15)

In which the friction coefficient  $\mu_f$  ranges from 0.2 to 0.7 (McClung and Schaerer, 2006), and the angle  $\alpha$  varies from 0 to 90° according to the experiment result. The resulting values of  $\phi$  are illustrated in the contour plot shown in Fig. 11. Notably,  $\phi$  decreases with increasing  $\mu_f$  at all  $\alpha$ , which indicates that higher friction reduces the need for a strong cohesive force to maintain stability. For a given  $\mu_f$ ,  $\phi$  increases at larger angles, meaning that higher cohesion is required to keep the particle stable at the edge. When angle  $\alpha$ =90°, the cohesion force is perpendicular to the gravity force; under this condition, the particle is most difficult to adhere to the edge. The bottom-right blank areas correspond to the instability status for particles where adhesion doesn't happen.

Considering the most challenging condition for a particle to adhere to the edge occurs when the angle  $\alpha = 90^{\circ}$  and the friction coefficient  $\mu_f = 0.2$ . The corresponding  $\phi \approx 5$ . Therefore, by combining Eq. (10), (14) and (15), we can obtain:

$$\frac{F_c}{F_g} = \frac{\pi x^2 \tau_b (1 + A(De - 1))}{4/3\pi R^3 \rho_i g} = 5$$
 (16)

Based on the experimental results shown in Fig. 6(b), the maximum radius of particles adhering on edge is 325 um. Therefore, we have:

$$\frac{4\delta^2 \tau_b (1 + A(De - 1))}{15\rho_i g} = 325 \times 10^{-6} \tag{17}$$

with  $\delta = 0.1$ ,  $\tau_b = 1$  kPa (Jamieson and Johnston, 1990), De = 1.9, and  $\varphi_{max} = 5$ , we can derive the value of parameter  $A \approx 0.07$ .

**Figure 11.** Variation of  $\phi$  as a function of friction coefficient  $\mu_f$  and angle  $\alpha$ .

Therefore, in general, the cohesion force for edge particles, considering the shape effects, can be expressed as:

$$F_c = \pi x^2 \tau_b (1 + 0.07(De - 1)) \tag{18}$$

and the threshold radius  $R_{th}$  for particles that can adhere to the edge can be estimated by:

$$R_{th} = \frac{4\delta^2 \tau_b (1 + 0.07(De - 1))}{3\rho_{tg} \varphi_{max}}$$
 (19)

The threshold radius  $R_{th}$  linearly increases with the increasing dendricity, in different  $\delta$  values, as is shown in Fig. 12. With the higher value of ratio  $\delta$ , it means a larger contact surface, therefore it allows larger particles adhering to the edge.  $\delta$  is mainly dependent on the air temperature and relative humidity (Colbeck, 1982). Furthermore, we divided the radius into different bins based on the dendricity values. By comparing the averaged radius of various dendricity, we found that the experimental result is in good agreement with the model-predicted results. The experimental data lie between the theoretical curves for  $\delta$ =0.15 and  $\delta$ =0.25. It can be concluded that the maximum particle size capable of adhering to the edge increases with increasing dendricity. Which means greater dendricity enables larger particles to remain attached, suggesting that the complexity of the particle shape helps counteract gravity.

**Figure 12.** Experimental and theoretical threshold radius of edge-adhering particles as a function of dendricity for various  $\delta$  values.

**References:**

McClung, D., & Schaerer, P. (2006). The Avalanche Handbook (3rd ed.), Table 9.1.

Colbeck, S. C. (1982), An overview of seasonal snow metamorphism, *Rev. Geophys.*, 20(1), 45–61, doi:10.1029/RG020i001p00045.

Jamieson JB, Johnston CD. In-Situ Tensile Tests of Snow-Pack Layers. *Journal of Glaciology*. 1990;36(122):102-106. doi:10.3189/S002214300000561X

Golubev, V. N. and Frolov, A.: On the correlation between tensile strength and stress wave velocities of dry coherent snow based on its structural model, Annals of Glaciology, 32, 70 – 74, https://doi.org/10.3189/172756401781819562, 2001.

**Comment 14:**

Given the complexity of dendritic particle shapes, factors such as branching and interlocking likely have greater influence. Additionally, the dependence of Fc/Fg on particle size is unclear. If the authors wish to explore particle size effects more rigorously, I strongly recommend analyzing the cornice structure post-experiment to measure particle size distributions and dendricity. This would greatly enhance the study's credibility.

**Response:** We thank the reviewer for this valuable suggestion. We have updated our model to account for the complex effects of particle shape. Specifically, we now derive

the cohesive force expression explicitly for dendritic particles and have refined the relevant parameters in the cohesion force equation using our experimental data.

Regarding analysis of post-experiment cornice particle size distributions and dendricity, we have examined both the size and morphology of snow particles on the cornice surface and edge. The particle size distribution observed is consistent in trend with the results reported in our manuscript. However, collecting snow particles from the cornice surface and edge post-experiment is easy to break the dendritic snow shape, making post-experiment measurements of their size and dendricity potentially unreliable. For this reason, we did not adopt this method for data analysis in the current study, as it could introduce significant uncertainties.

**Comment 15:**

Line 224: The authors state that wind velocity and shear stress near the cornice edge approach zero. Please provide clear evidence for this. Figure S1 is unsatisfactory and hard to interpret. Moreover, since the ridge model differs from natural terrain shown in the reference, the airflow at the edge may not reduce to zero but instead decrease rapidly on the leeward side.

**Response:** Thanks for this suggestion. Regarding the wind velocity and shear stress from numerical simulation, we have interpreted in Comment 1 in the specific comment.

There are two principal types of mountain terrain where snow cornice is commonly formed, the plateau type, characterized by relatively flat mountaintop areas with abrupt edge, and the ridge type, which refers to elongated, narrow mountain crest exposed to prevailing winds. In this work, we specifically investigate the plateau-type setting, where cornices form along the abrupt edges of relatively flat mountaintops, as is shown in Figures below. We have tested these two types of snow model and showed the plateau type mountain in this work.

**Figure.** Snow fences to the left and wind baffles to the right in use in Switzerland (photos: Stefan Margreth).

**Figure.** Snow cornices. (photo by Lea Frye in Buena Vista, Colorado, United States of America. Source: Smithsonian Magazine Photo Contest.)
[https://photocontest.smithsonianmag.com/photocontest/detail/snow-cornice-while-snowmobiling-near-cottonwood-pass-colorado/]

**Figure.** Snow cornices overhanging Gruvefjellet. (photo by Holt Hancock et al., 2022) [https://www.ntnu.no/blogger/richard-hann/2022/05/18/monitoring-the-snow-cornices-for-avalanche-risks/]

Moreover, the effect of mountain terrain on cornice growth is needed to be discussed in the future work. We have added the sentence in lines 383-385: "Numerical simulations will be essential for a more comprehensive understanding of the coupling between the flow field and snow cornice dynamics, and investigate the effects of mountain morphology on cornice growth."

**Comment 16:**

**Line 275:**

The claim about increases in both the thickness and length of the cornice should be supported by time-series data. If such data were obtained, please present them.

**Response:** We appreciate the reviewer's suggestion. As reported in Yu et al. (2022), we have previously published the time-series data illustrating the temporal evolution of cornice thickness and length. However, the present study focuses on particle motions

by observations with high temporal resolution and short time period, during which such macroscopic changes are not as evident. To avoid potential misunderstanding, we have deleted this sentence in Conclusion and added lines 100-102: "A total 18 cases were conducted during the cornice growth, with each case lasting 4-5 seconds and yielding 12455 images to record the particle trajectories. The duration of each case was limited by the camera memory. A sequence of the different growth steps is illustrated in Fig. 3 (a) of Yu et al. (2022)."

**References**

Yu, H., Li, G., Walter, B., Lehning, M., Zhang, J., and Huang, N.: Wind conditions for snow cornice formation in a wind tunnel, The Cryosphere, 17, 639–651, https://doi.org/10.5194/tc-17-639-2023, 2023.

**Additional Comment:**

Looking at Figures 4 and 6, it appears that snow accumulation on the flat surface at the model's right end increased. This area may play a key role in conveying rolling particles toward the edge and increasing the cornice root's thickness. This process could be crucial to understanding how thin plates evolve into full cornices in nature. This process deserves attention in future discussions and analyses.

**Response:** We appreciate this comment concerning the accumulation at the root of the cornice and its role in subsequent cornice growth. This area indeed plays a key role in conveying not only rolling particles but also saltating particles toward the edge. The relationship between air mass transported and the growth rate of cornice has been studied in our previous work of Yu et al. (2022). To be clear, we have incorporated a relevant description of the cornice growth process in the Introduction of the current paper. As lines 29-44: "The growth of a snow cornice can be divided into several stages (Montagne, 1980; Vogel et al., 2012; Eckerstorfer et al., 2013). In the initial stage, a thin slab forms at the mountain edge (highlighted in red in Fig. 1), mainly by adhesion of wind-transported snow particles. When more snow accumulates on the relatively flat surface behind the edge, it can gradually be conveyed toward the slab tip—especially via wind-transported particles—thereby increasing the thickness at the cornice root. This sustained supply of snow from the platform region plays a key role in the transformation of a small slab into a fully developed cornice in nature. In the subsequent stage, repeated deposition from intermittent drifting and precipitation successively adds new layers of snow to the cornice. This layer-by-layer accumulation is accompanied by a gradual increase in both length and thickness of the cornice. As the cornice grows larger, the overhanging mass of snow is increasingly influenced by gravitational forces, which may cause it to bend downward (shown in the white dashed line in Fig. 1) and promote internal compaction near the edge. Eventually, when the cornice becomes too large and shear stress exceeds a critical threshold, it breaks off and collapses. The evolution of a wedge-shaped cornice—from initial slab formation to subsequent snow accumulation on the flat surface —has been experimentally investigated in our previous work (Yu et al., 2022), with particular focus on the

relationship between cornice growth rate and air mass transport. However, the specific mechanisms governing the very initial stage, that is, how airborne snow particles first adhere and accumulate to form the incipient slab at the edge, remain unexplored."

**References**

Yu, H., Li, G., Walter, B., Lehning, M., Zhang, J., and Huang, N.: Wind conditions for snow cornice formation in a wind tunnel, The Cryosphere, 17, 639–651, https://doi.org/10.5194/tc-17-639-2023, 2023.

**Responses to Reviewer #2:**

**General comments**

**Comment 1:**

An explanation of the effect of particle shape (spherical/dendritic) on snow cornice formation has been added. The discussion of the balance of forces has been simplified without introducing excessive parameters, resulting in increased reliability, and consistency with the experimental results has also been noted. Other explanations of the experimental results have also been revised for clarity, and as a result the paper is now acceptable for publication with some corrections.

**Response:** Thank you very much for your time and efforts in reviewing our manuscript and for your constructive comments that have helped us improve the quality of this work. We appreciate your positive assessment and are grateful for the opportunity to address the reaming corrections.

**Specific comments**

Equation (3): Although this notation is understandable, it would be clearer to write the expression within the square root as  $(vpx(t))^2 + (vpy(t))^2$ .

**Response:** Thank you for pointing this out. We have revised the equation (3) to:

$$v_{\rm p}(t) = \sqrt{(v_{\rm px}(t))^2 + (v_{\rm py}(t))^2}$$
 (3)

Line 248:  $R\cos(\arcsin(x/R))$  A bracket is missing.

**Response:** Thank you for pointing this out. We have added the bracket and the sentence have been revised to: "The friction force  $F_f$  acts on point P through the moment  $arm\ Rcos(arcsin(x/R))$ ."

Lines 293-294: ...and snow accumulation on train bogies... It is unclear which state of accumulation is considered. If the accumulation is not due to natural wind, but rather to the movement of the train, additional factors such as different speed ranges and mechanical heat generation would have to be considered.

**Response:** We agree on this point. We have deleted the description on the train bogies. This sentence has been revised as lines 379-381: "Our experiments and findings enhance predictions of cornice growth and avalanche risk, with broader implications for understanding snow adhesion on both natural features and infrastructure, such as ice crevasse formation and wire icing."

Lines 327-330: The same paper by Eckerstorfer is listed as 2013a and 2013b. Line 351: The author Seligman is listed twice.

**Response:** Thank you for pointing this out. We have deleted the repeated reference.

---

## Author Response (AR3)

**Responses to Editor:**

Title: Snow Particle Motion in Process of Cornice Formation

ID: egusphere-2024-2458

Authors: Hongxiang Yu, Guang Li, Benjamin Walter, Jianping Huang, Ning Huang, and

Michael Lehning

Submitted to: The Cryosphere

The comments are in blue. Page and line numbers refer to the revised manuscript version with changes marked in italic.

**Dear authors,**

Having reviewed your responses to the referees and the amendments made to your manuscript, I am pleased to provisionally accept your paper for publication in TC. However, I suggest a number of minor, mainly technical, corrections, which are detailed in the attached file. These mainly concern typos, grammatical issues and areas that need clarification. More generally, I recommend thoroughly proofreading the manuscript, particularly the newly revised sections, before submitting the final version.

Best regards, Guillaume Chambon / TC Topical Editor

**Response:**

Dear Editor,

We would like to thank you for your positive feedback on our revision. We greatly appreciate your time in reviewing the manuscript and for pointing out the errors. We have thoroughly proofread the manuscript, and we hope the modifications address all your concerns. Please find below a point-to-point reply to each of the comments.

Best regards, Hongxiang Yu, on behalf of all authors

**Comment 1:**

Line 57: Problem with sentence: Previous field research mainly focused on the morphology variation due to limitations of observation equipment (Vogel et al., 2012; Eckerstorfer et al., 2013; van Herwijnen and Fierz, 2014; Hancock et al., 2020), for observing how particles adhering to mountain edges are hardly realized.

**Respond:** Thanks for pointing it out. We have revised this sentence as lines 45-47: "Previous field studies (Vogel et al., 2012; Eckerstorfer et al., 2013; van Herwijnen and Fierz, 2014; Hancock et al., 2020) mainly focused on the morphological variation, constrained by the limitations of observation equipment, making it difficult to observe how particles adhere to the mountain edges."

**Comment 2:**

Line 94: Redundant: A ridge model with a fixed size (Fig. 2) is built with compacted snow before each experiment. A ridge model with a fixed size (height 0.125 m, total length 0.4 m, flat surface length 0.1 m) is built with compacted snow before each experiment, and its side view is shown in Fig. 2.

**Respond**: We have deleted the first sentence, and it has been revised as lines 82-83: "A ridge model with a fixed size (height 0.125 m, total length 0.4 m, flat surface length 0.1 m) is built with compacted snow before each experiment, and its side view is shown in Fig. 2."

**Comment 3:**

Line 114: What are these different cases? Do they correspond to different experiments, or were they all recorded during the same experiment?

**Respond**: Yes, they are different cases under the same experimental conditions. This sentence has been revised as lines 98-100: "A total of 18 repeated individual experiments were conducted under the same experimental conditions. Each experiment lasted 4-5 seconds and produced 12455 images of particle trajectories."

**Comment 4:**

Line 115: Please indicate during which phase of cornice growth the images are taken. The introduction, as well as your responses to Reviewer#1, suggest that you only focus here on the initial stage of cornice formation. If so, this should probably be mentioned.

**Respond:** This sentence has been revised as lines 101-103: "Here we concentrate on the initial stage  $(t_1-t_3)$  of snow cornice growth, as described in Fig. 3(a) by Yu et al. (2022). Where particles accumulate at the edge to form a thin slab on the leeward side of the ridge model."

**Comment 5:**

**Line 145: Notations (P i, A i) are not defined**

Respond: Actually, notations  $(P_i, A_i)$  are P and  $A_p$ . This formula has been revised in line 132: " $SSA = \frac{4P}{\pi A_p}$ " and the notation has been added into the table NOTATION.

**Comment 6:**

Line 148: Which frames? Please clarify what is meant here.

**Respond:** The frames refer to the frames in which each particle is detected along its trajectory in the time-series. This sentence has been revised in lines 133-135: "For each dendritic particle, the projected area  $A_p$  and perimeter P were averaged over the frames in which the particle was detected along its trajectory in the time-series, which effectively minimizes the influence of particle orientation on the calculated size."

**Comment 7:**

Line 175: Why giving the error on particle diameter here? It would make more sense to give it in 2.1. Furthermore, can you clarify what is meant by "visual observation", and how exactly these errors are estimated?

**Respond:** We have moved the error on particle diameter to section 2.1. Visual observation refers to the process of determining the position and size of a particle in each frame through direct human inspection, from which its velocity and radius are subsequently derived. The error is calculated as:  $\frac{\phi_{pro} - \phi_{vis}}{\phi_{vis}}$ , where  $\phi_{pro}$  is the value (diameter/velocity/angle) from program recognition,  $\phi_{vis}$  is the visual observation value.

We have incorporated the above explanation in section 2.1, lines 137-140: "To assess the uncertainty of the shadowgraphy technique, particle diameter information was obtained through both visual identification, where the number of pixels occupied by the particle was counted by eye, and algorithmic recognition. By comparing the two methods, we found that the maximum relative error in particle diameter identification using the shadowgraphy technique is approximately 16%."

Meantime, in section 2.2, the sentence has been revised as lines 164-166: "Similarly, we analyzed the differences between manual identification and particle tracking algorithm recognition and found that the maximum relative errors in particle velocity and angle identified by the algorithm are 5% and 18%, respectively."

**Comment 8:**

Line 234: What is meant by "retaining" here?

**Respond:** We have revised it to "settling", in line 217.

**Comment 9:**

Line 236: Unclear statement.

**Respond**: The paragraph describing creeping particles has been revised as lines 218-223: "Particles in creeping mode, predominantly larger particles, exhibit two distinct patterns of adherence to the snow cornice surface. One pattern involves particles slowly rolling forward and stopping at a certain point, referred to as 'creeping' (Fig. 8(a)), while the other involves particles rolling to the edge, where they are captured through interlocking with the dendritic structure, termed 'hanging' (Fig. 8(b)). These two patterns together account for approximately 14% of the total adhered particles and typically move slowly."

**Comment 10:**

Fig 8: The link between these "patterns" and the two "modes" (creep and saltation) described in the text, should be better discussed. In particular, the "hanging" pattern does not seem to be mentioned in the text.

**Respond:** We have rewritten section 3.2, lines 216-227:

"Near-surface moving particles were captured using a high-speed camera. Creeping (particles rolling or sliding over the surface before settling) and saltating (particles successively jumping over the surface before settling) (Bagnold, 2012) are the two primary modes contributing to cornice growth. Particles in creeping mode, predominantly larger particles, exhibit two distinct patterns of adherence to the snow cornice surface. One pattern involves particles slowly rolling forward and stopping at a certain point, referred to as 'creeping' (Fig. 8(a)), while the other involves particles rolling to the edge, where they are captured through interlocking with the dendritic structure, termed 'hanging' (Fig. 8(b)). These two patterns together account for approximately 14% of the total adhered particles and typically move slowly. In contrast, most of the adhered particles are transported primarily via saltation. Among these, approximately 82% of the total adhered particles deposit before reaching the front end of the snow cornice, which we refer to as 'impact' (Fig. 8(c)). About 4% of the total adhered particles are saltating particles that detach from the edge, move back, and are recaptured by the snow cornice edge, influenced by reflux vortex or the potential electric field, a process we term 'back moving' (Fig. 8d)."

**Comment 11**

Line 258: Problem with sentence: Here, we define the vim is the impact velocity of the particle

**Respond:** We have revised this sentence in line 229: "Here,  $v_{im}$  is defined as the impact velocity of the particle."

**Comment 12**

Line 271: "The relative frequencies of vim and  $\theta$ im represent the probabilities of particle adhesion on the cornice with a certain impact velocity or impact angle." Is this really a probability of adhesion, since only adhered particles are considered in the statistics?

**Respond:** To avoid misunderstanding, we have revised this sentence in lines 239-240: "The relative frequencies of  $v_{im}$  and  $\theta_{im}$  represent the probabilities of adhesion particles with a certain impact velocity or impact angle range."

**Comment 13:**

**Line 274: edge particle velocities**

**Respond:** We have revised this sentence to: "Specifically, the relative frequency of impact velocities of edge particles follows the exponential function...", in lines 241 to 242. To be consistent in the whole manuscript, we have revised other places with the same expression.

**Comment 14:**

Line 282: Is this process (erosion or fracture induced by impacts) effectively observed in the experiments? If so, why not describing it earlier?

**Respond:** Yes, erosion or fracture induced by impacts is observed in the experiments. Erosion and fracture are one of the processes during cornice growth. However, this work only focuses on the particle adhesion mechanism during cornice growth, instead of providing a detailed introduction to the erosion and fracture processes. We mention it here to distinguish the adhesion mechanism between surface and edge particles.

**Comment 15:**

Line 300: Not very clear

**Respond:** This sentence has been revised, in lines 269-271: "The differences in impact velocity and angle distribution between surface and edge particles are due to the variations in the fluid field caused by unique topography at the edge, where a sudden change in velocity and pressure (Yu et al., 2025)."

**Comment 16:**

Line 304: "can be affected", or "is mainly affected"?

**Respond**: We have revised to "is mainly affected", in 274.

**Comment 17:**

Line 310: topo

**Respond**: We have revised to "particle", in line 278.

**Comment 18:**

Line 323: not defined:

**Respond**: We have added the definition in line 300: "where  $\mu_f$  is the friction coefficient of the ice surface."

**Comment 19:**

Line 337: What does this mean? I would think that the cohesive force is applied at the contact point between the two particles (here, the center of the bond).

**Respond:** Yes, the cohesive force is applied at the contact point. To avoid misunderstanding, we have deleted this sentence and reorganized the whole section 3.4.

**Comment 20:**

Line 340: Please better explain this expression.

**Respond:** This sentence has been revised as lines 295-297: "The friction force  $F_f$  acts at P through moment arm  $R\cos(\arcsin(x/R))$ , which is the perpendicular distance from the line of action of  $F_f$  to the point P, projected along the particle-center line."

**Comment 21:**

Line 344: What is relation between this equation and Eq. (8)? Furthermore, what is the point of Eq. (8), since it does not seem to be used in the following analysis?

**Respond**: To make it more concise, we have deleted Eq. (8) and the relevant description, and have revised the whole section 3.4 as:

"To investigate the effect of dendricity on the particle adherence at the edge, forces acting on a particle adhering to the edge are analyzed in this section.

Considering the differences in particle size distribution between the edge particles and surface particles, we conducted a static analysis of the particles at the edge. As shown in Fig. 10, a newly deposited particle i adheres to the foremost particle j at the edge of the cornice. Particle i is subjected to gravity  $F_g$ , the bond cohesive force  $F_c$  exerted by particle j, and the frictional force  $F_f$  at the contact surface. Due to the separation of flow, the wind velocity and surface shear stress near the edge of the cornice are close to zero (Shehadi, 2018; DeBonis, 2022; Yu et al., 2025), allowing the drag and lift forces acting on particle i to be neglected compared to other forces (Schmidt, 1980).

The force and moment balance equations for particle i can be expressed as:

$$F_g \cos \alpha + F_c - F_s = 0 (5)$$

$$F_a \sin \alpha - F_f = 0 \tag{6}$$

$$(F_s - F_c)x + F_a R \sin\beta - F_f R \cos(\arcsin\delta) = 0 \tag{7}$$

where  $\alpha$  is the angle between the line connecting the centers of the two particles and the direction of gravity, x is the radius of the bond (blue shadowed area). The radius of

the bond can be assumed to vary linearly with particle radius  $x=\delta R$ , with ratio  $\delta=0.1-0.25$  (Golubev and Frolov, 2001). The gravity force  $F_g$  acts at point P through the moment arm  $R\sin\beta$ , with the angle between Fg and line OP given by  $\beta=\alpha-\arcsin\delta$ . The friction force  $F_f$  acts at P through moment arm  $R\cos(\arcsin\delta)$ , which is the perpendicular distance from the line of action of  $F_f$  to the point P, projected along the particle-center line.  $F_s$  is the support force exerted by particle  $f_f$  on particle  $f_f$ . The frictional force  $f_f$  is given by:

$$F_f = \mu_f F_s \tag{8}$$

where  $\mu_f$  is friction coefficient.

Substituting Eq. (5), Eq. (6) and Eq. (8) into Eq. (7) yields:

$$\frac{F_c}{F_g} = \frac{\delta cos\alpha + sin(\alpha - arcsin\delta) - \mu_f cos\alpha cos(arcsin\delta)}{\mu_f cos(arcsin\delta)} \tag{9}$$

The bond cohesive force  $F_c$  for a spherical particle is given by (Szabo and Schneebeli, 2007):

$$F_c = \pi x^2 \tau_b \tag{10}$$

where  $\tau_b$  is the bond shear stress. While for non-spherical particles, particularly those with dendritic structures, the cohesion force is higher than that of spherical particles, due to the stronger geometrical interlocking between particles. Thus, dendricity should be considered in the calculation of the cohesion force for non-spherical particles. Previous study shows that the presence of sharp edges of particles can lead to an increase in cohesion via interlocking (Vivacqua et al., 2019). Thus, here we introduce a weighting parameter A into Eq. (10) for dendritic particles:

$$F_c = \pi x^2 \tau_b (1 + A(dd - 1)) \tag{11}$$

where dd is the dendricity of non-spherical particles. For spherical particles, dd=1, Eq. (11) is equivalent to Eq. (10). The weighting parameter A for snow should be determined by the experiment. For those particles that adhere on the edge in our experiment, the average value of dendricity dd=1.9, as is shown in Fig. 7."

**Comment 22:**

Line 346: This definition is already provided in line 324. Is "sigma" the same quantity as "tau b"? Please better define this quantity.

**Respond**: We have deleted the repeated definition. Yes, they are the same quantity. We have unified the quantity with the  $\tau_b$  in the manuscript.

**Comment 23:**

Line 351: Can you provide a better justification for this expression? Is it supported by previous studies?

**Respond**: We have added the reference for this expression, in lines 308-309: "Previous study shows that the sharp edges of particles can lead to an increase in cohesion via interlocking (Vivacqua et al., 2019). Thus, here we introduce a weighting parameter A into Eq. (10) for dendritic particles:"

**Comment 24:**

Line 363: A division seems to be missing here.

**Respond**: We have revised Eq. (14) to:

$$\varphi = \frac{\delta cos\alpha + \sin(\alpha - arcsin\delta) - \mu_f cos\alpha \cos(arcsin\delta)}{\mu_f \cos(arcsin\delta)}$$

**Comment 25:**

Line 386: unclear

**Respond**: have revised this sentence in line 342: "For each  $\delta$ , the threshold radius  $R_{th}$  increases linearly with dendricity, as is shown in the Fig. 12."

---

## Author Response (AR4)

**Responses to Editor:**

Title: Snow Particle Motion in Process of Cornice Formation

ID: egusphere-2024-2458

Authors: Hongxiang Yu, Guang Li, Benjamin Walter, Jianping Huang, Ning Huang, and

Michael Lehning

Submitted to: The Cryosphere

The comments are in blue. Page and line numbers refer to the revised manuscript

version with changes marked in italic.

**Dear authors,**

I am pleased to accept your paper for publication in TC.

I just have one final technical query: I may have missed it, but I could not find the definition of times t 1 and t 3 mentioned in line 102 of the revised manuscript.

Best regards, Guillaume Chambon / TC Topical Editor

**Response**

Dear Editor,

Thank you very much for pointing this out. The  $t_1$  and  $t_3$  refer to the time step in our previous work of Yu et al. (2022b). To avoid any misunderstanding, we have revomed these terms from the manuscript. The sentence has been revised to: "A sequence of the different growth steps is illustrated in Fig. 3(a) of Yu et al. (2022b). Here we only concentrate on the initial stage of snow cornice growth, where particles accumulate at the edge to form a thin slab on the leeward side of the ridge model."

Best regards,

Hongxiang Yu, on behalf of all authors